# Copresheaf Topological Neural Networks:
# A Generalized Deep Learning Framework

**Mustafa Hajij**[1,2]  **Lennart Bastian**[3,7]  **Sarah Osentoski**[1]

**Hardik Kabaria**[1]  **John L. Davenport**[1]  **Sheik Dawood**[1]

**Balaji Cherukuri**[1]  **Joseph G. Kocheemoolayil**[1]  **Nastaran Shahmansouri**[1]

**Adrian Lew**[4]  **Theodore Papamarkou**[5]  **Tolga Birdal**[6]

[1]Vinci4D  [2]University of San Francisco  [3]Technical University of Munich
[4]Stanford University  [5]PolyShape  [6]Imperial College London  [7]MCML

## Abstract

We introduce copresheaf topological neural networks (CTNNs), a powerful unifying framework that encapsulates a wide spectrum of deep learning architectures, designed to operate on structured data, including images, point clouds, graphs, meshes, and topological manifolds. While deep learning has profoundly impacted domains ranging from digital assistants to autonomous systems, the principled design of neural architectures tailored to specific tasks and data types remains one of the field's most persistent open challenges. CTNNs address this gap by formulating model design in the language of copresheaves, a concept from algebraic topology that generalizes most practical deep learning models in use today. This abstract yet constructive formulation yields a rich design space from which theoretically sound and practically effective solutions can be derived to tackle core challenges in representation learning, such as long-range dependencies, oversmoothing, heterophily, and non-Euclidean domains. Our empirical results on structured data benchmarks demonstrate that CTNNs consistently outperform conventional baselines, particularly in tasks requiring hierarchical or localized sensitivity. These results establish CTNNs as a principled multi-scale foundation for the next generation of deep learning architectures.

## 1  Introduction

Deep learning has excelled by exploiting structural biases, such as convolutions for images [Krizhevsky et al., 2012], transformers for sequences [Vaswani et al., 2017], and message passing for graphs [Gilmer et al., 2017]. However, the design of architectures that generalize across domains with complex, irregular, or multiscale structure remains a notorious challenge [Bronstein et al., 2017, Hajij et al., 2023b]. Real-world data, which span physical systems, biomedical signals, and scientific simulations, rarely adhere to the regularity assumptions embedded in conventional architectures. These data are inherently heterogeneous, directional, and hierarchical, often involving relations beyond pairwise connections or symmetric neighborhoods.

Convolutional neural networks (CNNs), designed for uniform grids, do not fully capture local irregularities; graph neural networks (GNNs) often rely on homophily and tend to oversmooth feature representations as depth increases; and transformers, while excellent at capturing long-range dependencies, assume homogeneous embedding spaces, incur quadratic complexity, and lack built-in notions of anisotropy or variable local structures. These shortcomings highlight the need for a framework that can natively encode diverse local behaviors, respect directional couplings, and

propagate information across scales without sacrificing local variations or imposing unwarranted homogeneity.

To address this foundational gap, we propose *copresheaf topological neural networks (CTNNs)*, a unifying framework for deep learning based on *copresheaves*, a categorical structure that equips each local region of a domain with its own feature space, along with learnable maps specifying how information flows between regions. Unlike traditional models that assume a global latent space and isotropic propagation, our framework respects local variability in representation and directional flow of information, enabling architectures that are multiscale, anisotropic and expressive.

By constructing CTNNs on *combinatorial complexes (CCs)* [Hajij et al., 2023b,a], which generalize graphs, simplicial complexes, and cell complexes, we enable a principled message-passing mechanism over general topological domains formulated within the theory of copresheaves. This unified perspective subsumes many deep learning paradigms, including GNNs, attention mechanisms, sheaf neural networks [Hansen and Ghrist, 2019b, Bodnar et al., 2022], and topological neural networks [Papillon et al., 2023, Hajij et al., 2023b, Bodnar et al., 2021a, Ebli et al., 2020, Giusti et al., 2023] within a single formalism. Our approach further departs from the traditional assumption of a single shared latent space by modeling task-specific, directional latent spaces that bridge diverse deep learning frameworks. Furthermore, CTNNs flexibly handle both Euclidean and non-Euclidean data, supporting expressive architectures such as copresheaf GNNs, transformers, and CNNs,

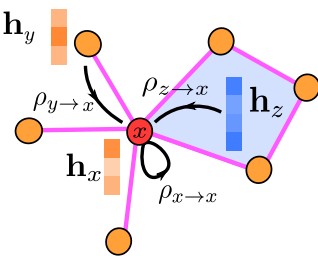

Figure 1: A copresheaf topological neural network (CTNN) operates on combinatorial complexes (CCs), which generalize Euclidean grids, graphs, meshes, and hypergraphs. A CTNN is characterized by a set of *locally indexed* copresheaf maps $\rho_{x_i \to x_j}$, defined between cells $x_i$ and $x_j$ in the CC, and directed from $x_i$ to $x_j$. The figure illustrates how a CTNN updates a local representation $\mathbf{h}_x$ of a cell $x$ using neighborhood representations $\mathbf{h}_y$ and $\mathbf{h}_z$, which are sent to $x$ via the learnable local copresheaf maps $\rho_{x \to x}$, $\rho_{y \to x}$ and $\rho_{z \to x}$.

which learn structure-aware, directional transport maps. CTNNs offer a promising framework for developing next-generation, topologically informed, structure-aware machine learning models. See Figure 1 for an illustration.

## 2 Related Work

Our work here is related to sheaf neural networks (SNNs), which extend traditional GNNs by employing the mathematical framework of cellular sheaves to capture higher-order or heterogeneous relationships. Early work by Hansen and Ghrist [2019a,b] introduced methods to learn sheaf Laplacians from smooth signals and developed a spectral theory that connects sheaf topology with graph structure. Building on these ideas, Hansen and Gebhart [2020] proposed the first SNN architecture, demonstrating that incorporating edge-specific linear maps can improve performance on tasks involving asymmetric or heterogeneous relations.

Recent advances have focused on mitigating common issues in GNNs, such as oversmoothing and heterophily. For instance, Bodnar et al. [2022] introduced neural sheaf diffusion processes that address these challenges by embedding topological constraints into the learning process. Similarly, Barbero et al. [2022a,b] developed connection Laplacian methods and attention-based mechanisms that further enhance the expressiveness and efficiency of SNNs. The versatility of the sheaf framework has also been demonstrated through its extension to hypergraphs and heterogeneous graphs [Duta et al., 2023, Braithwaite et al., 2024], which enables modeling of higher-order interactions. Moreover, novel approaches incorporating joint diffusion processes [Hernandez Caralt et al., 2024] and Bayesian formulations [Gillespie et al., 2024] have improved the robustness and uncertainty quantification of SNNs. Finally, the application of SNNs in recommender systems [Purificato et al., 2023] exemplifies their practical utility in real-world domains. Together, these contributions demonstrate the potential of SNNs to enrich graph-based learning by integrating topological and geometric information directly into neural architectures. Our proposed CTNNs generalize these architectures while avoiding restrictive co-boundary maps or rank-specific Laplacian operators. Appendix I provides a more thorough literature review of related work.

# 3 Preliminaries

This section presents preliminary concepts needed for developing our theoretical framework. It revisits CCs and neighborhood structures, reviews sheaves and copresheaves on directed graphs, and compares cellular sheaves with copresheaves in graph-based modeling.

## 3.1 Combinatorial Complexes and Neighborhood Structures

To ensure generality, we base our framework on CCs [Hajij et al., 2023b,a], which unify set-type and hierarchical relations over which data are defined. CC-neighborhood functions then formalize local interactions forming a foundation for defining sheaves and higher-order message passing schemes for CTNNs.

**Definition 1** (Combinatorial complex [Hajij et al., 2023b]). A CC is a triple $(\mathcal{S}, \mathcal{X}, \mathrm{rk})$, where $\mathcal{S}$ is a finite non-empty set of vertices, $\mathcal{X} \subset \mathcal{P}(\mathcal{S}) \setminus \{\emptyset\}$, with $\mathcal{P}(\mathcal{S})$ denoting the power set of $\mathcal{S}$, and $\mathrm{rk} : \mathcal{X} \to \mathbb{Z}_{\geq 0}$ is a rank function such that if $\{s\} \in \mathcal{X}$, $\mathrm{rk}(\{s\}) = 0$ for all $s \in \mathcal{S}$, and $x \subseteq y \implies \mathrm{rk}(x) \leq \mathrm{rk}(y)$ for all $x, y \in \mathcal{X}$.

When context permits, we write a CC $(\mathcal{S}, \mathcal{X}, \mathrm{rk})$ simply as $\mathcal{X}$. Each $x \in \mathcal{X}$ has rank $\mathrm{rk}(x)$, and $\dim \mathcal{X} = \max_{x \in \mathcal{X}} \mathrm{rk}(x)$. We refer to elements of $\mathcal{X}$ by *cells*. The $k$-cells $x^k$ of $\mathcal{X}$ are defined to be the cells $x$ with $\mathrm{rk}(x) = k$. We use the notation $\mathcal{X}^k = \{x \in \mathcal{X} : \mathrm{rk}(x) = k\} = \mathrm{rk}^{-1}(\{k\})$. See Fig 2 for an example.

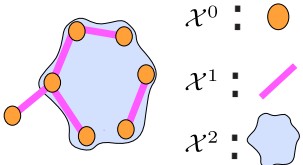

Figure 2: A combinatorial complex of dimension 2.

**Definition 2** (Neighborhood function). A *neighborhood function* on a CC $(\mathcal{S}, \mathcal{X}, \mathrm{rk})$ is a map $\mathcal{N} : \mathcal{X} \to \mathcal{P}(\mathcal{X})$, which assigns to each cell $x$ in $\mathcal{X}$ a collection of neighbor cells $\mathcal{N}(x) \subset \mathcal{X}$, referred to as *the neighborhood* of $x$. In our context, two neighborhood functions are commonly used, namely the *adjacency neighborhood* $\mathcal{N}_{\mathrm{adj}}(x) = \{y \in \mathcal{X} \mid \exists z \in \mathcal{X} : x \subset z, y \subset z\}$ and the *incidence neighborhood* $\mathcal{N}_{\mathrm{inc}}(x) = \{y \in \mathcal{X} \mid x \subset y\}$.

In practice, neighborhood functions are stored via matrices called *neighborhood matrices*.

**Definition 3** (Neighborhood matrix). Let $\mathcal{N}$ be a neighborhood function on a CC $\mathcal{X}$. Let $\mathcal{Y} = \{y_1, \ldots, y_n\} \subset \mathcal{X}$, $\mathcal{Z} = \{z_1, \ldots, z_m\} \subset \mathcal{X}$ be two collections of cells such that $\mathcal{N}(y_j) \subseteq \mathcal{Z}$ for all $1 \leq j \leq n$. An element of the *neighborhood matrix* $\mathbf{G} \in \{0,1\}^{m \times n}$ is defined as

$$[\mathbf{G}]_{ij} = \begin{cases} 1 & \text{if } z_i \in \mathcal{N}(y_j), \\ 0 & \text{otherwise.} \end{cases}$$

The copresheaf structure that we develop on CCs depends on the neighborhood function. To introduce it, we first review sheaves and copresheaves on graphs, and then extend these notions to CCs.

## 3.2 Sheaves and Copresheaves on Directed Graphs

The *copresheaf* formalism assigns each vertex its unique feature space $\mathcal{F}(x)$, respecting the potentially *heterogeneous* nature of the data, and each directed edge a transformation $\rho_{x \to y}$ that tells *how* data move between those spaces, $\mathcal{F}(x) \to \mathcal{F}(y)$. This separation between *where* data reside and *how* they travel provides a foundation for learning beyond the single-latent-space assumption of standard deep learning architectures. Concretely, copresheaves are defined as follows[1]:

**Definition 4** (Copresheaf on directed graphs). A *copresheaf* $(\mathcal{F}, \rho, G)$ on a directed graph $G = (V, E)$ is given by

- a real vector space $\mathcal{F}(x)$ for every vertex $x \in V$;
- a linear map $\rho_{x \to y} : \mathcal{F}(x) \longrightarrow \mathcal{F}(y)$ for every directed edge $x \to y \in E$.

Think of a copresheaf as a system for sending messages across a network, where each node has its own language (stalk), and edges translate messages (linear maps) to match the recipient's language. More specifically, on a directed graph $G = (V, E)$, each vertex $x \in V$ carries a task-specific latent space

---

[1]While we avoid overly complicated jargon, the appendix links our constructs to those of category theory for a more rigorous exposition.

$\mathcal{F}(x)$, and every edge $x \to y \in E$ applies a learnable, *edge-indexed* linear map $\rho_{x \to y} : \mathcal{F}(x) \to \mathcal{F}(y)$ that re-embeds $x$'s features into $y$'s coordinate frame, thus realizing directional, embedding-level message passing throughout the network.

While much of the recent literature has focused on *sheaf learning* [Ayzenberg et al., 2025], our approach is based on a copresheaf perspective. This setup departs from the traditional deep learning core assumption of a *single, shared latent space*, enabling the modeling of heterogeneous, task-specific latent spaces and directional relations. The significance of this approach lies in its ability to generalize and connect different deep learning paradigms. Copresheaf-type architectures extend beyond SNNs [Hansen and Gebhart, 2020, Bodnar et al., 2022, Barbero et al., 2022b, Duta et al., 2023, Battiloro et al., 2024b] and TNN architectures [Hajij et al., 2023b], typically designed for non-Euclidean data, by also accommodating Euclidean data effectively. This versatility allows them to unify applications across diverse data domains and architectural frameworks, providing a unified structure that uses directional information flow and adapts to task-specific requirements.

In graph-based modeling, *cellular sheaves* provide a formal framework to ensure data consistency in undirected graphs by encoding symmetric local-to-global relations via an incidence structure between vertices and edges. These structural points, formalized in the following definition, are encoded through *restriction maps*, ensuring data consistency between vertices and incident edges.

**Definition 5** (Cellular sheaf). Let $G = (V, E)$ be a undirected graph. Let $x \trianglelefteq e$ indicate that vertex $x \in V$ is incident to edge $e \in E$. A *cellular sheaf* on $G$ consists of:

- a vector space $\mathcal{F}(x)$ to each vertex $x \in V$;
- a vector space $\mathcal{F}(e)$ to each edge $e \in E$;
- a linear *restriction map* $\mathcal{F}_{x \trianglelefteq e} : \mathcal{F}(x) \to \mathcal{F}(e)$ for each incidence $x \trianglelefteq e$.

For any cell $c$ (vertex or edge), the vector space $\mathcal{F}(c)$ is typically called the *stalk at c*. The data on nodes $x$ and $y$, denoted by $\mathbf{h}_x \in \mathcal{F}(x)$ and $\mathbf{h}_y \in \mathcal{F}(y)$, "agree" on the edge $e$ if their images under the restriction maps coincide:

$$\mathcal{F}(x) \xrightarrow{\mathcal{F}_{x \trianglelefteq e}} \mathcal{F}(e) \xleftarrow{\mathcal{F}_{y \trianglelefteq e}} \mathcal{F}(y). \tag{1}$$

A *global section* of a sheaf on a graph $G$ assigns data $\mathbf{h}_v \in \mathcal{F}(v)$ to each vertex $v$ and $\mathbf{h}_e \in \mathcal{F}(e)$ to each edge $e$, such that for every edge $e$ between nodes $x$ and $y$ it holds that $\mathcal{F}_{x \trianglelefteq e}(\mathbf{h}_x) = \mathcal{F}_{y \trianglelefteq e}(\mathbf{h}_y)$. This *consistency condition* ensures data consistency across local connections in a network. Global sections represent equilibrium states, where this local agreement holds across the entire graph, enabling unified data representations for a complex system. Most sheaf-based architectures have focused on diffusion-type models [Hansen and Gebhart, 2020, Bodnar et al., 2022, Barbero et al., 2022b, Duta et al., 2023, Battiloro et al., 2024b], where the *sheaf Laplacian* $\Delta_{\mathcal{F}}$ minimizes the Dirichlet energy, ensuring global consistency. Precisely, let $C^0(G; \mathcal{F}) = \bigoplus_{v \in V} \mathcal{F}(v)$ and $C^1(G; \mathcal{F}) = \bigoplus_{e \in E} \mathcal{F}(e)$ denote the spaces of vertex-valued and edge-valued cochains of the sheaf $\mathcal{F}$, respectively. Then for some arbitrary choice of orientation for each edge, define the *coboundary map*

$$\delta : C^0(G; \mathcal{F}) \to C^1(G; \mathcal{F}), \qquad (\delta\mathbf{h})_e = \mathcal{F}_{x \trianglelefteq e}(\mathbf{h}_x) - \mathcal{F}_{y \trianglelefteq e}(\mathbf{h}_y) \quad \text{for } e = (x, y) \in E, \tag{2}$$

which measures local disagreement with respect to the edge $e$. The *sheaf Laplacian* $\Delta_{\mathcal{F}} = \delta^T \delta$ aggregates all the restriction maps $\{\mathcal{F}_{x \trianglelefteq e}\}$ into a single symmetric, positive semidefinite operator. Its associated quadratic form, $\mathbf{h}^T \Delta_{\mathcal{F}} \mathbf{h}$, has a trace that defines the *sheaf Dirichlet energy*.

Unlike sheaves, which ensure data consistency across overlaps, *copresheaves* model directional data flow, making them well-suited for processes such as information propagation, causality, and hierarchical dependencies. Copresheaves assign vector spaces $\mathcal{F}(x)$ only to vertices and define learnable linear maps $\rho_{x \to y} : \mathcal{F}(x) \to \mathcal{F}(y)$ along directed edges, without imposing sheaf consistency constraints. This vertex-centric, anisotropic framework naturally integrates with message-passing architectures such as GNNs and TNNs, allowing parameterized maps to adapt during training. See Appendix B.3 for further discussion, and Appendix C for the definition and properties of the *copresheaf Laplacian*.

## 4 Copresheaf Topological Neural Networks

We are now ready to introduce copresheaf topological neural networks (CTNNs), a higher-order message-passing mechanism that generalizes the modeling of relational structures within TNNs, as

illustrated in Figure 1. We begin by making use of copresheaves induced by neighborhood functions on CCs, providing a structured way to model general, local-to-global relations.

Let $\mathcal{X}$ be a CC and let $\mathcal{N} : \mathcal{X} \to \mathcal{P}(\mathcal{X})$ be a neighborhood function. We define the *effective support* of $\mathcal{N}$ as the set $\mathcal{X}_{\mathcal{N}} := \{x \in \mathcal{X} \mid \mathcal{N}(x) \neq \emptyset\}$. This set identifies cells that receive input from neighbors. The neighborhood function $\mathcal{N}$ induces a directed graph $G_{\mathcal{N}} = (V_{\mathcal{N}}, E_{\mathcal{N}})$, where the vertex set is

$$V_{\mathcal{N}} := \mathcal{X}_{\mathcal{N}} \cup \bigcup_{x \in \mathcal{X}} \mathcal{N}(x),$$

and the edge set is $E_{\mathcal{N}} := \{y \to x \mid x \in \mathcal{X}, y \in \mathcal{N}(x)\}$. The vertex set $V_{\mathcal{N}}$ includes both cells with non-empty neighborhoods (targets) and their neighbors (sources). This graph determines how data propagates across the complex, with each edge encoding a directional relation from neighbor $y$ to target $x$.

**Definition 6** (Neighborhood-dependent copresheaf). Let $\mathcal{X}$ be a CC, $\mathcal{N}$ a neighborhood function, and $G_{\mathcal{N}} = (V_{\mathcal{N}}, E_{\mathcal{N}})$ the induced directed graph with $V_{\mathcal{N}} = \mathcal{X}_{\mathcal{N}} \cup \bigcup_{x \in \mathcal{X}} \mathcal{N}(x)$. An $\mathcal{N}$-dependent copresheaf assigns a vector space $\mathcal{F}(x)$ to each $x \in V_{\mathcal{N}}$, and a linear map $\rho_{y \to x} : \mathcal{F}(y) \to \mathcal{F}(x)$ for each edge $y \to x \in E_{\mathcal{N}}$.

See Appendix B.4 for concrete examples. When clear from context, we simplify the notation from $\mathcal{F}^{\mathcal{N}}$ to $\mathcal{F}$ and $\rho_{y \to x}^{\mathcal{N}}$ to $\rho_{y \to x}$.

**Copresheaf neighborhood matrices**. Having introduced neighborhood matrices as binary encodings of local interactions, we now generalize this notion to define *copresheaf neighborhood matrices* (CNMs). Instead of binary entries, CNMs consist of copresheaf maps between data assigned to cells in a CC, allowing richer encoding of local dependencies. Subsequently, we define specialized versions, such as *copresheaf adjacency and incidence matrices*, that capture specific topological relations, facilitating structured message-passing in our CTNNs.

In particular, define the *k-cochain space* to be the direct sum $C^k(\mathcal{X}, \mathcal{F}^{\mathcal{N}}) = \bigoplus_{x \in \mathcal{X}^k} \mathcal{F}^{\mathcal{N}}(x)$, and denote by $\mathrm{Hom}(\mathcal{F}^{\mathcal{N}}(i), \mathcal{F}^{\mathcal{N}}(j))$ the space of linear maps from the data at the $i$-th stalk to that at the $j$-th stalk. Here, the maps encode how data is transferred or transformed between neighboring cells. We next define the CNM.

**Definition 7** (Copresheaf neighborhood matrices). For a CC $\mathcal{X}$ equipped with a neighborhood function $\mathcal{N}$, let $(\mathcal{X}, \rho, G_{\mathcal{N}})$ be a copresheaf on $\mathcal{X}$. Also let $\mathcal{Y} = \{y_1, \dots, y_n\} \subseteq \mathcal{X}$ and $\mathcal{Z} = \{z_1, \dots, z_m\} \subseteq \mathcal{X}$ be two collections of cells such that $\mathcal{N}(y_j) \subseteq \mathcal{Z}$ for all $1 \leq j \leq n$. The *copresheaf neighborhood matrix of $\mathcal{N}$ with respect to $\mathcal{Y}$ and $\mathcal{Z}$* is the $m \times n$ matrix $\mathbf{G}^{\mathcal{N}}$:

$$[\mathbf{G}^{\mathcal{N}}]_{ij} = \begin{cases} \rho_{z_i \to y_j} \in \mathrm{Hom}(\mathcal{F}(z_i), \mathcal{F}(y_j)), & z_i \in \mathcal{N}(y_j), \\ 0, & \text{otherwise.} \end{cases} \tag{3}$$

The neighborhood function $\mathcal{N}$ determines the directed relationships between the cells, and the CNM encodes the interactions between cells induced by these relationships, collecting the maps of a cell from its neighbors. Next, analogous to Definition 2, we define *copresheaf adjacency* and *copresheaf incidence* matrices as specialized forms of the general CNM.

**Definition 8** (Copresheaf adjacency/incidence matrices). For fixed $r, k$, define $\mathcal{N}_{\text{adj}}^{(r,k)}(x) = \{y \in \mathcal{X}^r \mid \exists z \in \mathcal{X}^{r+k}, x \preceq z, y \preceq z\}$ and $\mathcal{N}_{\text{inc}}^{(r,k)}(x) = \{y \in \mathcal{X}^k \mid x \preceq y\}$. The *copresheaf adjacency matrix* (CAM) $\mathbf{A}_{r,k} \in \mathbb{R}^{|\mathcal{X}^r| \times |\mathcal{X}^r|}$ and *copresheaf incidence matrix* (CIM) $\mathbf{B}_{r,k} \in \mathbb{R}^{|\mathcal{X}^k| \times |\mathcal{X}^r|}$ are

$$[\mathbf{A}_{r,k}]_{ij} = \begin{cases} \rho_{y_i \to x_j} \in \mathrm{Hom}(\mathcal{F}(y_i), \mathcal{F}(x_j)) & \text{if } y_i \in \mathcal{N}_{\text{adj}}^{(r,k)}(x_j), \\ 0 & \text{otherwise.} \end{cases} \tag{4}$$

$$[\mathbf{B}_{r,k}]_{ij} = \begin{cases} \rho_{z_i \to y_j} \in \mathrm{Hom}(\mathcal{F}(z_i), \mathcal{F}(y_j)) & \text{if } z_i \in \mathcal{N}_{\text{inc}}^{(r,k)}(y_j), \\ 0 & \text{otherwise.} \end{cases} \tag{5}$$

These specialized matrices capture distinct topological relations. A CAM encodes relations in which cells share an upper cell, while a CIM encodes relations in which one cell is incident to another.

**Copresheaf-based message passing**. We now generalize traditional graph message-passing to *heterogeneous and higher-order interactions* involving cells of varying ranks or multi-way relations

going beyond pairwise connections, by explicitly incorporating copresheaf structures defined over CCs. This leads to *copresheaf-based message-passing*, a flexible and expressive tool for capturing complex, multi-scale relations in structured data.

**Definition 9** (Copresheaf message-passing neural network). Let $G = (V, E)$ be a directed graph and $(\mathcal{F}, \rho, G)$ a copresheaf. For each layer $l$ and vertex $x \in V$, let $\mathbf{h}_x^{(l)} \in \mathcal{F}(x)$. A *copresheaf message-passing neural network (CMPNN)* is a neural network whose meassage passing is defined by

$$\mathbf{h}_x^{(l+1)} = \beta\Big(\mathbf{h}_x^{(l)}, \bigoplus_{(y \to x) \in E} \alpha\big(\mathbf{h}_x^{(l)}, \rho_{y \to x} \mathbf{h}_y^{(l)}\big)\Big),$$

where $\rho_{y \to x} \colon \mathcal{F}(y) \to \mathcal{F}(x)$ is the linear map associated with edge $y \to x$, $\alpha$ is a learnable message function, $\oplus$ a permutation-invariant aggregator, and $\beta$ a learnable update function.

Definition 9 establishes a unifying framework that generalizes graph-based message passing neural networks (MPNNs) [Gilmer et al., 2017] by incorporating learnable, anisotropic linear maps associated with directed edges. This formulation subsumes many standard GNNs, such as graph convolutional networks [Kipf and Welling, 2017], graph attention networks [Veličković et al., 2018], and their variants [Veličković, 2022], by viewing message passing as copresheaf maps on directed graphs. Consequently, all architectures derived from these foundational GNN models fit within our copresheaf message-passing paradigm. Moreover, SNNs, which operate on cellular sheaves over undirected graphs, can be adapted into this framework by reinterpreting their edge-mediated transport operator as direct vertex-to-vertex morphisms on a bidirected graph. This adaptation is detailed in the following theorem.

**Theorem 1.** (SNNs are CMPNNs) Let $G = (V, E)$ be an undirected graph equipped with a cellular sheaf $\mathcal{F}$ assigning vector spaces to vertices and edges, and linear maps $\mathcal{F}_{x \trianglelefteq e}$ for each vertex $x \in e$. Then for each edge $e = \{x, y\} \in E$, the SNN message passing from $y$ to $x$, given by the composition $\mathcal{F}_{x \trianglelefteq e}^\top \circ \mathcal{F}_{y \trianglelefteq e} \colon \mathcal{F}(y) \to \mathcal{F}(x)$, can be realized as a single morphism $\rho_{y \to x} \colon \mathcal{F}(y) \to \mathcal{F}(x)$ in a copresheaf on the bidirected graph $G' = (V, E')$, $\quad E' = \{(x, y), (y, x) \mid \{x, y\} \in E\}$ by setting $\rho_{y \to x} = \mathcal{F}_{x \trianglelefteq e}^\top \circ \mathcal{F}_{y \trianglelefteq e}$.

Theorem 1 establishes that the message-passing scheme employed by SNNs can be interpreted as a special case of CMPNNs when restricted to bidirected graphs. This perspective generalizes most existing SNN architectures found in the literature, including those in Hansen and Gebhart [2020], Bodnar et al. [2022], Barbero et al. [2022b]. The map $\mathcal{F}_{x \trianglelefteq e}^\top \circ \mathcal{F}_{y \trianglelefteq e}$ arises from composing the restriction map $\mathcal{F}_{y \trianglelefteq e} \colon \mathcal{F}(y) \to \mathcal{F}(e)$ with its adjoint $\mathcal{F}_{x \trianglelefteq e}^\top \colon \mathcal{F}(e) \to \mathcal{F}(x)$, thereby capturing both vertex-to-edge and edge-to-vertex transformations defined by the cellular sheaf. As a consequence of this connection between SNNs and CMPNNs, diffusion-style updates (commonly used in sheaf-based models, such as those based on the sheaf Laplacian) can be succinctly expressed within the CMPNN framework. This result is formally stated in Proposition 1, with a full proof provided in Appendix E.

**Proposition 1** (Neural-sheaf diffusion [Bodnar et al., 2022] as copresheaf message-passing). Let $G = (V, E)$ be an undirected graph endowed with a cellular sheaf $\mathcal{F}$. Given vertex features $\mathbf{H} = [\mathbf{h}_v]_{v \in V}$ with $\mathbf{h}_v \in \mathcal{F}(v)$, and learnable linear maps $W_1, W_2$, define the diffusion update

$$\mathbf{H}^+ = \mathbf{H} - \big(\Delta_\mathcal{F} \otimes I\big)(I_n \otimes W_1)\mathbf{H}W_2, \tag{6}$$

where $\Delta_\mathcal{F} = [L_{F,v,u}]_{v,u \in V}$ has blocks $L_{F,vv} = \sum_{v \trianglelefteq e} \mathcal{F}_{v \trianglelefteq e}^\top \mathcal{F}_{v \trianglelefteq e}$, $L_{F,vu} = -\mathcal{F}_{v \trianglelefteq e}^\top \mathcal{F}_{u \trianglelefteq e}$, for $u \neq v$, $u \trianglelefteq e$, $v \trianglelefteq e$. Then, $\mathbf{H}^+$ can be expressed in the copresheaf message-passing form of Definition 9.

The next definition formalizes the notion of general multi-way propagation.

**Definition 10** (Copresheaf-based higher-order message passing). Let $\mathcal{X}$ be a CC, and $\mathfrak{N} = \{\mathcal{N}_k\}_{k=1}^n$ a collection of neighborhood functions. For each $k$, let $(\mathcal{F}^{\mathcal{N}_k}, \rho^{\mathcal{N}_k}, G_{\mathcal{N}_k})$ be a copresheaf in which the maps $\rho_{y \to x}^{\mathcal{N}_k} \colon \mathcal{F}^{\mathcal{N}_k}(y) \to \mathcal{F}^{\mathcal{N}_k}(x)$ define the transformations associated to the copresheaf. Given features $\mathbf{h}_x^{(\ell)}$, the next layer features are defined as

$$\mathbf{h}_x^{(\ell+1)} = \beta\Bigg(\mathbf{h}_x^{(\ell)}, \bigotimes_{k=1}^n \bigoplus_{y \in \mathcal{N}_k(x)} \alpha_{\mathcal{N}_k}\big(\mathbf{h}_x^{(\ell)}, \rho_{y \to x}^{\mathcal{N}_k}(\mathbf{h}_y^{(\ell)})\big)\Bigg),$$

where $\alpha_{\mathcal{N}_k}$ is the message function, $\oplus$ a permutation-invariant aggregator over neighbors $y \in \mathcal{N}_k(x)$, $\otimes$ combines information from different neighborhoods, and $\beta$ is the update function.

Table 1: A unified view **across domains and architectures**—CNNs, GNNs, Transformers, SNNs, and TNNs—as instances of **Copresheaf Topological Neural Networks (CTNNs)** defined by neighborhood graphs $G_{\mathcal{N}}$ and directional transports $\rho_{y\to x}$.

| Classical model | CTNN form | Domain / $\mathcal{N}$ | $\rho_{y\to x}$ / refs |
|---|---|---|---|
| **CNN** Li et al. [2021] | CopresheafConv | Grid; adjacency (CAM) | Translation-consistent transport (shared local filters). App. G |
| **MPNN** Gilmer et al. [2017] | CMPNN | Graph; adjacency | Shared / edge-indexed linear maps; GCN/GAT special cases. Def. 10 |
| **Euclidean / cellular Tr.** Vaswani et al. [2017], Barsbey et al. [2025] | Copresheaf Transformer | Tokens on grid/seq; full/masked adj | In-head value transport; $\rho = I \Rightarrow$ dot-product attention. Sec. 5, App. F |
| **SNN** Bodnar et al. [2022], Hansen and Gebhart [2020] | CMPNN on bidirected graph | Graph; incidence (CIM) | $\rho_{u\leftarrow v} = F_{u \triangleleft e}^{\top} F_{v \triangleleft e}$ (vertex–vertex). Prop. 1, Thm. 1, App. E, B.3 |
| **TNN (hyper-graph/simplicial/cellular/CC)** Hajij et al. [2023b] | Higher-order CMPNN | CC; multi-$\mathcal{N}$ via CAM/CIM | Rank-aware maps across overlaps; multi-way $\otimes$. Def. 10 |

*Abbrev:* CC = combinatorial complex; CAM/CIM = copresheaf adjacency/incidence; Adj/Inc = adjacency/incidence.

Definition 10 lays the foundational framework that unifies a broad class of topological deep learning architectures, bridging higher-order message passing methods, transformers and SNNs. This synthesis not only consolidates existing approaches but also opens avenues for novel architectures based on topological and categorical abstractions. Notably, the formulation in Proposition 10 encompasses simplicial message passing [Ebli et al., 2020, Bunch et al., 2020, Bodnar et al., 2021b], cellular message passing [Hajij et al., 2020, Bodnar et al., 2021a], stable message passing via Hodge theory [Hayhoe et al., 2022], and recurrent simplicial architectures for sequence prediction [Mitchell et al., 2024]. It also subsumes more recent developments that harness multiple signals and higher-order operators such as the Dirac operator [Calmon et al., 2022, Hajij et al., 2023a] and TNNs [Hajij et al., 2023b]. These diverse models are unified under the copresheaf-based formulation by interpreting neighborhood aggregation, feature transport, and signal interaction within a coherent framework. See Appendix E for derivations showing how several of these architectures emerge as special cases of this general formulation. See also Table 1 for a summary.

**Remark 1** (Graph vs CC copresheaf models). Unlike graph-based models, which propagate information edge by edge, copresheaf models on a CC aggregate messages across all overlapping neighborhoods at once. Overlapping neighborhoods have common cells, potentially at different ranks, allowing simultaneous aggregation of multi-way interactions. Applying each neighborhood function $\mathcal{N}_k$ in turn, we compute its map-driven messages and then merge them into a single update.

## 5 Architectures Derived from the Copresheaf Framework

Having established the abstract copresheaf-based framework on a CC $\mathcal{X}$, we now present several concrete instantiations. Copresheaf transformers (CTs) extend the standard attention mechanism by dynamically learning linear maps $\rho_{y\to x} : \mathcal{F}(y) \to \mathcal{F}(x)$ encoding directional, anisotropic relationships between tokens, i.e., cells, in $\mathcal{X}$. Integrating these maps into attention enables CTs to capture rich, structured interactions. Copresheaf graph neural networks (CGNNs) generalize message-passing GNNs by incorporating copresheaf linear maps to model relational structures. Copresheaf convolutional networks define convolution-like operations on CCs, modeled as a Euclidean grid, using these linear maps. We present the CT construction next and leave the exact formulations of copresheaf networks, CGNNs, and copresheaf convolution layers to the appendices E, F and G.

**Copresheaf transformers**. Having established the abstract framework of CTNNs on a CC $\mathcal{X}$, we now introduce a concrete instantiation: the *copresheaf transformer (CT)* layer. This layer extends the standard attention mechanism by dynamically learning linear maps $\rho_{y\to x} : \mathbb{R}^{d_y} \to \mathbb{R}^{d_x}$ that encode both the combinatorial structure and directional, anisotropic relationships within the complex. By integrating these maps into the attention computation, the CT layer captures rich, structured

Table 2: Mean squared error (mean ± standard deviation) of classical vs copresheaf architectures for learning various physics simulations.

| Network | Heat (Transformer) | Advection (Transformer) | Unsteady stokes (Conv-transformer) |
|---|---|---|---|
| Classical | $2.64 \times 10^{-4} \pm 3.50 \times 10^{-5}$ | $3.52 \times 10^{-4} \pm 7.70 \times 10^{-5}$ | $1.75 \times 10^{-2} \pm 1.32 \times 10^{-3}$ |
| Copresheaf | $\mathbf{9.00 \times 10^{-5} \pm 7.00 \times 10^{-6}}$ | $\mathbf{1.20 \times 10^{-4} \pm 1.20 \times 10^{-5}}$ | $\mathbf{1.48 \times 10^{-2} \pm 1.48 \times 10^{-4}}$ |

interactions across $\mathcal{X}$. At layer $\ell$, each cell $x \in \mathcal{X}$ is associated with a feature $\mathbf{h}_x^{(\ell)} \in \mathbb{R}^{d_x}$, where $d_x$ denotes the feature dimension of cell $x$.

**Definition 11** (Copresheaf self-attention). For a fixed rank $k$ and neighborhood $\mathcal{N}_k$ (e.g., adjacency between $k$-cells), let $W_q, W_k \in \mathbb{R}^{p \times d}, W_v \in \mathbb{R}^{d \times d}$ denote learnable projection matrices, where $p$ is the dimension of the query and key spaces, and $d$ is the feature dimension (assumed uniform across cells for simplicity). For each $k$-cell $x \in \mathcal{X}^k$, *copresheaf self-attention* defines the message aggregation and feature update as $\mathbf{h}_x^{(\ell+1)} = \beta(\mathbf{h}_x^{(\ell)}, m_x)$, where $m_x = \sum_{y \in \mathcal{N}_k(x)} a_{xy} \rho_{y \to x}(v_y)$ and

$$a_{xy} = \frac{\exp(\langle q_x, k_y \rangle / \sqrt{p})}{\sum_{y' \in \mathcal{N}_k(x)} \exp(\langle q_x, k_{y'} \rangle / \sqrt{p})}, \tag{7}$$

where $q_x = W_q h_x^{(\ell)}, k_x = W_k h_x^{(\ell)}$, and $v_x = W_v h_x^{(\ell)}$. Here, the softmax normalizes over all neighbors $y' \in \mathcal{N}_k(x)$ and $\rho_{y \to x} : \mathcal{F}(y) \to \mathcal{F}(x)$ is the learned copresheaf map. The update function $\beta$ is chosen to be a neural network.

Similarly, we define *copresheaf cross-attention* among $s$ and $t$ rank cells in $\mathcal{X}$, as well as a general algorithm for a corpresheaf transformer layer (Appendix F).

## 6 Experimental Evaluation

We conduct experiments on synthetic and real data in numerous settings to support the generality of our framework. These include learning physical dynamics, graph classification in homophillic and heterophilic cases and classifying higher-order complexes.

### 6.1 Evaluations on Physics Datasets

To verify the validity of our networks in toy setups of different phenomena, we generate a series of synthetic datasets. These include:

1. *Heat.* We generate 600 realisations by solving the heat equation $u_t = \nu u_{xx}$ on $[0, 1)$ with $\nu = 0.1$ to horizon $T = 0.1$; each $u_0$ is a 10-mode sine series and $u_T$ is its Gaussian-kernel convolution, sampled on $N = 100$ grid points.
2. *Advection.* Similar to heat, we generate 600 realisations of $u_t + c\, u_x = 0$ with $c = 1$; the solution is a pure phase shift of $u_0$, sampled on $N = 130$ points. Each pair is normalized to the interval $[0, 1]$ and the dataset is split into 500:100 train/test samples.
3. *Unsteady stokes.* Let $\mathbf{u}(x, y, t) = (u(x, y, t), v(x, y, t)) \in \mathbb{R}^2$ denote the incompressible velocity field, $p(x, y, t) \in \mathbb{R}$ the kinematic pressure, and $\nu > 0$ the kinematic viscosity. Throughout, $\partial_t$ is the time derivative, $\nabla = (\partial_x, \partial_y)$ the spatial gradient, and $\Delta = \partial_{xx} + \partial_{yy}$ the Laplacian operator. The periodic unsteady Stokes system reads $\partial_t \mathbf{u} - \nu \Delta \mathbf{u} + \nabla p = \mathbf{0}$, where $\nabla \cdot \mathbf{u} = 0$. We synthesize 200 samples by drawing an 8-mode Fourier stream-function $\psi$, setting $\mathbf{u}_0 = \nabla^\perp \psi$, and evolving to $T = 0.1$ with $\nu = 0.1$ via the analytic heat kernel. Each pair is sampled on a $16 \times 16$ grid, with each channel normalized to the interval $[0, 1]$, and the dataset is split into 160:40 train/test samples.

**Model and training**. For heat and advection, we use two transformer layers (positional encoding, four heads, stalk dimension equal to 16), followed by a mean pooling and linear head yielding 64D token embeddings. We train our networks using AdamW wiht a learning rate of $10^{-3}$, cosine LR scheduling, and batch size of two. We use 50 epochs for the heat dataset and 80 for the advection dataset, and report the results over three seeds. For the unsteady Stokes data, we test a compact convolution-transformer U-Net consisting of a convolutional encoder with two input and 32 output channels, followed by two transformer layers (four heads, hidden dimension equal to 32, and stalk dimension 8), and a convolutional decoder mapping back to two output channels. We train it for 300

epochs using AdamW with a batch size of four. The classical baselines use dot-product attention, whereas the copresheaf variants employ learned outer-product maps.

**Results**. As shown in Table 2, in the heat and advection tests, copresheaf attention significantly outperforms classical dot-product attention, reducing the test MSE by over $50\%$ and achieving more stable results across seeds. For unsteady Stokes, the copresheaf attention lowers the test MSE by $\approx 15\%$ and reduces variance by an order of magnitude, confirming that pair-specific linear transports capture viscous diffusion of vorticity more faithfully than standard self-attention under identical compute budgets.

## 6.2  Graph Classification

We evaluate whether incorporating copresheaf structure into GNNs improves performance on graph classification tasks.

**Data**. MUTAG dataset, a nitroaromatic compound classification benchmark consisting of $188$ molecular graphs, where nodes represent atoms and edges represent chemical bonds. Each node is associated with a 7-dimensional feature vector encoding atom type, and each graph is labeled as mutagenic or non-mutagenic (two classes). The dataset is split into $80\%$ train and $20\%$ test samples.

**Baselines, backbone and training**. We compare standard GNN models (GCN, GraphSAGE, GIN) against their copresheaf-counterparts (CopresheafGCN, CopresheafSage, CopresheafGIN) derived below. All models are two-layer networks with a hidden dimension of 32 for GCN and GraphSAGE, and 16 for GIN, followed by global mean pooling and a linear classifier to predict graph labels. The standard models (GCN [Kipf and Welling, 2017], GraphSAGE [Hamilton et al., 2017], GIN [Xu et al., 2019]) use conventional GNN convolutions: GCN with symmetric normalization, GraphSAGE with mean aggregation, and GIN with sum aggregation. The copresheaf-enhanced models augment these with learned per-edge copresheaf maps, introducing local consistency constraints via transformations. All models are trained using Adam with a learning rate of $0.01$, and a batch size of 16. GCN and GIN models are trained for $100$ epochs, while GraphSAGE for $50$. The negative log-likelihood loss is minimized, and performance is evaluated via test accuracy. For GCN and GraphSAGE, we use five runs, while GIN uses ten runs.

**Enhancing GNNs via copresheaves**. The copresheaf structure enhances each GNN by learning a transport map $\rho_{ij} = I + \Delta_{ij}$ for each edge $(i, j)$, where $\Delta_{ij}$ is a learned transformation and $I$ is the identity matrix. In what follows $D$ represents the dimension of the input feature in where we apply the copresheaf maps $\rho_{ij}$. Denote by $[\,\mathbf{h}_i; \mathbf{h}_j\,]$ to the concatenation of the two node feature vectors $\mathbf{h}_i$ and $\mathbf{h}_j$ along their feature dimension. The process for each model is as follows:

Table 3: Mean test accuracy ($\pm$ std).

| Model | Accuracy |
|---|---|
| GCN | $0.674 \pm 0.014$ |
| CopresheafGCN | $0.721 \pm 0.035$ |
| GraphSAGE | $0.689 \pm 0.022$ |
| CopresheafSage | $0.732 \pm 0.029$ |
| GIN | $0.700 \pm 0.039$ |
| CopresheafGIN | $0.724 \pm 0.021$ |

- *CopresheafGCN*. For node features $\mathbf{h}_i, \mathbf{h}_j$, compute $\Delta_{ij} = \tanh(\text{Linear}([\mathbf{h}_i; \mathbf{h}_j]))$, take its diagonal to get a diagonal $D \times D$ matrix with $D = 7$. Form $\rho_{ij} = I + \Delta_{ij}$. Aggregate neighbor features as $\mathbf{h}_i' = \sum_{j \in \mathcal{N}(i)} \frac{1}{\sqrt{d_i d_j}} \rho_{ij} \mathbf{h}_j$, where $d_i, d_j$ are the degrees of nodes $i$ and $j$, respectively. Combine with the self-feature: $\mathbf{h}_i'' = (1 + \epsilon)\mathbf{h}_i + \mathbf{h}_i'$.
- *CopresheafSage*. Compute $\Delta_{ij} = \tanh(\text{Linear}([\mathbf{h}_i, \mathbf{h}_j]))$, a diagonal matrix, and form $\rho_{ij} = I + \Delta_{ij}$. Aggregate via mean: $\mathbf{h}_i' = \text{mean}_{j \in \mathcal{N}(i)}(\rho_{ij} \mathbf{h}_j)$. Combine: $\mathbf{h}_i'' = (1 + \epsilon)\mathbf{h}_i + \mathbf{h}_i'$. The map $\rho_{ij}$ enhances local feature alignment.
- *CopresheafGIN*. Compute $\Delta_{ij} = \tanh(\text{Linear}([\mathbf{h}_i, \mathbf{h}_j]))$, a full $D \times D$ matrix, and form $\rho_{ij} = I + \Delta_{ij}$. Aggregate: $\mathbf{h}_i' = \sum_{j \in \mathcal{N}(i)}(\rho_{ij} \mathbf{h}_j)/d_i$. Combine: $\mathbf{h}_i'' = (1 + \epsilon)\mathbf{h}_i + \mathbf{h}_i'$.

**Results**. On MUTAG, copresheaf-enhanced GNNs consistently outperform their standard versions across GCN, GraphSAGE, and GIN. CopresheafSAGE achieves the highest average accuracy (0.732) and the largest relative gain over the GraphSAGE baseline (0.689). Learned per-edge transport maps better capture complex structure and enforce local consistency, improving classification. These results demonstrate the promise of copresheaf structures for molecular graph classification.

## 6.3 Combinatorial Complex Classification

Finally, we assess whether incorporating copresheaf structure into transformer-based attention mechanisms improves performance on classifying higher-order, general data structures, such as CCs. Specifically, we compare a classical transformer model against two CT variants (CT-FC and CT-SharedLoc) on a synthetic dataset of CCs.

**Data**. Our synthetic dataset comprises 200 training and 50 test CCs derived from Erdős-Rényi graphs, each with 10 nodes and a base edge probability of 0.5. Triangles are added to form higher-order structures with probability $q = 0.1$ for class 0 (low density) or $q = 0.5$ for class 1 (high density). Each node has 2D feature vectors, consisting of its degree and the number of triangles it participates in, with added Gaussian noise $\mathcal{N}(0, 0.1)$.

**Backbone and training**. All models are transformer-based classifiers with a single block, using two attention heads, an embedding dimension of 8, and a head dimension of 4. The model embeds 2D node features, applies attention, performs global average pooling, and uses a linear classifier to predict one of two classes (low or high triangle density). The classical model employs standard multi-head attention. The *CT-FC* and *CT-SharedLoc* models augment attention with learned per-edge transport maps to enforce local consistency. Models are trained for four epochs using Adam with a learning rate of $10^{-3}$ and a batch size of 8, minimizing cross-entropy loss. Performance is evaluated via test accuracy. Experiments are over four runs with different random seeds to ensure robustness.

**Copresheaf attention**. Similar to GNNs, the copresheaf structure enhances attention by learning a transport map $\rho_{ij} = I + \Delta_{ij}$. The value vector $\mathbf{v}_j$ is transformed as $\rho_{ij}\mathbf{v}_j$ before attention-weighted aggregation. The process for each model is as follows:

- *CT-FC*. For node features $\mathbf{h}_i, \mathbf{h}_j$, compute $\Delta_{ij} = \tanh(\text{Linear}([\mathbf{h}_i, \mathbf{h}_j]))$, a full $d \times d$ matrix ($d = 4$). Form $\rho_{ij} = I + \Delta_{ij}$. Apply $\rho_{ij}$ to value vectors in attention: $\mathbf{v}_t = \rho_{ij}\mathbf{v}_j$. The map $\rho_{ij}$ enables rich feature transformations across nodes.
- *CT-SharedLoc*. Compute a shared $\Delta_{ij} = \tanh(\text{MLP}([\mathbf{h}_i, \mathbf{h}_j]))$, a full $d \times d$ matrix, and a local scalar $\alpha_{ij} = \sigma(\text{MLP}([\mathbf{h}_i, \mathbf{h}_j]))$. Form $\rho_{ij} = I + \alpha_{ij}\Delta_{ij}$. Apply $\rho_{ij}$ to value vectors: $\mathbf{v}_t = \rho_{ij}\mathbf{v}_j$. The map $\rho_{ij}$ balances shared transformations with local modulation.

**Results**. Copresheaf transformer models outperform the standard transformer on the CC classification task, with CT-SharedLoc achieving the highest average accuracy (0.970) and competitive stability (std 0.010). The learned per-edge transport maps $\rho_{ij}$ enhance the model's ability to capture higher-order structural patterns, such as triangle density, by aligning node features effectively. CT-SharedLoc's combination of shared transport maps and local modulation yields the best performance, showcasing the value of copresheaf structures in transformer-based models for CC classification.

Table 4: Mean $\pm$ std test accuracy for CC classification.

| Model | Accuracy |
|---|---|
| Classic | $0.940 \pm 0.014$ |
| CT-FC | $0.955 \pm 0.009$ |
| CT-SharedLoc | $0.970 \pm 0.010$ |

## 7 Discussion and Conclusions

We proposed CTNNs, a unified deep learning framework on (un)structured data. By develping models on copresheaves over CCs, CTNNs generalize GNNs, SNNs, and TNNs through directional, heterogeneous message passing. Besides theoretical advances, CTNNs offer empirical benefits across diverse tasks, laying the principles for multiscale and anisotropic representation learning.

**Limitations and future work**. CTNNs incur additional overhead from per-edge transformations and have so far been evaluated in modest-scale settings. We plan to explore well-engineered scalable parameterizations, extend CTNNs to large-scale and dynamic domains, and further connect categorical structure with robustness and inductive bias in deep learning.

## Acknowledgments and Disclosure of Funding

Professor Adrian J. Lew's contributions to this publication were as a paid consultant and were not part of his Stanford University duties or responsibilities. T. Birdal acknowledges support from the Engineering and Physical Sciences Research Council [grant EP/X011364/1]. T. Birdal was supported by a UKRI Future Leaders Fellowship [grant number MR/Y018818/1] as well as a Royal Society

Research Grant RG/R1/241402. The authors thank Hans Riess for pointing out the relationship between the CTNNs and the quiver Laplacian.

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

# Copresheaf Topological Neural Networks:
# A Generalized Deep Learning Framework
# –Supplementary Material–

## Table of Contents

# A Notation

We provide a reference summary of the notation and acronyms used throughout the main text. Table 5 details key mathematical symbols, while Table 6 lists abbreviations and their expansions.

Table 5: Summary of key notation used throughout the paper.

| Notation | Description |
|---|---|
| $\mathcal{S}$ | Underlying vertex set of a combinatorial complex (Def. 1) |
| $\mathcal{X}$ | Set of nonempty cells $\subseteq \mathcal{P}(\mathcal{S})$ (Def. 1) |
| $\mathcal{P}(\mathcal{S})$ | Power set of $\mathcal{S}$ (Def. 1) |
| $\mathrm{rk} : \mathcal{X} \to \mathbb{Z}_{\geq 0}$ | Rank function on cells, mapping to non-negative integers (Def. 1) |
| $\mathbb{Z}_{\geq 0}$ | Non-negative integers |
| $\mathcal{X}^{\overline{k}}$ | $\{x \in \mathcal{X} : \mathrm{rk}(x) = k\}$, the $k$-cells (Def. 1) |
| $\dim \mathcal{X}$ | Dimension of the complex, $\max_{x \in \mathcal{X}} \mathrm{rk}(x)$ (Sec. 2.1) |
| $\mathcal{N} : \mathcal{X} \to \mathcal{P}(\mathcal{X})$ | Neighborhood function, mapping cells to sets of neighbor cells (Def. 2) |
| $\mathcal{N}_{\mathrm{adj}}$ | Adjacency neighborhood function (Def. 2) |
| $\mathcal{N}_{\mathrm{inc}}$ | Incidence neighborhood function (Def. 2) |
| $\mathcal{X}_{\mathcal{N}}$ | Effective support of $\mathcal{N}$, $\{x \in \mathcal{X} \mid \mathcal{N}(x) \neq \emptyset\}$ (Sec. 3) |
| $\mathbf{G} \in \{0,1\}^{m \times n}$ | Binary neighborhood matrix (Def. 3) |
| $\mathbf{G}^{\mathcal{N}}$ | Copresheaf neighborhood matrix, entries $\rho_{z_i \to y_j}$ or 0 (Def. 7) |
| $\rho_{y \to x}$ | Copresheaf morphism $\mathcal{F}(y) \to \mathcal{F}(x)$ for edge $y \to x$ (Def. 4) |
| $\mathbf{A}_{r,k}$ | Copresheaf adjacency matrix between $r$-cells (Def. 8) |
| $\mathbf{B}_{r,k}$ | Copresheaf incidence matrix between ranks $r$ and $k$ (Def. 8) |
| $C^k(\mathcal{X}, \mathcal{F})$ | $k$-cochain space, $\bigoplus_{x \in \mathcal{X}^k} \mathcal{F}(x)$ (Sec. 3) |
| $\mathrm{Hom}(\mathcal{F}(i), \mathcal{F}(j))$ | Space of linear maps from $\mathcal{F}(i)$ to $\mathcal{F}(j)$ (Def. 7) |
| $\mathbf{h}_x^{(\ell)}$ | Feature vector at cell $x$ in layer $\ell$ (Prop. 1) |
| $\alpha$ | Learnable message function (Prop. 1) |
| $\beta$ | Learnable update function (Prop. 1) |
| $\oplus$ | Permutation-invariant aggregator (Prop. 1, Prop. 3) |
| $G = (V, E)$ | Directed or undirected graph (Sec. 2.2) |
| $G_{\mathcal{N}} = (\mathcal{X}_{\mathcal{N}}, E_{\mathcal{N}})$ | Directed graph induced by $\mathcal{N}$, edges $y \to x$ if $y \in \mathcal{N}(x)$ (Def. 6) |
| $\mathcal{F}(x), \mathcal{F}(e)$ | Stalks: vector spaces at vertex $x$ or edge $e$ (Def. 4, Def. 5) |
| $\mathcal{F}_{x \leq e} : \mathcal{F}(x) \to \mathcal{F}(e)$ | Restriction map in a cellular sheaf for $x \leq e$ (Def. 5) |
| $\delta$ | Coboundary map, measures local disagreement in cellular sheaf (Sec. 2.2) |
| $\Delta_{\mathcal{F}}$ | Sheaf Laplacian, $\Delta_{\mathcal{F}} = \delta^T \delta$ (Sec. 2.2) |
| $\mathcal{Y}, \mathcal{Z}$ | Collections of cells, used in neighborhood matrices (Def. 3, Def. 7) |
| $W_q, W_k, W_v$ | Learnable projection matrices for queries, keys, values in copresheaf self-attention (Prop. 4) |
| $q_x, k_x, v_x$ | Query, key, and value vectors for cell $x$ in copresheaf self-attention (Prop. 4) |
| $a_{xy}$ | Attention coefficient for cells $x$ and $y$ in copresheaf self-attention (Prop. 4) |

# B Sheaves and Copresheaves on Graphs: A Category Theoretical Look

This appendix provides a category-theoretic exposition of sheaves and copresheaves, emphasizing their definitions within the language of category theory. Additionally, we illustrate the construction of copresheaf neighborhood matrices through explicit combinatorial examples, instantiating the concepts developed in the main text.

## B.1 Copresheaves

Before diving into the technical definition, it is helpful to think of a *copresheaf* as a way of assigning data that flows along the structure of a graph, like signals along neurons, or resources in a network. In categorical terms, this structure formalizes the idea of consistently associating elements of some category $\mathcal{C}$.

**Definition 12** (Copresheaf on a directed graph). Let $G = (V, E)$ be a directed graph, and let $\mathcal{C}$ be a category. A *copresheaf* on $G$ is a functor $\mathcal{F} : G \to \mathcal{C}$, where the graph $G$ is regarded as a category whose objects are the vertices $V$, and whose morphisms are the directed edges $(x \to y) \in E$. When $\mathcal{C} = \mathbf{Vect}_{\mathbb{R}}$, this structure corresponds to a *quiver representation*.

Table 6: List of acronyms used throughout the paper.

| Acronym | Expansion |
| --- | --- |
| CC | Combinatorial Complex |
| CTNN | Copresheaf Topological Neural Network |
| CMPNN | Copresheaf Message-Passing Neural Network |
| SNN | Sheaf Neural Network |
| GNN | Graph Neural Network |
| CNN | Convolutional Neural Network |
| CT | Copresheaf Transformer |
| CGNN | Copresheaf Graph Neural Network |
| GCN | Graph Convolutional Network |
| GraphSAGE | Graph Sample and Aggregate |
| GIN | Graph Isomorphism Network |
| CopresheafGCN | Copresheaf Graph Convolutional Network |
| CopresheafSage | Copresheaf Graph Sample and Aggregate |
| CopresheafGIN | Copresheaf Graph Isomorphism Network |
| NSD | Neural Sheaf Diffusion |
| SAN | Sheaf Attention Network |
| MLP | Multi-Layer Perceptron |
| GAT | Graph Attention Network |
| CAM | Copresheaf Adjacency Matrix |
| CIM | Copresheaf Incidence Matrix |
| CNM | Copresheaf Neighborhood Matrix |

## B.2 Cellular Sheaves

**Definition 13** (Cellular sheaf on an undirected graph). Let $G = (V, E)$ be an undirected graph. Define the *incidence poset* $(P, \leq)$, where $P = V \cup E$, and the order relation is given by $x \leq e$ whenever vertex $x \in V$ is incident to edge $e \in E$. A *cellular sheaf* on $G$ with values in a category $\mathcal{C}$ is a functor $\mathcal{F} : P \to \mathcal{C}$, which assigns:

- to each vertex $x \in V$, an object $\mathcal{F}(x) \in \mathcal{C}$;

- to each edge $e \in E$, an object $\mathcal{F}(e) \in \mathcal{C}$;

- to each incidence relation $x \leq e$, a morphism $\mathcal{F}_{x \leq e} : \mathcal{F}(x) \to \mathcal{F}(e)$, called a *restriction map*,

such that the functoriality condition is satisfied on composable chains in the poset.

## B.3 Comparison Between Copresheaves and Cellular Sheaves

Copresheaves provide a versatile and powerful framework for machine learning applications across diverse domains, particularly excelling in scenarios where directional data flow and hierarchical dependencies are paramount. Unlike cellular sheaves, which are defined over undirected graphs and enforce consistency through restriction maps $\mathcal{F}_{x \leq e} : \mathcal{F}(x) \to \mathcal{F}(e)$, copresheaves operate on directed graphs, assigning learnable linear maps $\rho_{x \to y} : \mathcal{F}(x) \to \mathcal{F}(y)$ along edges. This enables anisotropic information propagation, making them ideal for tasks such as physical simulations, where data flows asymmetrically, as in fluid dynamics or heat transfer, or natural language processing, where sequential word dependencies dominate. More importantly, the vertex-centric design, assigning vector spaces $\mathcal{F}(x)$ solely to vertices, aligns seamlessly with message-passing architectures like Graph Neural Networks (GNNs) and Topological Neural Networks (TNNs), allowing these maps to be parameterized and optimized during training. Empirical evidence from our experiments highlights that copresheaf-based models outperform traditional architectures in capturing complex dynamics, demonstrating their superior ability to model spatially varying patterns and long-range dependencies in general applications. See Table 7 for a summary of the comparison between copresheaves and cellular sheaves.

Furthermore, copresheaves enhance machine learning models with a principled approach to regularization and expressiveness, broadening their suitability across heterogeneous domains. Standard neural network regularizers, such as $\ell_2$ decay or dropout, can be readily applied to copresheaf maps,

with optional structural losses like path-consistency ensuring morphism compositionality. This adaptability stands in stark contrast to the rigid cohomological constraints of cellular sheaves, which enforce local-to-global agreement through terms like $\|\mathcal{F}_{x \leq e}(\mathbf{h}_x) - \mathbf{h}_e\|^2$, limiting their flexibility in domains with asymmetric relationships. Copresheaves, by learning edge-wise maps, offer greater expressiveness for tasks involving non-Euclidean or multi-scale data, as evidenced by the superior performance of CopresheafConv layers on grid-based tasks. This makes them particularly effective for applications such as image segmentation, 3D mesh processing, or token-relation learning, where traditional methods like Convolutional Neural Networks (CNNs) struggle with directional or hierarchical structures. Our experiments further corroborate that copresheaf-augmented models consistently improve accuracy and detail recovery across diverse tasks, positioning them as a more suitable and generalizable tool for machine learning applications spanning Euclidean and non-Euclidean domains alike.

Table 7: Comparison between copresheaves and cellular sheaves.

| Aspect | Copresheaf | Cellular sheaf [Hansen and Ghrist, 2019b] |
|---|---|---|
| Graph type | Directed graph $G = (V, E)$ | Undirected graph $G = (V, E)$ |
| Assigned to vertices | Vector space $\mathcal{F}(x)$ for each $x \in V$ | Vector space $\mathcal{F}(x)$ for each $x \in V$ |
| Assigned to edges | Linear map $\rho_{x \to y} : \mathcal{F}(x) \to \mathcal{F}(y)$ for each directed edge $x \to y \in E$ | Vector space $\mathcal{F}(e)$ for each edge $e \in E$ |
| Associated maps | Pushforward: moves data forward along edges | Restriction: pulls data back from vertices to edges |
| Map direction | $\mathcal{F}(x) \to \mathcal{F}(y)$ (source to target) | $\mathcal{F}(x) \to \mathcal{F}(e)$ (vertex to incident edge) |
| Interpretation | Nodes have local features; edges transform and transmit them | Edges represent shared contexts; vertex features are restricted into them |
| Goal / Objective | Learn and compose edge-wise feature-space maps | Enforce coherence across by gluing local data |
| Typical Regularization | Standard NN regularizers ($\ell_2$ decay, spectral-norm, dropout, norm); optional structure losses (path-consistency, holonomy) | Agreement between restricted vertex features and the edge-stalk, e.g., $\|\mathcal{F}_{x \leq e}(\mathbf{h}_x) - \mathbf{h}_e\|^2$ |
| Use in learning | Embedding-level message passing, directional influence, anisotropic information flow | Compatibility across shared structures, enforcing local consistency, cohomological constraints |

## B.4 Copresheaf Neighborhood Matrix Example

**Example 1. Setup**. Let the combinatorial complex $\mathcal{X} = (\mathcal{S}, \mathcal{X}, \mathrm{rk})$ have symbols $\mathcal{S} = \{a, b, c, d\}$ and cells

$$\mathcal{X}^0 = \{\{a\}, \{b\}, \{c\}, \{d\}\},$$
$$\mathcal{X}^1 = \{\{a, b\}, \{b, c\}, \{c, a\}, \{d, b\}, \{c, d\}\},$$
$$\mathcal{X}^2 = \{\{a, b, c\}, \{d, b, c\}\}.$$

Ranks satisfy $\mathrm{rk}(\mathcal{X}^0) = 0$, $\mathrm{rk}(\mathcal{X}^1) = 1$, $\mathrm{rk}(\mathcal{X}^2) = 2$. Geometrically, this is the union of two triangles $(a, b, c)$ and $(d, b, c)$ sharing the edge $\{b, c\}$.

Let $\mathcal{E} = [e_1, e_2, e_3, e_4, e_5]$ with

$$e_1 = \{a, b\}, \qquad e_2 = \{b, c\}, \qquad e_3 = \{c, a\}, \qquad e_4 = \{d, b\}, \qquad e_5 = \{c, d\}.$$

**Edge–via–face neighborhood**. Define $\mathcal{N}_\triangle : \mathcal{X}^1 \to \mathcal{P}(\mathcal{X}^1)$ by

$$\mathcal{N}_\triangle(e) = \left\{ e' \in \mathcal{X}^1 : \exists f \in \mathcal{X}^2 \text{ with } e \subset f, \ e' \subset f, \ e' \neq e \right\}.$$

Thus, two edges are neighbors iff they both bound the same 2-cell. Concretely,

$$\mathcal{N}_\triangle(e_1) = \{e_2, e_3\}, \quad \mathcal{N}_\triangle(e_2) = \{e_1, e_3, e_4, e_5\},$$
$$\mathcal{N}_\triangle(e_3) = \{e_1, e_2\}, \quad \mathcal{N}_\triangle(e_4) = \{e_2, e_5\}, \quad \mathcal{N}_\triangle(e_5) = \{e_2, e_4\}.$$

The effective support is $\mathcal{X}_\mathcal{N} = \mathcal{X}^1$ (edges only).

**Induced directed graph**. The induced directed graph $G_{\mathcal{N}_\triangle} = (V_\mathcal{N}, E_\mathcal{N})$ has vertices $V_\mathcal{N} = \mathcal{X}^1$ and directed edges

$$E_\mathcal{N} = \{ e' \to e \mid e \in \mathcal{X}^1, \ e' \in \mathcal{N}_\triangle(e) \}.$$

Messages flow from an edge $e'$ to an adjacent edge $e$ whenever both bound a common triangle.

**Copresheaf on the edge–adjacency poset**. Assign to every edge a feature space $\mathcal{F}(e) = \mathbb{R}^2$. For each directed adjacency $e' \to e$, attach a linear map $\rho_{e' \to e} : \mathbb{R}^2 \to \mathbb{R}^2$. To keep notation compact,

we write diagonal maps as $\mathrm{diag}(\alpha, \beta)$ and $\mathrm{Id}_2$ for the $2 \times 2$ identity:

$$\rho_{e_2 \to e_1} = \mathrm{diag}(1, 0.8), \quad \rho_{e_3 \to e_1} = \mathrm{diag}(1, 0.6);$$
$$\rho_{e_1 \to e_2} = \mathrm{diag}(1, 0.8), \quad \rho_{e_3 \to e_2} = \mathrm{diag}(1, 0.8), \quad \rho_{e_4 \to e_2} = \mathrm{Id}_2, \quad \rho_{e_5 \to e_2} = \mathrm{diag}(1, 0.6);$$
$$\rho_{e_1 \to e_3} = \mathrm{diag}(1, 0.6), \quad \rho_{e_2 \to e_3} = \mathrm{diag}(1, 0.8);$$
$$\rho_{e_2 \to e_4} = \mathrm{Id}_2, \quad \rho_{e_5 \to e_4} = \mathrm{diag}(1, 0.7);$$
$$\rho_{e_2 \to e_5} = \mathrm{diag}(1, 0.6), \quad \rho_{e_4 \to e_5} = \mathrm{diag}(1, 0.7).$$

(Any consistent choice works; the point is a map per directed adjacency $e' \to e$.)

**Copresheaf Neighborhood Matrix (CNM).** For the ordering $\mathcal{E} = [e_1, e_2, e_3, e_4, e_5]$, the CNM $\mathbf{G}^{\mathcal{N}_\triangle} \in \left(\mathbb{R}^{2 \times 2}\right)^{5 \times 5}$ has block entries

$$\left[\mathbf{G}^{\mathcal{N}_\triangle}\right]_{i,j} = \begin{cases} \rho_{e_j \to e_i}, & \text{if } e_j \in \mathcal{N}_\triangle(e_i), \\ \mathbf{0}_{2 \times 2}, & \text{otherwise.} \end{cases}$$

Displayed explicity as:

$$\mathbf{G}^{\mathcal{N}_\triangle} = \begin{pmatrix} \mathbf{0}_{2\times2} & \rho_{e_2 \to e_1} & \rho_{e_3 \to e_1} & \mathbf{0}_{2\times2} & \mathbf{0}_{2\times2} \\ \rho_{e_1 \to e_2} & \mathbf{0}_{2\times2} & \rho_{e_3 \to e_2} & \rho_{e_4 \to e_2} & \rho_{e_5 \to e_2} \\ \rho_{e_1 \to e_3} & \rho_{e_2 \to e_3} & \mathbf{0}_{2\times2} & \mathbf{0}_{2\times2} & \mathbf{0}_{2\times2} \\ \mathbf{0}_{2\times2} & \rho_{e_2 \to e_4} & \mathbf{0}_{2\times2} & \mathbf{0}_{2\times2} & \rho_{e_5 \to e_4} \\ \mathbf{0}_{2\times2} & \rho_{e_2 \to e_5} & \mathbf{0}_{2\times2} & \rho_{e_4 \to e_5} & \mathbf{0}_{2\times2} \end{pmatrix}$$

See Figure 3.

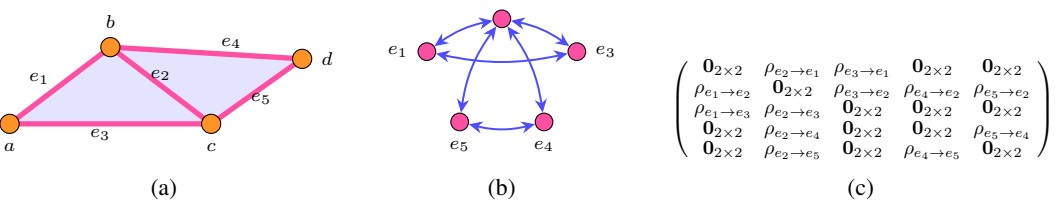

Figure 3: (a) A combinatorial complex $\mathcal{X} = (\mathcal{S}, \mathcal{X}, \mathrm{rk})$ with $\mathcal{S} = \{a, b, c, d\}$, edges $\mathcal{X}^1 = \{\{a, b\}, \{b, c\}, \{c, a\}, \{d, b\}, \{c, d\}\}$, and faces $\mathcal{X}^2 = \{\{a, b, c\}, \{d, b, c\}\}$. (b) Induced edge–adjacency digraph: nodes represent the edges of $\mathcal{X}$ and the edges represent the face adjacencies. (c) Copresheaf neighborhood matrix $\mathbf{G}^{\mathcal{N}_\triangle}$.

This CNM performs *edge-to-edge* directional message passing along face-adjacency: each edge $e_i$ aggregates transformed features from its face-adjacent neighbors $e_j$ via the maps $\rho_{e_j \to e_i}$. The shared edge naturally becomes a high-degree conduit between the two triangles.

**Example 2.** We define a copresheaf neighborhood matrix (CNM) for a combinatorial complex with an incidence neighborhood, guiding the reader through the setup, neighborhood, graph, copresheaf, and matrix.

**Setup.** Consider a combinatorial complex $\mathcal{X} = (\mathcal{S}, \mathcal{X}, \mathrm{rk})$ with $\mathcal{S} = \{a, b, c\}$, cells $\mathcal{X} = \{\{a\}, \{b\}, \{c\}, \{a, b\}, \{b, c\}\}$, and ranks $\mathrm{rk}(\{a\}) = \mathrm{rk}(\{b\}) = \mathrm{rk}(\{c\}) = 0$, $\mathrm{rk}(\{a, b\}) = \mathrm{rk}(\{b, c\}) = 1$. Thus, $\mathcal{X}^0 = \{\{a\}, \{b\}, \{c\}\}$, $\mathcal{X}^1 = \{\{a, b\}, \{b, c\}\}$. See Figure 4.

**Incidence neighborhood.** The incidence neighborhood $\mathcal{N}_{\mathrm{inc}} : \mathcal{X} \to \mathcal{P}(\mathcal{X})$ is:

$$\mathcal{N}_{\mathrm{inc}}(x) = \{y \in \mathcal{X} \mid x \subset y\}.$$

For 0-cells: $\mathcal{N}_{\mathrm{inc}}(\{a\}) = \{\{a, b\}\}$, $\mathcal{N}_{\mathrm{inc}}(\{b\}) = \{\{a, b\}, \{b, c\}\}$, $\mathcal{N}_{\mathrm{inc}}(\{c\}) = \{\{b, c\}\}$. For 1-cells: $\mathcal{N}_{\mathrm{inc}}(\{a, b\}) = \mathcal{N}_{\mathrm{inc}}(\{b, c\}) = \emptyset$. The effective support is $\mathcal{X}_\mathcal{N} = \{\{a\}, \{b\}, \{c\}\}$.

**Induced graph.** We induce a directed graph $G_\mathcal{N} = (V_\mathcal{N}, E_\mathcal{N})$, with:

$$V_\mathcal{N} = \mathcal{X}_\mathcal{N} \cup \bigcup_{x \in \mathcal{X}} \mathcal{N}_{\mathrm{inc}}(x) = \{\{a\}, \{b\}, \{c\}, \{a, b\}, \{b, c\}\} = \mathcal{X},$$

$$E_\mathcal{N} = \{y \to x \mid x \in \mathcal{X}_\mathcal{N}, y \in \mathcal{N}_{\mathrm{inc}}(x)\}$$
$$= \{\{a, b\} \to \{a\}, \{a, b\} \to \{b\}, \{b, c\} \to \{b\}, \{b, c\} \to \{c\}\}.$$

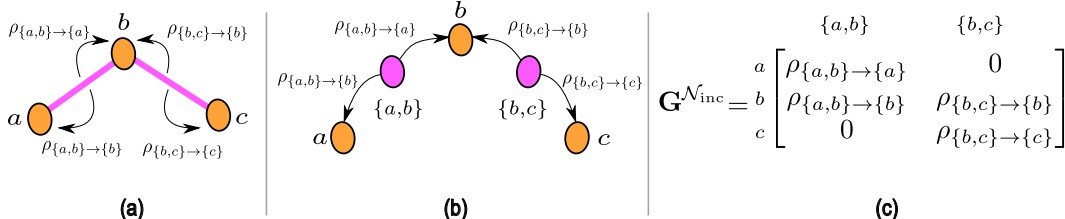

$$\mathbf{G}^{\mathcal{N}_{\text{inc}}} = \begin{array}{c} \phantom{a} \\ a \\ b \\ c \end{array}\begin{bmatrix} \rho_{\{a,b\}\to\{a\}} & 0 \\ \rho_{\{a,b\}\to\{b\}} & \rho_{\{b,c\}\to\{b\}} \\ 0 & \rho_{\{b,c\}\to\{c\}} \end{bmatrix}$$

**(a)**      **(b)**      **(c)**

Figure 4: (a) A combinatorial complex $\mathcal{X} = (\mathcal{S}, \mathcal{X}, \text{rk})$ with $\mathcal{S} = \{a, b, c\}$, cells $\mathcal{X} = \{\{a\}, \{b\}, \{c\}, \{a, b\}, \{b, c\}\}$. The figure also indicates the induced directed graph $G_{\mathcal{N}_{\text{inc}}} = (\mathcal{V}_\mathcal{N}, E_\mathcal{N})$ from the incidence neighborhood structure on the combinatorial complex $\mathcal{X}$. Each arrow $z \to y$ represents a directed edge from a 1-cell to a 0-cell where $y \subset z$, and is associated with a linear map $\rho_{z \to y}$ as part of the copresheaf. (b) The induced directed graph $G_{\mathcal{N}_{\text{inc}}} = (\mathcal{V}_\mathcal{N}, E_\mathcal{N})$ from the incidence neighborhood structure $\mathcal{N}_{\text{inc}}$. (c) The copresheaf neighborhood matrix (CNM) $\mathbf{G}^{\mathcal{N}_{\text{inc}}}$, where rows are indexed by 0-cells $\{a\}, \{b\}, \{c\}$ and columns by 1-cells $\{a, b\}, \{b, c\}$. The matrix entries are linear maps $\rho_{z \to y}$ when $z \in \mathcal{N}_{\text{inc}}(y)$, and 0 otherwise. This matrix supports directional feature propagation from 1-cells to 0-cells.

**Copresheaf**. The $\mathcal{N}_{\text{inc}}$-dependent copresheaf assigns $\mathcal{F}(x) = \mathbb{R}^2$ to each $x \in V_\mathcal{N}$, and linear maps $\rho_{y \to x} : \mathbb{R}^2 \to \mathbb{R}^2$ for $y \to x \in E_\mathcal{N}$:

$$\rho_{\{a,b\}\to\{a\}} = \begin{bmatrix} 1 & 0 \\ 0 & 0.5 \end{bmatrix}, \quad \rho_{\{a,b\}\to\{b\}} = \rho_{\{b,c\}\to\{b\}} = \begin{bmatrix} 1 & 0 \\ 0 & 1 \end{bmatrix}, \quad \rho_{\{b,c\}\to\{c\}} = \begin{bmatrix} 1 & 0 \\ 0 & 0.75 \end{bmatrix}.$$

**Neighborhood matrix**. The CNM $\mathbf{G}^\mathcal{N}$ for $\mathcal{Y} = \mathcal{X}^0$, $\mathcal{Z} = \mathcal{X}^1$ is:

$$[\mathbf{G}^\mathcal{N}]_{i,j} = \begin{cases} \rho_{z_j \to y_i} & \text{if } z_j \in \mathcal{N}_{\text{inc}}(y_i), \\ 0 & \text{otherwise}. \end{cases}$$

For $\mathcal{Y} = \{\{a\}, \{b\}, \{c\}\}$, $\mathcal{Z} = \{\{a, b\}, \{b, c\}\}$:

$$\mathbf{G}^\mathcal{N} = \begin{bmatrix} \rho_{\{a,b\}\to\{a\}} & 0 \\ \rho_{\{a,b\}\to\{b\}} & \rho_{\{b,c\}\to\{b\}} \\ 0 & \rho_{\{b,c\}\to\{c\}} \end{bmatrix}.$$

This matrix facilitates message passing from 1-cells to 0-cells, e.g., bond-to-atom feature propagation.

More generally, Figure 5 shows an illustrative example of the general setup of copresheaf higher-order message passing on a CC with multiple neighborhood functions.

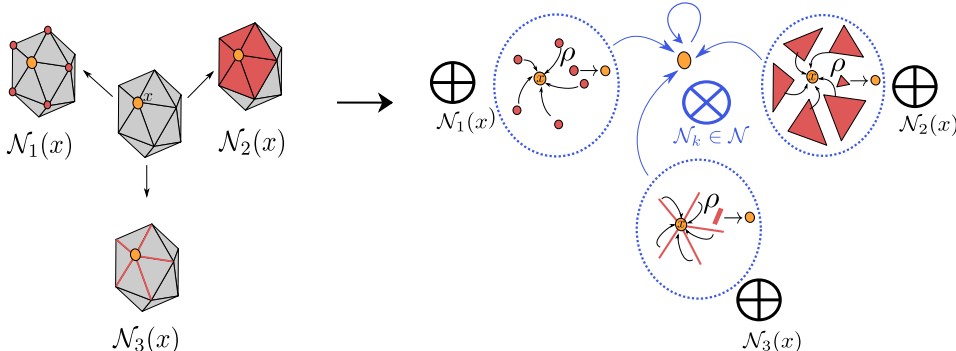

Figure 5: Illustration of copresheaf higher-order message passing. Left-hand side: Shows a central cell $x$ (as a circle) with arrows pointing to boxes labeled $\mathcal{N}_1(x), \mathcal{N}_2(x), \ldots, \mathcal{N}_k(x)$, representing the collection of neighborhood functions $\mathfrak{N} = \{\mathcal{N}_k\}_{k=1}^n$. Right-hand side: Depicts the message-passing process for the same cell $x$. Each neighborhood $\mathcal{N}_k(x)$ produces an aggregated message using $\bigoplus_{y \in \mathcal{N}_k(x)} \alpha_{\mathcal{N}_k}(\rho_{y \to x}(h_y^{(\ell)}))$. These messages are then combined using the inter-neighborhood function $\beta$, shown as a box, with an arrow updating $x$.

# C  Copresheaf Laplacian, Energy, and CTNN Transport–diffusion

In this section we introduce a linear *transport-discrepancy* operator $B_\rho$ that maps node fields to edge discrepancies, define the quadratic energy $E_\rho$ and the associated Laplacian $L_\rho = B_\rho^\top W B_\rho$, derive its block form and kernel, and show that a CTNN residual layer implements an explicit gradient step that monotonically decreases $E_\rho$ for suitable step sizes. This energy perspective provides an interpretable and theoretical foundation for the CTNN architecture, framing it as a diffusion process that minimizes a specific energy function. Our copresheaf Laplacian also coincides with the quiver Laplacian [Sumray et al., 2024] when viewing the transport maps as arrow representations, connecting our framework to established spectral methods while enabling directional, anisotropic information flow distinct from the symmetric diffusion in sheaf-based models Bodnar et al. [2022].

**Spaces and operator**. Let $(F, \rho, G)$ be a copresheaf defined on a directed graph $G = (V, E)$. To each node $x \in V$ attach a finite-dimensional real inner-product space $F(x)$. For every edge $y \to x \in E$ fix a linear *transport* $\rho_{y \to x} : F(y) \to F(x)$. Define the node and edge product spaces

$$\mathcal{H}_V := \bigoplus_{x \in V} F(x), \qquad \mathcal{H}_E := \bigoplus_{(y \to x) \in E} F(x),$$

equipped with canonical (blockwise) inner products $\langle \mathbf{h}, \tilde{\mathbf{h}} \rangle_V = \sum_{x \in V} \langle \mathbf{h}_x, \tilde{\mathbf{h}}_x \rangle$ and $\langle \boldsymbol{\xi}, \tilde{\boldsymbol{\xi}} \rangle_E = \sum_{(y \to x) \in E} \langle \boldsymbol{\xi}_{y \to x}, \tilde{\boldsymbol{\xi}}_{y \to x} \rangle$.

**Definition 14** (Transport-discrepancy operator). The linear operator

$$B_\rho : \mathcal{H}_V \longrightarrow \mathcal{H}_E$$

is defined componentwise by

$$(B_\rho \mathbf{h})_{(y \to x)} = \mathbf{h}_x - \rho_{y \to x} \mathbf{h}_y \in F(x), \qquad \forall (y \to x) \in E. \tag{8}$$

Let $B_\rho^\top : \mathcal{H}_E \to \mathcal{H}_V$ denote the Euclidean adjoint. A direct calculation yields

$$(B_\rho^\top \boldsymbol{\xi})_x = \sum_{y \to x} \boldsymbol{\xi}_{y \to x} - \sum_{x \to z} \rho_{x \to z}^\top \boldsymbol{\xi}_{x \to z}, \qquad \forall x \in V, \, \boldsymbol{\xi} \in \mathcal{H}_E. \tag{9}$$

**Definition 15** (**Energy and copresheaf Laplacian**). Let $w_{y \to x} > 0$ be edge weights and define the diagonal operator $W : \mathcal{H}_E \to \mathcal{H}_E$ by $(W\boldsymbol{\xi})_{y \to x} = w_{y \to x} \boldsymbol{\xi}_{y \to x}$. The *weighted transport energy and the copresheaf Laplacian* are

$$E_\rho(\mathbf{h}) = \|B_\rho \mathbf{h}\|_W^2 = \sum_{(y \to x) \in E} w_{y \to x} \|\mathbf{h}_x - \rho_{y \to x} \mathbf{h}_y\|^2, \qquad L_\rho = B_\rho^\top W B_\rho. \tag{10}$$

**Remark 2** (Relation to quiver Laplacian). Interpreting the transports $\rho_{y \to x}$ as arrow maps of a quiver representation, the copresheaf Laplacian $L_\rho$ in Equation 10 coincides with the standard (weighted) quiver Laplacian $B^\top W B$ of that representation. See Sumray et al. [2024] for more about the quiver Laplacian.

It is sometimes useful to have the copresheaf Laplacian in its exact nodewise form. Namely, for every outgoing edge $x \to z$ with map $\rho_{x \to z}$ and weight $w_{x \to z}$, include the reverse edge $z \to x$ with $\rho_{z \to x} := \rho_{x \to z}^\top$ and $w_{z \to x} := w_{x \to z}$. With this convention one has the nodewise form of the copresheaf Laplacian:

$$(L_\rho \mathbf{h})_x = \sum_{y \in N(x)} w_{y \to x} (\mathbf{h}_x - \rho_{y \to x} \mathbf{h}_y). \tag{11}$$

**Theorem 2.** With $w_{y \to x} > 0$, $L_\rho = B_\rho^\top W B_\rho : \mathcal{H}_V \to \mathcal{H}_V$ is symmetric positive semidefinite and $\mathbf{h}^\top L_\rho \mathbf{h} = \|B_\rho \mathbf{h}\|_W^2 \geq 0$. Its block action at node $x$ is

$$(L_\rho \mathbf{h})_x = \sum_{y \to x} w_{y \to x} (\mathbf{h}_x - \rho_{y \to x} \mathbf{h}_y) + \sum_{x \to z} w_{x \to z} \rho_{x \to z}^\top (\rho_{x \to z} \mathbf{h}_x - \mathbf{h}_z).$$

Moreover,

$$\ker L_\rho = \ker B_\rho = \{\mathbf{h} \in \mathcal{H}_V : \mathbf{h}_x = \rho_{y \to x} \mathbf{h}_y \text{ for all } y \to x \in E\}.$$

*Proof.* P.s.d. and the quadratic identity follow from $L_\rho = B_\rho^\top W B_\rho$ with $W \succ 0$. The block formula follows by expanding $B_\rho^\top (W B_\rho \mathbf{h})$ via 10. Finally, $\|B_\rho \mathbf{h}\|_W^2 = 0 \iff B_\rho \mathbf{h} = 0$. $\qquad \square$

## C.1 CTNN/CMPNN residual as copresheaf diffusion.

From the copresheaf message passing equation in Definition 9, choose the edge message $\alpha(\mathbf{h}_x, \rho_{y\to x}\mathbf{h}_y) = \mathbf{h}_x - \rho_{y\to x}\mathbf{h}_y$, sum aggregation, and residual update $\beta(\mathbf{h}_x, m) = \mathbf{h}_x - \eta m$. Then a CTNN/CMPNN layer updates

$$\mathbf{h}_x^{(\ell+1)} = \mathbf{h}_x^{(\ell)} - \eta \sum_{y\in N(x)} w_{y\to x}\left(\mathbf{h}_x^{(\ell)} - \rho_{y\to x}\mathbf{h}_y^{(\ell)}\right) = \left(I - \eta L_\rho\right)\mathbf{h}^{(\ell)}. \tag{14}$$

We have the following theorem.

**Theorem 3.** For $E_\rho$ in (13),

$$\nabla E_\rho(\mathbf{h}) = 2\,L_\rho\mathbf{h}, \qquad \mathbf{h}^{(\ell+1)} = \mathbf{h}^{(\ell)} - \eta\,\nabla\!\left(\tfrac{1}{2}E_\rho\right)\!(\mathbf{h}^{(\ell)}).$$

Hence (14) is an explicit gradient step on $\frac{1}{2}E_\rho$. If $0 < \eta < \|L_\rho\|_2^{-1}$, then $E_\rho(\mathbf{h}^{(\ell+1)}) \le E_\rho(\mathbf{h}^{(\ell)})$, with strict inequality whenever $\mathbf{h}^{(\ell)} \notin \ker L_\rho$. Moreover, $I - \eta L_\rho$ is non-expansive in $\|\cdot\|_2$ and strictly contractive on $\ker L_\rho^\perp$, so $\mathbf{h}^{(\ell)}$ converges to the orthogonal projection of $\mathbf{h}^{(0)}$ onto $\ker L_\rho$.

*Proof.* Since $E_\rho(\mathbf{h}) = (B_\rho\mathbf{h})^\top W(B_\rho\mathbf{h})$, the chain rule gives $\nabla E_\rho(\mathbf{h}) = 2B_\rho^\top W B_\rho\mathbf{h} = 2L_\rho\mathbf{h}$, so (14) is gradient descent on $\frac{1}{2}E_\rho$. Let $L_\rho = U\Lambda U^\top$ with $\Lambda = \mathrm{diag}(\lambda_i \ge 0)$ and write $\mathbf{h}^{(\ell)} = U\mathbf{c}^{(\ell)}$. Then $c_i^{(\ell+1)} = (1 - \eta\lambda_i)c_i^{(\ell)}$ and $E_\rho(\mathbf{h}^{(\ell)}) = \sum_i \lambda_i(c_i^{(\ell)})^2$. If $0 \le \eta\lambda_i \le 1$ for all $i$, then $\lambda_i(1 - \eta\lambda_i)^2 \le \lambda_i$, giving monotone decay, strict if some $\lambda_i > 0$ has $c_i^{(\ell)} \ne 0$. Non-expansiveness and convergence follow from $|1 - \eta\lambda_i| \le 1$ (and $< 1$ for $\lambda_i > 0$). $\qquad\square$

**Remark 3.** $\rho_{y\to x}$ may be (i) direct per-edge linear maps, (ii) factored as $\rho_{y\to x} = F_x^\top F_y$ to reduce parameters, or (iii) constrained (e.g. softly orthogonal via $\|\rho_{y\to x}^\top\rho_{y\to x} - I\|_F^2$) to regularize the spectrum of $L_\rho$. A learnable (possibly per-layer/head) stepsize $\eta$, normalized by a running estimate of $\|L_\rho\|_2^{-1}$, enforces the energy-decay guarantee in Theorem 3. When $\rho_{y\to x} = I$, (13) reduces to the (vector-valued) graph Laplacian; orthogonal $\rho$ recovers a connection Laplacian.

## D Expressive Power of CTNNs

### D.1 Universal Approximation of $\mathcal{N}$-Dependent Copresheaves

Here, we demonstrate that multilayer perceptrons (MLPs) can approximate arbitrary copresheaf morphisms induced by neighborhood functions. This result ensures that the proposed sheaf-based model is sufficiently expressive to capture complex data interactions.

**Proposition 2** (Universal approximation of $\mathcal{N}$-dependent copresheaves). Let $\mathcal{X}$ be a finite combinatorial complex and $\mathcal{N}$ a neighborhood function on $\mathcal{X}$. Suppose $G_\mathcal{N} \longrightarrow \mathbf{Vect}_\mathbb{R}$ is an $\mathcal{N}$-dependent copresheaf with stalks $\mathcal{F}^\mathcal{N}(x) = \mathbb{R}^d$ and morphisms $\rho_{y\to x}^\mathcal{N}$. Let a feature map

$$h: \mathcal{X} \to \mathbb{R}^d, \quad x \mapsto \mathbf{h}_x$$

be given so that the $2d$-dimensional vectors $(\mathbf{h}_y, \mathbf{h}_x)$ are pairwise distinct for every directed edge $y \to x$. Define

$$A = \left\{(\mathbf{h}_y, \mathbf{h}_x) \,\middle|\, y \to x\right\} \subset \mathbb{R}^{2d}, \quad g: A \to \mathbb{R}^{d\times d}, \quad g(\mathbf{h}_y, \mathbf{h}_x) = \rho_{y\to x}^\mathcal{N}.$$

Then for any $\varepsilon > 0$ there exists a multilayer perceptron $\Phi: \mathbb{R}^{2d} \to \mathbb{R}^{d\times d}$ with sufficiently many hidden units such that $\left\|\Phi(\mathbf{h}_y, \mathbf{h}_x) - \rho_{y\to x}^\mathcal{N}\right\| < \varepsilon$ for all $y \to x$.

*Proof.* Since $A$ is finite and its elements are distinct, the assignment $g: A \to \mathbb{R}^{d\times d}$ is well-defined.

Enumerate $A = \{a_i\}_{i\in I}$, choose disjoint open neighborhoods $U_i \ni a_i$, and pick smooth "bump" functions

$$\varphi_i: \mathbb{R}^{2d} \to [0,1], \quad \varphi_i(a_i) = 1, \quad \mathrm{supp}(\varphi_i) \subset U_i.$$

Then the sum

$$f(a) = \sum_{i\in I} g(a_i)\,\varphi_i(a)$$

is a smooth map $f \colon \mathbb{R}^{2d} \to \mathbb{R}^{d \times d}$ satisfying $f|_A = g$.

Since $A$ is finite, choose a compact set $K \subseteq \mathbb{R}^{2d}$ containing $A$, ensuring the applicability of the Universal Approximation Theorem. By that theorem, for any $\varepsilon > 0$ there is an MLP $\Phi$ such that

$$\sup_{a \in K} \left\| \Phi(a) - f(a) \right\| < \varepsilon.$$

In particular, for each $a_i \in A$ we have

$$\left\| \Phi(a_i) - g(a_i) \right\| = \left\| \Phi(\mathbf{h}_y, \mathbf{h}_x) - \rho_{y \to x}^{\mathcal{N}} \right\| < \varepsilon,$$

for every $y \to x$. This completes the proof. $\qquad\square$

# E  Sheaf Neural Networks Are Copresheaf Message-Passing Neural Networks

The computational use of a cellular sheaf on a graph rests on the *incidence poset*. Let $G = (V, E)$ and consider the poset on $V \cup E$ with $x \preceq e$ whenever $x \in e$. A cellular sheaf $\mathcal{F}$ assigns a vector space to each cell and a linear structure map to each incidence. In our convention, the "restriction" along $x \preceq e$ is implemented as a vertex-to-edge lift $\mathcal{F}_{x \trianglelefteq e} : \mathcal{F}(x) \to \mathcal{F}(e)$. Equipping $\mathcal{F}(e)$ with an inner product yields the adjoint $\mathcal{F}_{x \trianglelefteq e}^\top : \mathcal{F}(e) \to \mathcal{F}(x)$, a canonical edge-to-vertex back-projection. Message passing between adjacent vertices then arises by composing incidence maps: for $e = \{x, y\}$, the message from $y$ to $x$ is

$$\mathcal{F}_{x \trianglelefteq e}^\top \circ \mathcal{F}_{y \trianglelefteq e} : \mathcal{F}(y) \to \mathcal{F}(x).$$

This composition induces a *direction of information flow* on an undirected graph while remaining faithful to the sheaf poset structure. Diffusion-style updates and sheaf-Laplacian operators are recovered by aggregating such edge-mediated messages over $\mathcal{N}(x)$. See Figure 6 for an illustration.

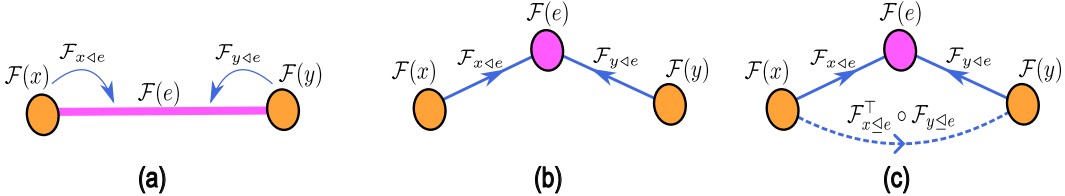

Figure 6: **Sheaf-induced message passing on an edge** $e = \{x, y\}$. *(a)* Local spaces $\mathcal{F}(x), \mathcal{F}(y), \mathcal{F}(e)$ with vertex-to-edge lifts $\mathcal{F}_{x \trianglelefteq e}$ and $\mathcal{F}_{y \trianglelefteq e}$ along the incidences $x \preceq e$, $y \preceq e$. *(b)* Incidence-poset view: arrows encode the sheaf's linear maps attached to $x \preceq e$ and $y \preceq e$. *(c)* Edge-mediated message passing: the inner product on $\mathcal{F}(e)$ yields adjoints $\mathcal{F}_{x \trianglelefteq e}^\top$. The message from $y$ to $x$ is the composition $\mathcal{F}_{x \trianglelefteq e}^\top \circ \mathcal{F}_{y \trianglelefteq e}$, i.e., a direct vertex-to-vertex morphism $\rho_{y \to x} : \mathcal{F}(y) \to \mathcal{F}(x)$, compatible with the bidirected expansion $G'$. This realizes SNN message passing as CMPNN morphisms assembled from sheaf structure maps.

In this appendix, we prove Theorem 1 and Proposition 1, which demonstrate that existing sheaf neural networks (SNNs), including sheaf diffusion networks, are special cases of the copresheaf message-passing neural network (CMPNN) framework (Definition 9). Moreover, we summarize how other sheaf-based neural architectures align with our unifying message-passing framework (see Table 8).

*Proof of Theorem 1.* We construct the bidirected graph $G'$ and define the copresheaf $\mathcal{G}$ on it, then demonstrate the equivalence of the message-passing operations.

First, construct the bidirected graph $G' = (V, E')$ from $G = (V, E)$ by replacing each undirected edge $\{x, y\} \in E$ with two directed edges $(x, y), (y, x) \in E'$. This ensures that $G'$ retains the connectivity of $G$ while introducing explicit directionality.

Next, define the copresheaf $\mathcal{G} : G' \to \mathbf{Vect}_{\mathbb{R}}$ as follows:

- For each vertex $x \in V$, assign $\mathcal{G}(x) = \mathcal{F}(x)$, where $\mathcal{F}(x)$ is the vector space associated with $x$ by the cellular sheaf $\mathcal{F}$.
- For each directed edge $(y, x) \in E'$, corresponding to the undirected edge $e = \{x, y\} \in E$, define the linear morphism $\rho_{y \to x} : \mathcal{G}(y) \to \mathcal{G}(x)$ by $\rho_{y \to x} = \mathcal{F}_{x \trianglelefteq e}^\top \circ \mathcal{F}_{y \trianglelefteq e}$, where $\mathcal{F}_{y \trianglelefteq e} : \mathcal{F}(y) \to \mathcal{F}(e)$

and $\mathcal{F}_{x\triangleleft e} : \mathcal{F}(x) \to \mathcal{F}(e)$ are the restriction maps of $\mathcal{F}$, and $\mathcal{F}_{x\triangleleft e}^\top : \mathcal{F}(e) \to \mathcal{F}(x)$ is the adjoint with respect to an inner product on $\mathcal{F}(e)$.

Now, consider the SNN message-passing mechanism along the edge $e = \{x, y\}$. For a feature vector $\mathbf{h}_y \in \mathcal{F}(y)$ at vertex $y$, the message transmitted to vertex $x$ is given by $\mathcal{F}_{x\triangleleft e}^\top \circ \mathcal{F}_{y\triangleleft e}(\mathbf{h}_y)$.

In the copresheaf $\mathcal{G}$ on $G'$, the morphism associated with the directed edge $(y, x)$ is $\rho_{y\to x} = \mathcal{F}_{x\triangleleft e}^\top \circ \mathcal{F}_{y\triangleleft e}$. Applying this morphism, the message-passing operation in the CMPNN yields $\rho_{y\to x}(\mathbf{h}_y) = \mathcal{F}_{x\triangleleft e}^\top \circ \mathcal{F}_{y\triangleleft e}(\mathbf{h}_y)$, which is identical to the SNN message.

To ensure that $\mathcal{G}$ is a well-defined copresheaf, observe that it assigns vector spaces to vertices and linear maps to directed edges in a functorial manner. Specifically, for each directed edge $(y, x) \in E'$, the map $\rho_{y\to x}$ is a composition of linear maps and thus linear. The identity and composition properties are satisfied implicitly through the consistency of the sheaf restriction maps.

Therefore, the SNN message passing, which operates via intermediate edge spaces in $\mathcal{F}$, is equivalently represented as direct vertex-to-vertex message passing in the copresheaf $\mathcal{G}$ on $G'$. This completes the proof. $\qquad\square$

*Proof of Proposition 1.* Compute:

$$(I_n \otimes W_1)\mathbf{H} = [W_1 \mathbf{h}_x]_{x \in V},$$

$$\big(\Delta_\mathcal{F} \otimes I\big)(I_n \otimes W_1)\mathbf{H}W_2 = \left[ \sum_{y \in V} L_{F,x,y}\, W_1 \mathbf{h}_y\, W_2 \right]_{x \in V},$$

$$\mathbf{h}_x^+ = \mathbf{h}_x - \sum_{y \in \mathcal{N}(x) \cup \{x\}} W_2\, L_{F,x,y}\, W_1 \mathbf{h}_y,$$

since $L_{F,x,y} = 0$ for $y \notin \mathcal{N}(x) \cup \{x\}$.

Interpret $G$ as a directed graph with edges $y \to x$ for $y \in \mathcal{N}(x)$ and $x \to x$. Define:

- Message function: $\alpha(\mathbf{h}_x, \rho_{y\to x}\mathbf{h}_y) = W_2\, L_{F,x,y}\, W_1 \mathbf{h}_y$,
- Morphisms: $\rho_{y\to x}$ implicitly encoded via $L_{F,x,y}$,
- Aggregator: $\oplus = \sum$,
- Update: $\beta(\mathbf{h}_x, m) = \mathbf{h}_x - m$.

Thus:

$$\mathbf{h}_x^+ = \beta\left( \mathbf{h}_x, \sum_{y \to x} \alpha(\mathbf{h}_x, \rho_{y\to x}\mathbf{h}_y) \right),$$

matching Definition 9. $\qquad\square$

Table 8 provides a summary of sheaf neural networks realized in terms of Definition 9.

We finally prove the following theorem to show the relationship more precisely between CTNNs, MPNNs and SNNs.

**Theorem 4** (CTNNs strictly subsume SNNs and contain MPNNs). Let $\mathcal{F}_{\mathrm{CTNN}}$, $\mathcal{F}_{\mathrm{SNN}}$, and $\mathcal{F}_{\mathrm{MPNN}}$ denote the function classes realized by Copresheaf Topological Neural Networks (CTNNs), Sheaf Neural Networks (SNNs), and Message-Passing Neural Networks (MPNNs), respectively. Then

$$\mathcal{F}_{\mathrm{SNN}} \subset \mathcal{F}_{\mathrm{CTNN}} \qquad \text{and} \qquad \mathcal{F}_{\mathrm{MPNN}} \subseteq \mathcal{F}_{\mathrm{CTNN}}.$$

*Proof.* (1) $\mathcal{F}_{\mathrm{SNN}} \subset \mathcal{F}_{\mathrm{CTNN}}$. First from Theorem 1 we know $\mathcal{F}_{\mathrm{SNN}} \subseteq \mathcal{F}_{\mathrm{CTNN}}$. To prove the strict containment, fix $\{u, v\} \in E$ and let $F(u) = F(v) = \mathbb{R}^2$. Define a CTNN on $G'$ by

$$\rho_{v\to u} = I_2, \qquad \rho_{u\to v} = 0.$$

No SNN can realize this, since SNN transports necessarily reciprocate across an undirected edge:

$$\rho_{v\to u} = F_{u\triangleleft e}^\top F_{v\triangleleft e} \implies \rho_{u\to v} = F_{v\triangleleft e}^\top F_{u\triangleleft e} = \rho_{v\to u}^\top.$$

Thus $\rho_{u\to v} = 0$ would force $\rho_{v\to u} = 0$, contradicting $\rho_{v\to u} = I_2$. Therefore $\mathcal{F}_{\mathrm{SNN}} \subset \mathcal{F}_{\mathrm{CTNN}}$ is strict.

Table 8: Unified message passing formulations of various sheaf neural networks using our copresheaf topological neural network (CTNN) notation given in Proposition 10. The restriction maps $\rho_{y \to x}$ may be linear, data-driven, or attentional depending on the model.

| Method (Paper) | Message Passing Equation | Notable Features | Restriction Map $\rho_{y \to x}$ |
|---|---|---|---|
| **Sheaf Neural Network (SNN)** Hansen & Gebhart (2020) | $\mathbf{h}_x^{(l+1)} = \sigma\left(\mathbf{h}_x^{(l)} + \sum_{y \in \mathcal{N}(x)} \rho_{y \to x}\mathbf{h}_y^{(l)}\right)$ | Linear restriction maps $\rho_{y \to x}$ assigned per edge; enables high-dimensional, direction-aware message passing via sheaf structure. | $\rho_{y \to x} = \mathcal{F}_{x \unlhd e}^\top \mathcal{F}_{y \unlhd e}$, fixed linear map, $e = \{x, y\}$ |
| **Neural Sheaf Diffusion (NSD)** Bodnar et al. (2022) | $\mathbf{h}_x^{(l+1)} = \mathbf{h}_x^{(l)} - \sigma\left(\sum_{x \unlhd e} \rho_{x \to x}\mathbf{h}_x^{(l)} - \sum_{y \in \mathcal{N}(x)} \rho_{y \to x}\mathbf{h}_y^{(l)}\right)$ | Diffusion over learned sheaf Laplacian; restriction maps $\rho_{y \to x}$ are learnable, reflecting edge-mediated interactions. | $\rho_{y \to x} = \mathcal{F}_{x \unlhd e}^\top \mathcal{F}_{y \unlhd e}$, learned linear map, $e = \{x, y\}$ |
| **Sheaf Attention Network (SAN)** Barbero et al. (2022) | $\mathbf{h}_x^{(l+1)} = \sigma\left(\sum_{y \in \mathcal{N}(x)} \alpha_{xy}(\mathbf{h}_x, \mathbf{h}_y)\, \rho_{y \to x}\mathbf{h}_y^{(l)}\right)$ | Attentional sheaf: attention weights $\alpha_{xy}$ modulate the restricted neighbor feature; mitigates oversmoothing in GAT-style setups. | $\rho_{y \to x}$: learned linear map, parameterized to capture feature space relationships |
| **Connection Laplacian SNN** Barbero et al. (2022) | $\mathbf{h}_x^{(l+1)} = \sigma\left(\mathbf{h}_x^{(l)} + \sum_{y \in \mathcal{N}(x)} O_{xy}\mathbf{h}_y^{(l)}\right)$ | Edge maps $O_{xy}$ are orthonormal, derived from feature space alignment; reduces learnable parameters and reflects local geometric priors. | $O_{xy}$: orthonormal matrix, learned to align feature spaces across edges |
| **Heterogeneous Sheaf Neural Network (HetSheaf)** Braithwaite et al. (2024) | $\mathbf{h}_x^{(l+1)} = \sigma\left(\mathbf{h}_x^{(l)} + \sum_{y \in \mathcal{N}(x)} \rho_{y \to x}(\mathbf{h}_x, \mathbf{h}_y)\mathbf{h}_y^{(l)}\right)$ | Type-aware sheaf morphisms: $\rho_{y \to x}$ depend on node and edge types, enabling structured heterogeneity across the graph. | $\rho_{y \to x}$: type-aware learned linear map, parameterized by node and edge types |
| **Adaptive Sheaf Diffusion** Zaghen et al. (2024) | $\mathbf{h}_x^{(l+1)} = \mathbf{h}_x^{(l)} + \sigma\left(\sum_{y \in \mathcal{N}(x)} \rho_{y \to x}(\mathbf{h}_x, \mathbf{h}_y)\,(\mathbf{h}_y^{(l)} - \mathbf{h}_x^{(l)})\right)$ | Nonlinear Laplacian-like dynamics with adaptive, feature-aware restriction maps $\rho_{y \to x}$; enhances expressiveness and locality. | $\rho_{y \to x}$: feature-aware learned linear map, parameterized by node features |

*(2)* $\mathcal{F}_{\mathrm{MPNN}} \subseteq \mathcal{F}_{\mathrm{CTNN}}$. Let an MPNN layer on a directed graph have the form

$$h_u^{(\ell+1)} = \phi_{\mathrm{upd}}^{(\ell)}\Big(h_u^{(\ell)},\ \square_{v \in \mathcal{N}(u)}\phi_{\mathrm{msg}}^{(\ell)}\big(h_v^{(\ell)}, h_u^{(\ell)}, e_{vu}\big)\Big),$$

for a permutation-invariant aggregator $\square$. The results follows immediately from Definition 1. $\square$

# F   A General Copresheaf-Based Transformer Layer

The main idea to introduce a copresheaf structure to the transformer is the following. For every ordered pair $y \to x$ (within an attention head) we define a parametrized copresheaf map

$$\rho_{y \to x}\ :\ \mathbb{R}^d \longrightarrow \mathbb{R}^d, \qquad v_y \longmapsto \rho_{y \to x}\, v_y,$$

which transports the value vector from the stalk at $y$ to the stalk at $x$. Given attention weights $\alpha_{xy} = \mathrm{softmax}_{y \in \mathcal{N}(x)}((q_x^\top k_y)/\sqrt{d})$, the head message is $m_x = \sum_{y \in \mathcal{N}(x)} \alpha_{xy}\, \rho_{y \to x}\, v_y$.

In Definition 11 we introduced the notion of copresheaf self-attention. A natural extension of this definition is the copresheaf cross-attention.

**Definition 16** (Copresheaf cross-attention). For source rank $k_s$ and target rank $k_t$, with neighborhood $\mathcal{N}_{s \to t}$, define learnable projection matrices $W_q^{s \to t} \in \mathbb{R}^{p \times d_t}, W_k^{s \to t} \in \mathbb{R}^{p \times d_s}, W_v^{s \to t} \in \mathbb{R}^{d_s \times d_t}$ where $d_s$ and $d_t$ are the feature dimensions of source and target cells, respectively. We then propose the **copresheaf cross-attention** as the aggregation and update $\mathbf{h}_x^{(\ell+1)} = \beta\big(\mathbf{h}_x^{(\ell)}, m_x\big)$ where $m_x = \sum_{y \in \mathcal{N}_{s \to t}(x)} a_{xy}\rho_{y \to x}(v_y)$ with $\rho_{y \to x} : \mathcal{F}(y) \to \mathcal{F}(x)$ being a learned map and

$$a_{xy} = \frac{\exp(\langle q_x, k_y \rangle / \sqrt{p})}{\sum_{y' \in \mathcal{N}_{s \to t}(x)} \exp(\langle q_x, k_{y'} \rangle / \sqrt{p})}, \tag{12}$$

where $k_y = W_k^{s \to t}h_y^{(\ell)}, v_y = W_v^{s \to t}h_y^{(\ell)}$ and for each target cell $x \in \mathcal{X}^{k_t}, q_x = W_q^{s \to t}h_x^{(\ell)}$.

Figure 7 illustrates the cross attention in the copresheaf transformer.

In Algorithm 1, we provide the pseudocode for our generic copresheaf-based transformer layer. This algorithm outlines the layer-wise update rule combining self-attention within cells of equal rank and cross-attention between different ranks, using learned copresheaf morphisms to transfer features between stalks. It generalizes standard transformer mechanisms by introducing neighborhood-dependent transformations.

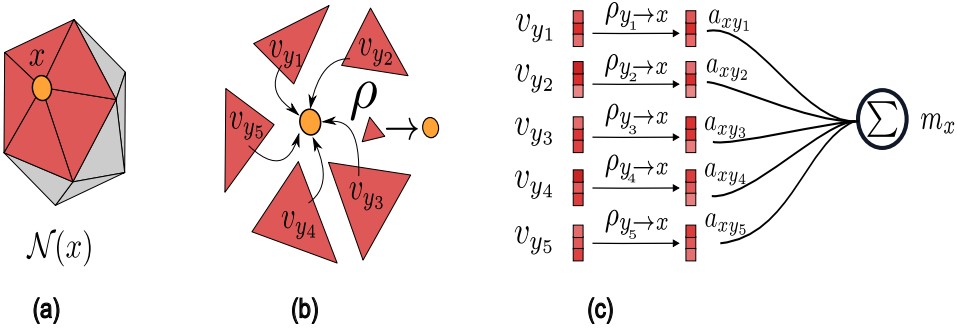

Figure 7: **Copresheaf cross-attention.** **(a)** A target cell $x$ (yellow) in a combinatorial complex with its neighborhood $\mathcal{N}(x)$ (red). Sources may be at a different rank than $x$ (e.g., faces $\to$ vertex). **(b)** Cross-attention schematic: each neighbor $y \in \mathcal{N}(x)$ contributes a value $v_y$ that is first transported into the target's local feature space via a learned map $\rho_{y\to x} : F(y) \to F(x)$; attention weights $a_{xy}$ are computed from $q_x$ and $k_y$. **(c)** Implementation view: for every $y$, apply $\rho_{y\to x}$ to $v_y$, scale by $a_{xy}$, and sum to form the message $m_x = \sum_{y\in\mathcal{N}(x)} a_{xy}\, \rho_{y\to x}(v_y)$, which updates $h_x$. Transporting values with $\rho$ enables directional, cross-rank, and anisotropic information flow beyond standard attention.

---

**General copresheaf-based transformer layer**

1: **procedure** COPRESHEAFTRANSFORMERLAYER($\mathcal{X}, \mathcal{N}, \{h_x^{(\ell)} \in \mathcal{F}(x)\}_{x\in\mathcal{X}}$)
2:     **for** $x \in \mathcal{X}^k$ **do**                                                      $\triangleright$ Self-attention on $k$-cells
3:          $q_x \leftarrow W_q h_x^{(\ell)},\ k_x \leftarrow W_k h_x^{(\ell)},\ v_x \leftarrow W_v h_x^{(\ell)}$
4:          $m_x \leftarrow 0$
5:          **for** $y \in \mathcal{N}_k(x)$ **do**
6:              $a_{xy} \leftarrow \mathrm{softmax}_{\mathcal{N}_k(x)}(\langle q_x, k_y\rangle/\sqrt{p})$
7:              $\widetilde{v}_{xy} \leftarrow \rho_{y\to x}(v_y)$
8:              $m_x \leftarrow m_x + a_{xy}\widetilde{v}_{xy}$
9:          **end for**
10:         $h_x^{(\ell+1)} \leftarrow \beta(h_x^{(\ell)}, m_x)$
11:     **end for**
12:     **for** $x \in \mathcal{X}'^{k_t}$ **do**                                          $\triangleright$ Cross-attention from $k_s$ to $k_t$
13:          $q_x \leftarrow W_q^{s\to t} h_x^{(\ell)}$
14:          $m_x \leftarrow 0$
15:          **for** $y \in \mathcal{N}_{s\to t}(x)$ **do**
16:              $k_y \leftarrow W_k^{s\to t} h_y^{(\ell)},\ v_y \leftarrow W_v^{s\to t} h_y^{(\ell)}$
17:              $a_{xy} \leftarrow \mathrm{softmax}_{\mathcal{N}_{s\to t}(x)}(\langle q_x, k_y\rangle/\sqrt{p})$
18:              $\widetilde{v}_{xy} \leftarrow \rho_{y\to x}(v_y)$
19:              $m_x \leftarrow m_x + a_{xy}\widetilde{v}_{xy}$
20:          **end for**
21:         $h_x^{(\ell+1)} \leftarrow \beta(h_x^{(\ell)}, m_x)$
22:     **end for**
23:     **return** $\{h_x^{(\ell+1)}\}_{x\in\mathcal{X}}$
24: **end procedure**

Algorithm 1: Copresheaf transformer layer integrating standard attention with learned copresheaf morphisms.

## G  Copresheaf Learning on Euclidean Data

The CopresheafConv layer leverages copresheaf structures to process data on a $D$-dimensional grid $\mathcal{X} \subset \mathbb{Z}^D$, offering distinct advantages over traditional convolutional neural networks (CNNs). By defining a copresheaf on a combinatorial complex (CC) constructed from the grid, where cells represent grid points (0-cells) and their pairwise connections (1-cells), the layer employs learnable morphisms $\rho_{y\to x} : \mathcal{F}(y) \to \mathcal{F}(x)$ that dynamically adapt to directional relationships between points. Unlike static convolutional filters, these morphisms capture anisotropic, directionally dependent interactions, preserving topological nuances of the grid's geometry. In contrast, regular convolutional kernels enforce translation invariance, limiting their ability to model spatially varying or directional

patterns. The copresheaf is defined over an adjacency neighborhood function $\mathcal{N}_{\text{adj}}(x) = \{y \in \mathcal{X} \mid \{x,y\} \in \mathcal{X}^1\}$, restricting computation to local, grid-adjacent neighbors, thus ensuring efficiency comparable to CNNs. The morphisms, potentially nonlinear, are conditioned on input features $\mathbf{h}_x^{(\ell)}, \mathbf{h}_y^{(\ell)}$ and grid positions, enabling the layer to model complex, multi-scale dependencies. This makes CopresheafConv ideal for tasks like image segmentation, 3D mesh processing, or geometric deep learning, where local and hierarchical relationships are critical. Empirical results demonstrate superior performance in capturing physical dynamics, showcasing the ability of CopresheafConv to handle spatially varying patterns. Algorithm 2 shows the pseudocode for the `CopresheafConv` used in our experiments.

---

**`CopresheafConv` on a $D$-dimensional grid**

1: **procedure** CopresheafConv($\mathcal{X} \subset \mathbf{Z}^D$, $\{h_x^{(\ell)} \in \mathcal{F}(x) = \mathbb{R}^{C_{\text{in}}}\}_{x \in \mathcal{X}}$)
2:     **for** $x \in \mathcal{X}$ **do**
3:         $m_x \leftarrow 0 \in \mathbb{R}^{C_{\text{out}}}$
4:         **for** $y \in \mathcal{N}(x)$ **do**
5:             $\rho_{y \to x} \leftarrow$ CopresheafMorphism$(y, x)$     ▷ map conditioned on $h_x^{(\ell)}, h_y^{(\ell)}$ (and thus on $(x, y)$)
6:             $m_x \leftarrow m_x + \rho_{y \to x}(h_y^{(\ell)})$
7:         **end for**
8:         $h_x^{(\ell+1)} \leftarrow m_x$
9:     **end for**
10:     **return** $\{h_x^{(\ell+1)}\}_{x \in \mathcal{X}}$
11: **end procedure**
12: **procedure** CopresheafMorphism$(y, x)$     ▷ Return the learned copresheaf morphism $\rho_{y \to x}$,
    potentially nonlinear,         ▷ conditioned on both source and target features $(h_x^{(\ell)}, h_y^{(\ell)})$.
13:     **return** $\rho_{y \to x}$
14: **end procedure**

Algorithm 2: `CopresheafConv` on a $D$-dimensional grid.

---

# H Experiments

## H.1 Mechanistic Notes for Physics Experiments

**Scope.** The empirical results for *advection* and *unsteady Stokes* appear in the main text (Sec. 6.1; Tab. 2). This appendix augments those experiments with architectural rationale and ablations-informed design choices, without introducing any new datasets, training budgets, or evaluation metrics.

**Advection (pure transport).** The advection equation

$$\partial_t u + \mathbf{c} \cdot \nabla u = 0, \qquad u(\mathbf{x}, t) = u_0(\mathbf{x} - \mathbf{c}\, t)$$

is a rigid translation. Classical self-attention aggregates values in a single global latent space, so $x \leftrightarrow y$ interactions are effectively symmetric and only weakly directional (positional encodings help but cannot enforce upwind behavior). The copresheaf transformer (CT) replaces value mixing

$$m_x^{\text{classical}} = \sum_{y \in N(x)} a_{xy}\, v_y \quad \text{by} \quad m_x^{\text{CT}} = \sum_{y \in N(x)} a_{xy}\, \rho_{y \to x}\, v_y,$$

with a learnable edge map $\rho_{y \to x}: F(y) \to F(x)$ *before* aggregation. This yields: (i) **Directionality** ($\rho_{y \to x} \neq \rho_{x \to y}$) for upwind-like asymmetry, (ii) **Phase-faithful shifts** (identity-near, head-wise maps accumulate small signed translations coherently), (iii) **Path compositionality**: products of $\rho$ along $y \to x$ chains bias the model toward consistent transports.

**Unsteady Stokes (incompressible viscous flow).** For

$$\partial_t \mathbf{u} - \nu \Delta \mathbf{u} + \nabla p = 0, \qquad \nabla \cdot \mathbf{u} = 0,$$

accuracy depends on encoding (i) anisotropic diffusion and (ii) rotational structure (vorticity), under a divergence-free constraint. Standard attention lacks built-in diffusion geometry or frame

alignment. CT's edge maps $\rho_{y\to x}$ act as local linear operators that: (i) **Align with diffusion tensors:** SPD/orthonormal variants (Table 18) mimic smoothing/rotation along principal directions. (ii) **Support structured coupling:** With cross-rank paths, CT allows vertex–edge/value transports akin to discrete parallel transport and pressure–velocity interactions, without imposing global sheaf consistency.

**Take-away** . Across both tasks, CT's gains come from a *geometric factorization* of attention into

$$(\text{weights}) \times (\text{directional transport}) \quad \text{with} \quad \rho_{y\to x} : F(y) \to F(x),$$

not merely extra parameters. This aligns with the CTNN principle that heterogeneous stalks + edge-specific morphisms provide natural inductive biases for transport-dominated and anisotropic dynamics.

## H.2 Synthetic Control Tasks

Six canonical univariate time–series patterns (*normal, cyclic, increasing trend, decreasing trend, upward shift, downward shift*) are procedurally generated. We obtain 600 sequences of length 60 (100 per class), normalised to the interval $[-1, 1]$, with an 80:20 split for training and test.

**Models and set-up**. A lightweight vanilla Transformer (32-d model, 4 heads, 2 layers) is compared with an identically sized Copresheaf Transformer, where multi-head attention is replaced by a gated outer-product tensor-attention layer with orthogonality ($\lambda = 0.01$) and sparsity ($\lambda = 10^{-4}$) regularisers. Both models share sinusoidal-with-linear-decay positional encodings, use Adam ($10^{-3}$ learning rate), batch 32, train for 15 epochs, and each experiment is repeated with three random seeds.

Table 9: Synthetic control: mean±std over 3 runs.

| Model | Max acc. (%) |
|---|---|
| Standard Transformer | $98.61 \pm 0.40$ |
| Copresheaf Transformer | $99.44 \pm 0.39$ |

**Results**. As seen in Table 9, the Copresheaf Transformer yields a consistent improvement of $+0.8$–$1.0$ percentage points (pp) over the vanilla Transformer while remaining lightweight and training in comparable wall-clock time (under one minute per run on a single GPU), highlighting the benefit of richer token-pair transformations for recognition tasks.

### H.2.1 Structure Recognition Datasets

In this experiment we consider two synthetic image datasets containing oriented ellipses or hierarchical triangles.

**Dataset (oriented ellipses)**. Each $32\times32$ RGB image contains a single black ellipse on a white background. The horizontal and vertical *semi-axes* $a, b$ are drawn uniformly from $\{4, 5, \ldots, 12\}$ pixels, and the ellipse is rotated by a random angle in $[0, 180°)$. The task is to predict the coarse orientation bin (4 bins of $45°$ each). We synthesise 6,000 images, keep 5,000:1,000 for train/validation, and rescale pixels to $[-1, 1]$.

**Dataset (hierarchical triangles)**. Each $32\times32$ RGB image contains six coloured circles—*red, green, blue, yellow, cyan, magenta*—placed on two nested equilateral triangles (inner radius 8 px, outer 12 px). Colours are randomly permuted. A hand-crafted hierarchy of linear maps (inner-triangle, outer-triangle, cross-level) is applied to the circles' one-hot colour vectors; the image is labelled 1 when the resulting scalar exceeds a fixed threshold, else 0. We generate 6,000 images and keep 5,000:1,000 for train/validation.

**Models and set-up**. Both tasks use the same compact Vision-Transformer backbone: 32-dim patch embeddings (patch size 8), 4 heads, 2 layers, learnable positional embeddings, AdamW ($3 \times 10^{-4}$ learning rate). The baseline is a *Regular ViT*; its counterpart is an identically sized *Copresheaf ViT* in which multi-head attention is replaced by an outer-product copresheaf mechanism (stalk-dim = 8). Oriented Ellipses is trained with batch 128; Hierarchical Triangles with batch 64. All runs use 30 epochs and three independent seeds.

Table 10: Validation accuracy on both synthetic vision tasks (mean±std over three seeds).

| Dataset | Regular ViT | Copresheaf ViT |
|---|---|---|
| Oriented Ellipses | $84.13 \pm 4.12$ | $96.23 \pm 0.33$ |
| Hierarchical Triang. | $95.47 \pm 1.31$ | $96.87 \pm 0.26$ |

**Results**. Across both synthetic vision tasks the Copresheaf ViT consistently surpasses the Regular ViT: a dramatic $+12.1$ pp gain on Oriented Ellipses and a subtler yet statistically tighter $+1.4$ pp on Hierarchical Triangles, while also cutting variance by an order of magnitude in the latter case (see Table 10). These outcomes underscore that replacing standard attention with copresheaf-guided outer-product maps yields robust improvements for both low-level geometric orientation recognition and higher-level nested-structure reasoning, all without a significant increase of model size or training budget.

### H.2.2   Classifying Hierarchical Polygons

Similar to the previous section, we now synthesize a hierarchy of nested regular polygons. In particular, each $32{\times}32$ RGB image contains a variable number $n \in \{6, 8, 10\}$ of coloured circles arranged on two nested regular polygons (inner radius 8 px, outer 12 px). The first $n/2$ circles form the inner polygon, the remainder the outer; colours are drawn from a palette of $n$ distinct hues and randomised per sample. A hierarchy of hand-crafted linear maps is applied to one-hot colour vectors: pairwise maps on the inner polygon ($F_{\text{inner}}$), on the outer polygon ($G_{\text{outer}}$), and a cross-level map $H$. The image is labelled 1 when the resulting scalar exceeds a threshold, else 0. For each $n$ we synthesise 6,000 images, keep 5,000:1,000 for train/validation, and normalise pixels to $[-1, 1]$.

**Models and training**. We reuse the compact ViT backbone (32-dim patch embeddings, patch size 8, 4 heads, 2 layers, learnable positional embeddings). The *Regular ViT* is compared with an identically sized *Copresheaf ViT*, which replaces multi-head attention with rank-restricted copresheaf outer-product maps (stalk-dim = 8). Both networks are trained for 10 epochs with AdamW ($3 \times 10^{-4}$ learning rate), batch 64; each configuration is run three times.

**Results**.   Figure 8 shows that the Copresheaf ViT consistently exceeds the Regular ViT at $n{=}6$ ( 0.72 vs 0.66) and regains a clear lead at $n{=}10$ ( 0.63 vs 0.57) despite both models dipping at $n{=}8$. The Copresheaf curve displays narrower uncertainty bands at the hardest setting, indicating greater run-to-run stability. Overall, copresheaf-guided attention scales more gracefully with combinatorial complexity, capturing cross-level dependencies that standard self-attention struggles to model.

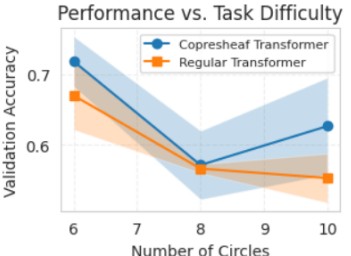

Figure 8:  Validation accuracy (mean $\pm$ 1 s.d., 3 runs) as task difficulty increases.

### H.2.3   Airfoil Self-Noise Regression

The UCI airfoil dataset ($1\,503$ rows) maps five continuous descriptors—frequency, angle of attack, chord length, free-stream velocity, Reynolds number—to the sound-pressure level (dB). Inputs and target are min–max scaled to $[0, 1]$; we keep only 400:100 train/test samples for a low-data setting.

**Models**. Both regressors share a minimalist backbone consisting sequentially of the following: 64-d token embedding 2-layer, 4-head transformer, mean pooling, scalar head. The copresheaf variant swaps dot-product attention for learned outer-product maps $\rho_{ij}$ that depend on each token pair, whereas the Regular baseline keeps standard self-attention. Training uses Adam ($10^{-4}$ learning rate), 1000 epochs, batch 32.

**Results**. On the small 100-sample test set the copresheaf regressor lowers MSE by  7.2% relative to the regular transformer and maintains sub-$10^{-4}$ run-to-run variance (see Table 11), confirming that pair-specific linear transports help model heterogeneous feature interactions even in data-scarce regimes.

Table 11: Test MSE (mean$\pm$std over two runs).

| Model | MSE |
| --- | --- |
| Regular Transformer | $0.0223 \pm 0.0001$ |
| Copresheaf Transf. | $\mathbf{0.0208} \pm 0.0002$ |

### H.3   Pixelwise Regression Tasks: Evaluating CopresheafConv2D Layers

We evaluate neural network models incorporating *CopresheafConv2D* layers, custom convolutional layers with patch-wise trainable linear morphisms, against standard convolutional models across four synthetic pixelwise regression tasks: PDE regression (Bratu and convection-diffusion equations), image denoising, distance transform regression, and edge enhancement. In all tasks, Copresheaf-based

models consistently achieve lower Mean Squared Error (MSE) and Root Mean Squared Error (RMSE) compared to standard convolutional models, suggesting improved modeling of spatial structures and relationships.

**Task Definitions**.

- *PDE regression.*
  - *Bratu equation.* A nonlinear reaction–diffusion PDE:
  
  $$-\Delta u = g(x, y)\, e^u, \quad u\big|_{\partial\Omega}= 0,$$
  
  where $g(x, y)$ is a source intensity.
  - *Convection-diffusion equation.* A transport PDE:
  
  $$-\nu\,\Delta u + c_x\,\partial_x u + c_y\,\partial_y u = g(x, y), \quad u\big|_{\partial\Omega}= 0,$$
  
  with diffusion $\nu$ and velocities $c_x, c_y$.

- *Image denoising.* Recovering clean structured images (sinusoidal patterns with a Gaussian bump, normalized to [0,1]) from Gaussian noise ($\sigma = 0.3$).

- *Distance transform regression.* Predicting the normalized Euclidean distance transform of a binary segmentation (thresholded at 0.5) of structured images.

- *Edge enhancement.* Predicting edge maps from structured images using a difference-of-anisotropic-Gaussians (DoG) transformation.

**Model and training setup**. For PDE regression and distance transform tasks, we use U-Net variants: *CopresheafUNet* (with CopresheafConv2D layers) and *ConvUNet* (with standard Conv2d layers), both with a four-level backbone (64→128→256→512 channels). For image denoising and edge enhancement, we use four-layer convolutional networks: *CopresheafNet* and *ConvNet* (1→8→16→8→1 channels). All models are trained on $64 \times 64$ inputs using the Adam optimizer and MSE loss, with task-specific settings (learning rates $10^{-3}$ or $10^{-4}$, batch sizes 8 or 16, 80–300 epochs). Results are averaged over 3 random seeds.

Table 12: Mean ($\pm$ std over 3 seeds) of MSE and RMSE across all tasks.

| Task | Model | MSE | RMSE |
|------|-------|-----|------|
| Bratu Equation | CopresheafUNet | $0.0001 \pm 0.00020$ | $0.0108 \pm 0.0003$ |
| | ConvUNet | $0.0003 \pm 0.00020$ | $0.0183 \pm 0.0007$ |
| Convection–Diffusion | CopresheafUNet | $0.0004 \pm 0.00010$ | $0.0205 \pm 0.0010$ |
| | ConvUNet | $0.0006 \pm 0.00020$ | $0.0232 \pm 0.0012$ |
| Image Denoising | CopresheafNet | $0.0010 \pm 0.00010$ | $0.0310 \pm 0.0010$ |
| | ConvNet | $0.0011 \pm 0.00020$ | $0.0336 \pm 0.0015$ |
| Distance Transform | CopresheafUNet | $0.0001 \pm 0.00002$ | $0.0105 \pm 0.0003$ |
| | ConvUNet | $0.0002 \pm 0.00003$ | $0.0156 \pm 0.0005$ |
| Edge Enhancement | CopresheafNet | $0.0008 \pm 0.00010$ | $0.0283 \pm 0.0010$ |
| | ConvNet | $0.0009 \pm 0.00020$ | $0.0300 \pm 0.0015$ |

**Take-away**. Across all tasks, replacing standard convolutional layers with CopresheafConv2D layers results in lower MSE and RMSE (see Table 12). This consistent improvement suggests that patch-wise linear maps enhance the models' ability to capture complex spatial patterns. These findings highlight the potential of Copresheaf-based architectures for pixelwise regression problems. Subsequently, we address two challenges related to token classification: real/fake token sequence detection and segment-wise token classification. Finally, we conduct a preliminary study on shape classification using copresheaf-augmented attention and graph classification a molecular benchmark, MUTAG. These are followed by applications in graph connectivity classification and text classification on TREC coarse label benchmark.

### H.3.1 Learning Token-Relations with Copresheaf Attention

We study five problems that differ only in the (non)linear operator *unknown* applied to the first half of a random token sequence (or to a second related sequence). The classifier must decide whether the tail is just a noisy copy (label 0) or a transformed version of the head (label 1).

- *Orthogonal block.* Eight 16-d "head" tokens are either copied ($+0.05$ noise) or rotated by a sample-specific orthogonal matrix before adding the same noise.

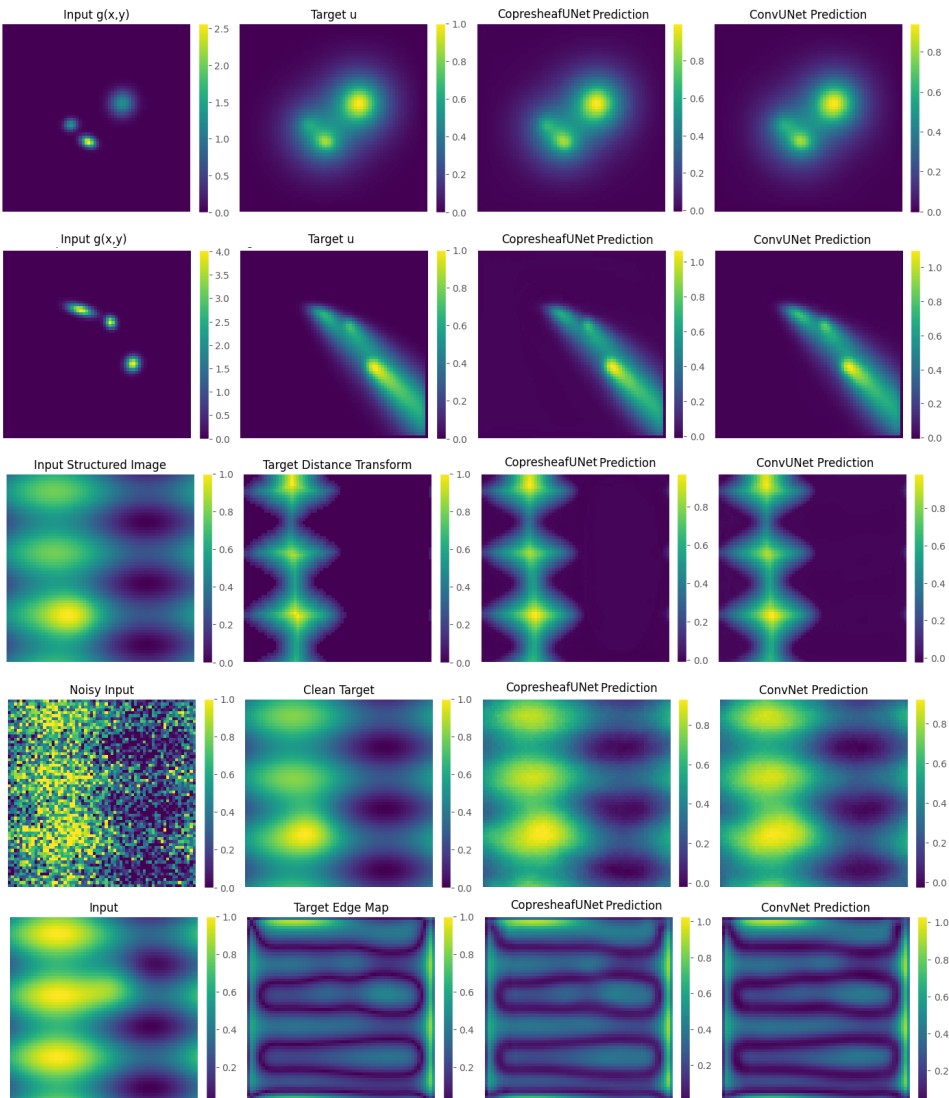

Figure 9: Model outputs across tasks. *A:* Bratu equation: input $g$, target $u$, CopresheafUNet vs. ConvUNet predictions. *B:* Convection-diffusion: input $g$, target $u$, CopresheafUNet vs. ConvUNet predictions. *C:* Distance transform: input image, target transform, CopresheafUNet vs. ConvUNet predictions. *D:* Image denoising: noisy input, clean target, CopresheafNet vs. ConvNet predictions. *E:* Edge enhancement: input image, target edge map, CopresheafNet vs. ConvNet predictions. Copresheaf-based models show subtle improvements in detail recovery.

- *Per-token scaling*. As above, but the tail is $\alpha_i\, x_i + $noise with $\alpha_i \sim$ U$[0.4, 1.6]$.
- *Rotated copy (embedded 2-D)*. Six 2-D points are mapped to 16 d by a fixed linear embed, duplicated to a 12-token sequence; the tail is either a noisy copy or the points after a random planar rotation.
- *Query and context linearity*. Two parallel sequences ($50 \times 16$ "query", $50 \times 24$ "context"). Class 0: context is a global affine transform of the query with partly correlated semantics. Class 1: context comes from a quadratic warp and weak semantic correlation.
- **Affine vs. quadratic token relations**. Two parallel sequences (length-6, query dim 16, context dim 24) are considered. For class 0, the context is a linear spatial transformation (rotation and translation) of the query plus correlated semantic noise. For class 1, the context is generated via a spatial quadratic (nonlinear) transformation with weaker semantic correlation.

Table 13: Mean accuracy ($\pm$ std, 3 seeds).

| Task | Classic | Copresheaf |
|------|---------|------------|
| Orthogonal block | $0.732 \pm 0.009$ | $0.928 \pm 0.007$ |
| Per-token scaling | $0.521 \pm 0.005$ | $0.707 \pm 0.004$ |
| Rotated copy (2-D) | $0.739 \pm 0.010$ | $0.896 \pm 0.033$ |
| Query to context | $0.608 \pm 0.046$ | $0.992 \pm 0.012$ |
| Affine vs. Quadratic Relations | $0.588 \pm 0.047$ | $0.900 \pm 0.027$ |

**Data**. Tasks 1–2 use 16 tokens, task 3 uses 12, task 4 uses two length-50 sequences, task 5 uses two length-6 sequences. For each of three seeds we draw $4{,}096{:}1{,}024$ train/test sequences (task 4–5: 320:80).

**Backbone and training**. A tiny Transformer encoder (4 heads, token dim 16, stalk-dim 4) $\rightarrow$ mean-pool $\rightarrow$ 2-way classifier. *Classic* uses vanilla dot-product attention; *copresheaf* augments it with learned token-pair copresheaf maps (we chose General Copresheaf for the first two tasks, and Non-linear MLP for tasks 3–5). We train for 8 / 12 / 10 / 10 / 10 epochs respectively with Adam ($10^{-3}$ learning rate), batch 64.

**Take-away**. Across in-sequence, element-wise, embedded-geometric, and cross-sequence settings—including varying degrees of spatial and semantic correlation—injecting copresheaf transports into self-attention consistently lifts accuracy significantly (up to +38 pp, as seen in Table 13). This highlights a general principle: tasks whose signals reside in *relations between tokens* rather than in absolute token content strongly benefit from explicitly modeling these relations through learnable copresheaf-induced attention.

**Limited attention capacity**. We study the impact of attention capacity on relational reasoning, by varying the number of heads in a small transformer and testing its ability to classify the query to context dataset provided in Section H.3.1.

We evaluate three setups: a baseline transformer, the same model augmented with positional encoding (PE), and a copresheaf-augmented transformer with 2 heads. Figure 10 shows accuracy as a function of attention capacity.

While positional encoding improves baseline accuracy slightly, the copresheaf-augmented attention with just two heads outperforms all classic models, even those with eight times more heads. This highlights the value of inductive relational structure over brute-force capacity scaling.

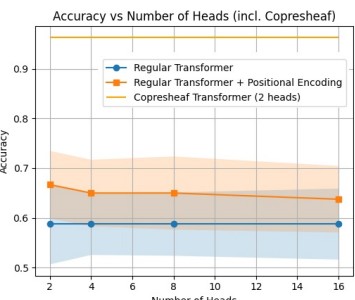

Figure 10: Accuracy as a function of number of attention heads. Even with low capacity, the copresheaf-augmented model perfect generalization of the Query to Context task.

#### H.3.2 Segment-wise Token Classification

We test whether copresheaf attention improves token-level classification in a sequence partitioned into contiguous segments with distinct patterns. The classifier must assign a segment label (0, 1, or 2) to each token based on its local context.

**Data**. Each input is a sequence of 100 tokens, where each token is a 16-dimensional feature vector. The sequence is divided into three contiguous segments, each following a different pattern: (i) a cosine oscillation, (ii) a linearly increasing ramp, or (iii) an exponentially decreasing signal, with additive noise. The task is to predict the correct segment label for each token. We generate 300 training sequences and evaluate on three random seeds.

**Backbone and training**. We use a 2-layer encoder with 4 heads, token dim 16, and stalk-dim 4. A linear classifier maps each token to one of 3 segment labels. *Classic* is standard attention; *copresheaf* augments each attention head with learned per-token transport maps. We train for 10 epochs using Adam ($10^{-3}$ learning rate), with batch size 32.

**Take-away.** Injecting copresheaf structure into attention substantially improves token-wise

segmentation accuracy, especially with expressive MLP kernels (see Table 14). This demonstrates that local consistency constraints enforced by per-token transport maps help resolve semantic boundaries even in noisy, positionally ambiguous settings.

Table 14: Mean segmentation accuracy ($\pm$ std, 3 seeds).

| Model | Accuracy |
|---|---|
| Classic | $0.705 \pm 0.010$ |
| Copresheaf-FC | $0.833 \pm 0.015$ |
| Copresheaf-MLP | $0.831 \pm 0.007$ |
| Copresheaf-SPD | $0.743 \pm 0.017$ |

### H.3.3 Topological Shape Classification

We evaluate copresheaf-augmented attention on a synthetic 3D point cloud classification task. Each input is a set of 128 points in $\mathbb{R}^3$ sampled from one of four classes: cube, sphere, torus, and twisted torus. Rotations are applied to remove alignment bias.

**Data**. The dataset consists of 480:160 train/test samples, balanced across the four classes. Each point cloud is processed as a sequence of 128 points with 3D coordinates.

**Backbone and training**. Both models use a 4-layer point transformer with 4 heads and head dimension 32. The *Classic* model uses standard self-attention. The *Copresheaf* model augments attention with diagonal copresheaf morphisms. We train each model for 50 epochs using AdamW and cosine learning rate decay across 3 random seeds.

**Take-away**. Copresheaf-augmented attention improves accuracy on 3D shape classification by enhancing sensitivity to latent geometric structure (see Table 15).

Table 15: Mean accuracy ($\pm$ std, 3 seeds).

| Model | Accuracy |
|---|---|
| Classic | $0.708 \pm 0.031$ |
| Copresheaf | $0.746 \pm 0.034$ |

### H.4 TREC Text Classification Task

We evaluate two transformer-based models on the TREC coarse-label question classification task, which involves categorizing questions into 6 classes (e.g., abbreviation, entity, description). The models are: *Classic*, a standard transformer with multi-head self-attention; and *Copresheaf-FC*, which incorporates a GeneralSheafLearner to model stalk transformations.

**Task definition**. The TREC dataset consists of questions labeled with one of 6 coarse categories. Inputs are tokenized questions truncated to 16 tokens, mapped to a vocabulary of size $|\mathcal{V}|$, and embedded into an 8-dimensional space. The task is to predict the correct class label for each question.

**Model and training setup**. Both models use a single transformer block with 2 attention heads, an embedding dimension of 8, and a stalk dimension of 4 for the Copresheaf-based model. The architecture includes an embedding layer, a transformer block with attention and feed-forward components, adaptive average pooling, and a linear classifier. Compared to state-of-the-art (SOTA) models, which often employ multiple transformer layers and high-dimensional embeddings for maximal performance, our networks are intentionally small, using a single block and low embedding dimension to prioritize computational efficiency and controlled experimentation. We train on the TREC training set (5452 samples) over 30 epochs with a batch size of 32, using the Adam optimizer with a learning rate of $10^{-3}$ and cross-entropy loss. The test set (500 samples) is used for evaluation. Each experiment is repeated over 4 random seeds. As seen in Table 16, the copresheaf models outperform their SOTA counterparts.

Table 16: Mean ($\pm$ std over 4 seeds) of test accuracy for the TREC classification task.

| Model | Test Accuracy |
|---|---|
| Classic | $0.7320 \pm 0.0080$ |
| Copresheaf-FC | $0.7500 \pm 0.0150$ |

### H.5 Mixed Dirichlet–Robin Reaction–Anisotropic Diffusion on Cellular Complexes

**Problem.**. Let $\Omega \subset \mathbb{R}^2$ be a Lipschitz domain with a hole (nonconvex), and let $\partial\Omega = \partial\Omega_D \cup \partial\Omega_R$ with $\partial\Omega_D \cap \partial\Omega_R = \varnothing$. We consider the steady reaction–anisotropic diffusion equation

$$-\nabla\cdot\big(\mathbf{K}(x)\,\nabla u(x)\big) \;+\; \lambda(x)\,u(x) \;+\; \sigma(x)\,u(x) \;=\; g(x) \quad \text{in } \Omega\setminus H, \tag{13}$$

with mixed boundary conditions

$$
\begin{aligned}
u(x) &= u_B(x) && \text{on } \partial\Omega_D, \\
\mathbf{n}(x)\cdot\mathbf{K}(x)\,\nabla u(x) \;+\; r(x)\,u(x) &= \alpha(x)\,u_B(x) && \text{on } \partial\Omega_R.
\end{aligned}
\tag{14}
$$

Here $\mathbf{K}(x) \in \mathbb{R}^{2\times2}$ is a symmetric positive definite anisotropy tensor (we store its local components as $(K_{xx}, K_{xy}, K_{yy})$), $\lambda, \sigma \geq 0$ are reaction weights, $g$ is a source, $\alpha, r$ are Robin coefficients, $u_B$ a boundary signal, and $\mathbf{n}$ the outward unit normal. All spatial fields are discretized and provided as vertex/edge attributes on the mesh.

**Discretization on simplicial complexes.** Let $\mathcal{K} = (S, \mathcal{X}, \mathrm{rk})$ be a 2D simplicial complex (triangulation) of $\Omega \setminus H$ with 0-, 1-, and 2-cells $\mathcal{X}^0, \mathcal{X}^1, \mathcal{X}^2$. We employ the neighborhood functions (Def. 2)

$$\mathcal{N}_{\mathrm{inc}}^{(0,1)}(v) = \{\, e \in \mathcal{X}^1 \mid v \subset e \,\}, \qquad \mathcal{N}_{\mathrm{adj}}^{(0,0)}(v) = \{\, w \in \mathcal{X}^0 \mid \exists t \in \mathcal{X}^2 : v \subset t,\ w \subset t \,\},$$

and the induced directed graphs $G_N$. Edgewise anisotropy enters through per-edge tensors $(K_{xx}, K_{xy}, K_{yy})$ and yields a stiffness-like coupling consistent with equation 13. Vertex features carry $(u_B, g, \lambda, \sigma, \alpha, r)$; edge features carry $(K_{xx}, K_{xy}, K_{yy})$. Masks on $\partial\Omega_D$ and $\partial\Omega_R$ enforce the boundary conditions in the loss.

**Higher-order copresheaf structure.** We equip $G_\mathcal{N}$ with an $\mathcal{N}$-dependent copresheaf $(\mathcal{F}, \rho, G_N)$ over multiple neighborhoods:

stalks: $\quad \mathcal{F}(v) = \mathbb{R}^{d_0},\ \mathcal{F}(e) = \mathbb{R}^{d_1},\ \mathcal{F}(t) = \mathbb{R}^{d_2} \quad (v \in \mathcal{X}^0,\ e \in \mathcal{X}^1,\ t \in \mathcal{X}^2),$

morphisms: $\quad \rho_{w\to v}^{(0\leftarrow0)} : \mathcal{F}(w) \to \mathcal{F}(v)$ for $w \in N_{\mathrm{adj}}^{(0,0)}(v)$, $\quad \rho_{e\to v}^{(0\leftarrow1)} : \mathcal{F}(e) \to \mathcal{F}(v)$ for $e \in \mathcal{N}_{\mathrm{inc}}^{(0,1)}(v)$,

$$\rho_{t\to v}^{(0\leftarrow2)} : \mathcal{F}(t) \to \mathcal{F}(v) \text{ for } t \in \{\, \tau \in \mathcal{X}^2 \mid v \subset \tau \,\}.$$

Thus information flows *across ranks* (edges/triangles $\to$ vertices) in addition to *within rank* (vertex $\leftrightarrow$ vertex). This realizes a higher-order CMPNN and may be viewed as a principled generalization of topological neural networks (TNNs), where transport maps are learned as copresheaf morphisms instead of being tied to fixed co(boundary) operators.

**Copresheaf Cellular Transformer layer.** For features $h_x^{(\ell)} \in \mathcal{F}(x)$ at cell $x$ in layer $\ell$, we use copresheaf self- and cross-attention :

$$h_v^{(\ell+1)} = \beta\left( h_v^{(\ell)}, \underbrace{\sum_{w\in\mathcal{N}_{\mathrm{adj}}^{(0,0)}(v)} a_{vw}\,\rho_{w\to v}^{(0\leftarrow0)}(v_w)}_{0\to0} \oplus \underbrace{\sum_{e\in\mathcal{N}_{\mathrm{inc}}^{(0,1)}(v)} a_{ve}\,\rho_{e\to v}^{(0\leftarrow1)}(v_e)}_{1\to0} \oplus \underbrace{\sum_{t\ni v} a_{vt}\,\rho_{t\to v}^{(0\leftarrow2)}(v_t)}_{2\to0} \right),$$

where $q_x = W_q h_x^{(\ell)}$, $k_x = W_k h_x^{(\ell)}$, $v_x = W_v h_x^{(\ell)}$, the coefficients $a_{xy} = \mathrm{softmax}_{y\in N(x)}(\langle q_x, k_y \rangle / \sqrt{p})$, and $\beta$ is a residual MLP with normalization. We instantiate $\rho$ with sheaf-style transport maps *SheafFC* $\rho_{y\to x} = \mathrm{Id} + \tanh(W\,[q_x; k_y])$ (zero-initialized), optionally constrained to SPD via $\rho = \mathrm{Id} + QQ^\top$. All morphisms act per head and per rank; weights are shared across complexes but conditioned on $(h_x, h_y)$, enabling directionality and anisotropy. Observe that when the copresheaf structure is the identity, we retain the Cellular Transfomer introduced in Barsbey et al. [2025].

**Dataset and training**. We synthesize simplicial complexes by jittered grids with warped holes to emulate nontrivial $\partial\Omega_R$. Per-sample scalar fields $(u_B, g, \lambda, \sigma, \alpha, r)$ are sampled on vertices; $(K_{xx}, K_{xy}, K_{yy})$ on edges. Ground-truth $u$ is obtained by a finite-element solve of equation 13–equation 14. We train to regress $\hat{u}$ at vertices with MSE, enforcing Dirichlet via clamped targets and Robin via boundary-weighted residuals. Optimization uses AdamW ($10^{-3}$ learning rate), cosine schedule, grad clip 0.8, layer norm, and vertex/edge feature normalization computed per split.

**Baselines and size control**. We compare the *Cellular Transformer on vertices* (same depth/width, no copresheaf maps) to our *Copresheaf Cellular Transformer on $\mathcal{K}$* using identical token dimensions and heads; differences arise solely from the learnable morphisms and cross-rank attention paths.

**Results**. Quantitative test MSE averaged across $n=4$ seeds is summarized below (lower is better).

Error maps highlight that copresheaf cross-rank transport (1$\to$0 and 2$\to$0) attenuates bias along anisotropy directions of $\mathbf{K}$ and reduces leakage across $\partial\Omega_R$. Despite identical model size, higher-order morphisms recover boundary layers and interior ridges more faithfully. See Figure 11 for a sample of the results.

Table 17: Mixed Dirichlet–Robin reaction–anisotropic diffusion on simplicial complexes. Test MSE (mean $\pm$ std over $n$=4 seeds). Lower is better.

| Model | Test MSE $\downarrow$ | Runs ($n$) |
|---|---|---|
| Cellular Transformer Barsbey et al. [2025] | $0.3277 \pm 0.0408$ | 4 |
| Copresheaf Cellular Transformer | $\mathbf{0.3172} \pm 0.0365$ | 4 |

**Take-away**. Endowing the simplicial complex with a *higher-order* copresheaf, learning $\rho_{y \to x}$ across adjacency and incidence neighborhoods and across ranks, yields a CMPNN that generalizes TNNs while remaining faithful to the PDE structure in equation 13. Observe that such a network is a generalization of the cellular transformer introduced in Barsbey et al. [2025]. The learned, directional transport improves anisotropic coupling and mixed-boundary handling without increasing parameter count, offering a principled and practical route to physics-aware attention on combinatorial domains.

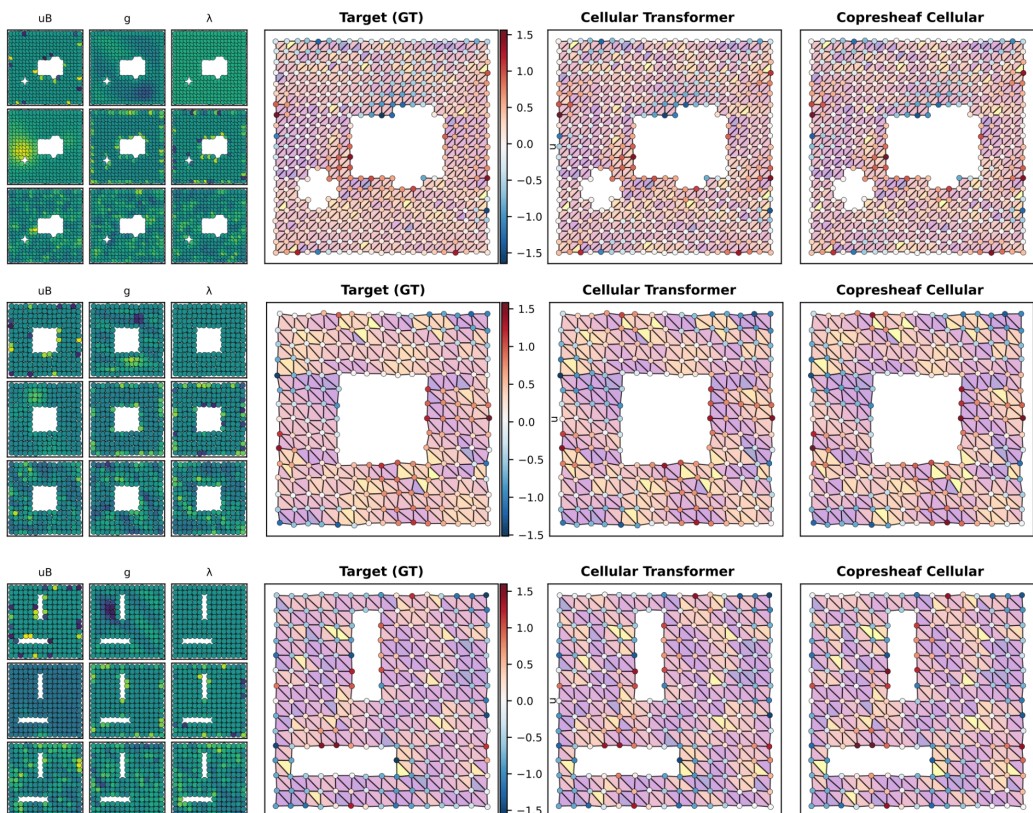

Figure 11: Mixed Dirichlet–Robin reaction–anisotropic diffusion on triangulated complexes. For each example (rows), the left block shows a 3×3 tile of normalized inputs: Dirichlet values $u_B$, source $g$, reaction $\lambda$, spatial reaction field $\sigma$, Robin coefficients $\alpha$ and $r$, and per-vertex edge-averaged conductivity features $(\overline{K_{xx}}, \overline{K_{xy}}, \overline{K_{yy}})$. The remaining panels show the ground-truth solution $u$, the Cellular Transformer prediction, and the Copresheaf Cellular Transformer prediction, respectively. Rows illustrate diverse geometries and boundary layouts. Copresheaf transport produces slightly crisper fields and cleaner boundary behavior.

### H.6 Catalogue of Copresheaf Maps

We also provide the table of copresheaf maps that we used throughout our experiments in Table 18.

## Computational Complexity

This section analyzes the computational complexity of the Copresheaf Message-Passing Neural Network (CMPNN) and the Copresheaf Transformer (CT), as defined in the framework of Copresheaf

Table 18: Catalogue of copresheaf maps $\rho_{y \to x}$ used in our training our copresheaf transformer model. All maps act stalk-wise and are evaluated independently for each attention head; $\sigma$ is the logistic function.

| Map family | Copresheaf map $\rho_{y \to x}$ (per head) | Learnable params |
|---|---|---|
| **General Copresheaf** | $\rho_{y \to x} = \tanh\left(W \begin{bmatrix} q_x \\ k_y \end{bmatrix}\right)$ | $W \in \mathbb{R}^{2d \times d^2}$ |
| **Pre-Linear Map** | $\rho_{y \to x} = q_x \, k_y^\top$ | none |
| **Diagonal MLP Map** | $\rho_{y \to x} = \mathrm{diag}(\sigma(\mathrm{MLP}[q_x, k_y]))$ | 2-layer MLP |
| **Graph Attention Map** | $\rho_{y \to x} = \sigma(\mathrm{MLP}[q_x, k_y]) \, I_d$ | 2-layer MLP |
| **Vision Spatial Map** | $\rho_{y \to x} = \sigma(\mathrm{MLP}(p_x - p_y))$ 
 $(p_x, p_y \in [0,1]^2$ pixel coords$)$ | 2-layer MLP |
| **Outer-Product Map** | $\rho_{y \to x} = W_q q_x \, (W_k k_y)^\top$ | $W_q, W_k \in \mathbb{R}^{d \times d}$ |
| **Non-linear MLP Map** | $\rho_{y \to x} = \mathrm{reshape}(\mathrm{MLP}[q_x, k_y])$ | 2-layer MLP $(2d \to 2d \to d^2)$ |
| **Gaussian RBF Map** | $\rho_{y \to x} = e^{-\|q_x - k_y\|^2 / 2\sigma^2} \, I_d$ | $\sigma$ (scalar) |
| **Dynamic Map** | $\rho_{y \to x} = \mathrm{reshape}(W_f q_x)$ | $W_f \in \mathbb{R}^{d \times d^2}$ |
| **Bilinear Map** | $\rho_{y \to x} = (b^\top (q_x, k_y)) \, I_d$ | $b \in \mathbb{R}^{d \times d}$ |
| **SheafFC Map** | $\rho_{y \to x} = I_d + \tanh\left(W \begin{bmatrix} q_x \\ k_y \end{bmatrix}\right)$ | $W \in \mathbb{R}^{2d \times d^2}$ (zero init) |
| **SheafMLP Map** | $\rho_{y \to x} = I_d + \tanh\left(\mathrm{MLP}[q_x, k_y]\right)$ | 2-layer MLP (last layer zero init) |
| **SheafSPD Map** | $\rho_{y \to x} = I_d + QQ^\top, \quad Q = W \begin{bmatrix} q_x \\ k_y \end{bmatrix}$ | $W \in \mathbb{R}^{2d \times d^2}$ (no bias) |

Topological Neural Networks (CTNNs). We consider a directed graph $G_N = (V_N, E_N)$, induced by a combinatorial complex $\mathcal{X}$ with neighborhood function $\mathcal{N}$, where $|V_N| = n$, $|E_N| = m$, and the average degree is $c = m/n$. Each vertex $x \in V_N$ is assigned a feature space $\mathcal{F}(x) = \mathbb{R}^d$, and each edge $y \to x \in E_N$ has a linear map $\rho_{y \to x} : \mathbb{R}^d \to \mathbb{R}^d$, computed at runtime using a map family from Table 16 (e.g., General Copresheaf, Low-rank). The complexity of computing a single morphism $\rho_{y \to x}$ is denoted $C(\rho)$.

Unless otherwise stated, we assume sparse graphs (i.e., $m = \Theta(n)$). We derive the following propositions for per-layer complexities, followed by a comparison with standard architectures.

**Proposition (Copresheaf Message-Passing Complexity).** Consider a directed graph $G_N = (V_N, E_N)$ induced by a combinatorial complex $X$ with neighborhood function $\mathcal{N}$, where $|V_N| = n$, $|E_N| = m$, $\dim \mathcal{F}(x) = d$, and let $\oplus$ be any permutation-invariant aggregator. Let each $y \to x \in E_N$ have a linear map $\rho_{y \to x} \in \mathbb{R}^{d \times d}$ computed at runtime via a map family with complexity $C(\rho)$. The per-layer computational complexity of the CMPNN, as defined in Definition 9, is

$$T_{\mathrm{CMPNN}} = O(n\, C(\rho) + m\, d^2 + n\, d^2).$$

*Proof.* The CMPNN message-passing operation involves computing morphisms, applying them to features, aggregating messages, and updating vertex states:

1. **Morphism Computation:** For each edge $y \to x \in E_N$, compute $\rho_{y \to x} \in \mathbb{R}^{d \times d}$ based on source and target features $(h_y^{(\ell)}, h_x^{(\ell)}) \in \mathbb{R}^{2d}$, using a map family from Table 18. This costs $O(m\, C(\rho))$.

2. **Morphism Application:** Apply $\rho_{y \to x}$ to the feature vector $h_y^{(\ell)} \in \mathbb{R}^d$. This matrix–vector multiplication requires $O(d^2)$ operations per edge, totaling $O(m\, d^2)$.

3. **Aggregation:** For each vertex $x \in V_N$, aggregate messages from neighbors $y \in \mathcal{N}(x)$ via $\bigoplus_{y \in \mathcal{N}(x)} \rho_{y \to x} h_y^{(\ell)}$. Each addition involves a $\mathbb{R}^d$ vector, costing $O(d)$ per neighbor. With $|\mathcal{N}(x)|$ neighbors, this step is $O(|\mathcal{N}(x)| \cdot d)$. Summing over all vertices: $O\left(\sum_{x \in V_N} |\mathcal{N}(x)| \cdot d\right) = O(m \cdot d)$.

4. **Update:** Assign $h_x^{(\ell+1)} = \beta(\cdot)$ with negligible cost. If an optional single-layer MLP with $d$ hidden units is applied (common in GNNs), the cost per vertex is $O(d^2)$, yielding $O(n \cdot d^2)$.

Total: $O\big(n\,C(\rho) \,+\, m\,d^2 \,+\, m\,d \,+\, n\,d^2\big)$. Since $m\,d^2$ dominates and $m = \Theta(n)$, this simplifies to $O\big(n\,C(\rho) + (m+n)\,d^2\big)$. □

**Proposition (Copresheaf Transformer Complexity).** Consider a directed graph $G_N = (V_N, E_N)$ induced by a combinatorial complex $X$ with neighborhood function $\mathcal{N}$, where $|V_N| = n$, $|E_N| = m$, and the average degree is $c = m/n$. Each vertex $x \in V_N$ has a feature space $\mathcal{F}(x) = \mathbb{R}^d$, and each of $H$ attention heads computes a morphism $\rho_{y \to x}^{(h)} \in \mathbb{R}^{(d/H) \times (d/H)}$ via a map family (Table 16) with complexity $C(\rho)$. The per-layer computational complexity of the Copresheaf Transformer, as defined in Algorithm 1, with sparse cross-attention based on $\mathcal{N}$, is

$$T_{\mathrm{CT}} \;=\; O\big(H\,m\,C(\rho) \,+\, m\,d^2 \,+\, n\,d^2 \,+\, n\,d\,H\big).$$

*Proof.* The Copresheaf Transformer integrates self-attention within cells of equal rank and sparse cross-attention between neighbors, using learned copresheaf morphisms. Each head attends on features (queries/keys) of dimension $p = d/H$:

1. **Morphism Construction:** For each edge $y \to x \in E_N$ and each head, compute $\rho_{y \to x}^{(h)} \in \mathbb{R}^{(d/H) \times (d/H)}$ based on $(h_y^{(\ell)}, h_x^{(\ell)})$. The complexity per morphism is $C(\rho)$, totaling $O(H\,m\,C(\rho))$.

2. **Self- and Cross-Attention:** Compute query ($Q$), key ($K$), and value ($V$) matrices per head ($d/H$ dimensions), costing $O(n\,d^2/H)$ across all heads. For sparse cross-attention, each vertex attends to $|\mathcal{N}(x)|$ neighbors, with attention scores $QK^\top$ and aggregation costing $O(|\mathcal{N}(x)| \cdot d/H)$ per vertex per head. Summing over vertices and heads: $O(H \cdot \sum_x |\mathcal{N}(x)| \cdot d/H) = O(m\,d)$.

3. **Morphism Application:** Apply $\rho_{y \to x}^{(h)}$ to value vectors ($d/H$ per head) on each edge, costing $O(d^2/H)$ per head, or $O(d^2)$ across $H$ heads. For $m$ edges this contributes $O(m\,d^2)$.

4. **Output and Feed-Forward:** Combine head outputs and apply a per-token feed-forward network (FFN) with $d$ hidden units, costing $O(n\,d^2)$.

Total: $O\big(H\,m\,C(\rho) + m\,d^2 + n\,d^2 + n\,d\,H\big)$. □

**Comparison to Standard Architectures.** Compared to standard GNNs (e.g., GCN) and dense Transformers, CMPNN and CT incur higher costs due to morphism computation (the $C(\rho)$ terms) and morphism application ($m\,d^2$). On sparse graphs with small $H$, CMPNN's complexity is $O(n\,C(\rho) + m\,d^2 + n\,d^2)$, and CT's is $O(H\,m\,C(\rho) + m\,d^2 + n\,d^2 + n\,d\,H)$. These costs enable superior expressiveness, as shown empirically.

| Model | Per-layer complexity (sparse graph) |
|---|---|
| GCN | $O(m\,d \,+\, n\,d^2)$ |
| CMPNN | $O\big(n\,C(\rho) \,+\, m\,d^2 \,+\, n\,d^2\big)$ |
| Transformer (Dense) | $O(n^2\,d \,+\, n\,d^2)$ |
| Copresheaf Transformer (CT) | $O\big(H\,m\,C(\rho) \,+\, m\,d^2 \,+\, n\,d^2 \,+\, n\,d\,H\big)$ |

Table 19: Per-layer computational complexity on sparse graphs. If $\rho$ is diagonal (or rank-$r$), replace $d^2$ by $d$ (or $r\,d$) in the morphism-application terms.

# I  Extended Related Work on Topological and Sheaf Neural Networks

**Foundations of sheaf theory**. Sheaf theory offers a unifying categorical framework across algebraic geometry, topology, and algebra [Bredon, 1997]. Early computer-science applications exploited its logical structure [Fourman et al., 1977, Goguen, 1992], and Srinivas generalized pattern matching via sheaves on Grothendieck topologies [Srinivas, 1993], later extended to NP-hard problems [Conghaile, 2022, Abramsky, 2022]. Cellular sheaves, formalized in [Curry, 2014], underpin discrete topological data analysis and signal processing [Ghrist and Hiraoka, 2011, Robinson, 2014]. Hansen & Ghrist

introduced the sheaf Laplacian [Hansen and Ghrist, 2019b], learnable by convex optimization [Hansen and Ghrist, 2019a]. Connection sheaves model discrete vector bundles [Singer and Wu, 2012] and support manifold learning and Gaussian processes [Peach et al.]. GKM-sheaves further connect equivariant cohomology and sheaf cohomology, enriching this framework with applications to torus actions on CW complexes [Al-Jabea and Baird, 2018].

**Higher-order representations in deep learning**. The growing interest in higher-order network models [Papamarkou et al., 2024, Battiston et al., 2020, Bick et al., 2023] has catalyzed geometric and topological deep learning. Techniques include simplicial, hypergraph, and cellular message-passing schemes [Gilmer et al., 2017, Ebli et al., 2020, Hayhoe et al., 2022, Hajij et al., 2020, Bunch et al., 2020], skip connections [Hajij et al., 2022], and convolutional operators [Jiang et al., 2019, Feng et al., 2019]. Recent years also witnessed a leap in higher-order diffusion models for graph generation [Huang and Birdal, 2025] as well as higher-order (cellular) transformers [Ballester et al., 2024].

**Sheaf neural networks**. In recent years, sheaf-based generalizations of graph neural networks (GNNs) have demonstrated notable improvements on tasks involving heterogeneous or non-Euclidean data. Hansen and Gebhart [Hansen and Gebhart, 2020] first introduced sheaf neural networks (SNNs), which generalize graph neural networks (GNNs) by replacing neighborhood aggregation with learnable linear "restriction maps", thereby customizing information flow between nodes. By allowing each edge to carry its own linear transformation, SNNs capture relationships in heterophilic graphs more effectively than degree-normalized convolutions. Building on this idea, Bodnar et al. [Bodnar et al., 2022] proposed Neural Sheaf Diffusion (NSD), which jointly learns the underlying sheaf structure and the diffusion dynamics. NSD layers adaptively infer the sheaf Laplacian from data, mitigating the oversmoothing problem common in deep GNNs and achieving superior performance on a range of heterophilic benchmark datasets. Barbero et al. [Barbero et al., 2022b] then combined NSD's principled diffusion with attention mechanisms to formulate Sheaf Attention Networks (SANs). SANs modulate self-attention weights by the learned sheaf maps, preserving long-range dependencies while respecting local sheaf geometry.

Alternative formulations include Bundle Neural Networks (BNNs) by Bamberger et al. [Bamberger et al., 2024], which reinterpret propagation as parallel transport in a flat vector bundle rather than discrete message passing. Duta et al. [Duta et al., 2023] extended sheaf methods to hypergraphs, defining linear and nonlinear hypergraph Laplacians that capture higher-order interactions among groups of nodes. On manifold-structured data, Tangent Bundle Neural Networks (TBNNs) proposed by Battiloro et al. [Battiloro et al., 2024b] treat features as elements of tangent spaces and propagate them along estimated geodesics, bridging continuous and discrete models.

**Attention mechanisms in higher-order structures**. Attention mechanisms have been generalized to hypergraphs [Kim et al., 2020, Bai et al., 2021] and simplicial complexes [Goh et al., 2022, Battiloro et al., 2024a], among else via Hodge or Dirac operators. On combinatorial complexes, feature-lifting attention facilitates hierarchical information propagation [Giusti et al., 2023, Hajij et al., 2023b].

**Applications and extensions**. These sheaf-theoretic architectures have found diverse applications, from multi-document summarization [Atri et al., 2023] and recommendation systems [Purificato et al., 2023] to community detection via sheaf cohomology [Wolf and Monod, 2023] and personalized federated learning with Sheaf HyperNetworks [Nguyen et al., 2024, Liang et al., 2024]. In representation learning, many knowledge-graph embedding techniques have been reinterpreted as sheaf global-section problems [Gebhart et al., 2023, Kvinge et al., 2021]. Collectively, these advances highlight the expressive power and flexibility of sheaf-based models in handling complex, heterogeneous, and higher-order data domains.

