# OpenReview forum: "Copresheaf Topological Neural Networks: A Generalized Deep Learning Framework"
_NeurIPS.cc/2025/Conference — NeurIPS 2025 poster_

### Official Review · Reviewer_wnSD · 2025-06-30

**Clarity:** 4
**Significance:** 4
**Originality:** 4
**Rating:** 5
**Confidence:** 3

**Summary:**

This paper introduces Copresheaf Topological Neural Networks (CTNNs), a generalization of standard message passing that extends beyond pairwise (node-to-node) interactions. CTNNs operate on Combinatorial Complexes (CCs) by leveraging neighboring functions to define Copresheaf Neighborhood Matrices (CNMs). Unlike traditional binary adjacency matrices, CNMs encode richer local dependencies through copresheaf maps between data associated with the cells of a CC. Building on this foundation, the authors define the Copresheaf Adjacency Matrix (CAM) and Copresheaf Incidence Matrix (CIM), enabling a novel message passing framework that unifies and generalizes across multiple architectures, including CNNs, GNNs, and Transformers. CTNNs provide a flexible and expressive mechanism for capturing complex, multi-scale relationships in structured data. Experiments on standard benchmarks demonstrate the superior expressivity of CTNNs compared to popular baselines such as GCNs and GraphSAGE.

**Questions:**

1) In your construction, what is the precise base category C over which the copresheaf is defined? Are the cells of the combinatorial complex treated as the objects of C, with morphisms defined by inclusion relations (e.g., x \subset y) or directed neighborhood links (e.g., x → y)? Moreover, are you modeling a strict copresheaf — that is, a covariant functor F: C → Vect_R — where each object x is assigned a vector space F(x), and each morphism f: x → y is assigned a linear map F(f): F(x) → F(y), such that F(id_x) = id_{F(x)} and F(g o f) = F(g) o F(f)? Or are you using the term "copresheaf" more loosely, to describe a set of learned linear maps ρ_{x→y} \in Hom(F(x), F(y)) without requiring functorial properties like compositionality or identity preservation? Just want to make sure I'm understanding your use of the term correctly.

2) CTNNs operate over combinatorial complexes that include cells of varying dimensions (e.g., vertices, edges, faces). The model defines neighborhood functions like N_adj^(r,k)(x) and N_inc^(r,k)(x), as seen in Definition 8, which explicitly allow message passing between cells of different ranks. This suggests that mixed-rank message aggregation is a fundamental part of the framework.

Given this, a few important questions arise:

* How are feature spaces F(x) defined and aligned across different cell types or ranks?

* When dim(F(y)) ≠ dim(F(x)), how are the linear maps rho_{y→x} in Hom(F(y), F(x)) implemented to handle these differences in dimensionality?

* In message passing (e.g., Proposition 3), where aggregation is performed over multiple neighborhood functions N_k(x), how are contributions from cells of different types or ranks aggregated meaningfully, especially when tensor products and summations are involved? Clarifying these points would help me better understand how CTNNs handle heterogeneous and multi-scale interactions in practice.

3) You describe the CTNN layer using copresheaf-based neighborhood matrices (CNM, CAM, CIM), which involve learned linear maps for each directed relation. However, the paper does not provide any complexity analysis or runtime/memory benchmarks.

* How does the runtime and memory usage of a CTNN layer scale compared to a standard GNN or Transformer layer on the same data?

* Are there cases where the number of ρ_{y→x} maps becomes a bottleneck (e.g., in dense or high-rank complexes)?

* Have you measured wall-clock time, peak memory usage, or FLOPs to quantify the overhead of this generalization?

**Ethical Concerns:**

["NO or VERY MINOR ethics concerns only"]

**Final Justification:**

I thank the authors for their thoughtful and detailed rebuttal. After carefully considering their responses and discussions with the other reviewers and the area chair, I am increasing my score to 5.

Since all of my core questions were convincingly addressed, and the overall contribution is sound and relevant to the community, I have given significant weight to the improved clarity and completeness offered during the rebuttal.

Overall, I believe this paper makes a meaningful contribution and merits acceptance.

**Limitations:**

Yes

**Quality:**

3

**Strengths And Weaknesses:**

# Technical Strengths

1. **Generalized Theoretical Framework**
CTNNs introduce a principled message-passing paradigm rooted in category theory and combinatorial complexes, unifying GNNs, Transformers, and sheaf-based models under a single copresheaf formalism.

2. **Directional and Heterogeneous Representations**
By assigning task-specific feature spaces and learnable maps to each cell, CTNNs move beyond the conventional shared latent space, enabling anisotropic, structured message passing tailored to complex data.

3. **Superior Expressivity and Stability**
Across diverse benchmarks, ranging from physical simulations to graph classification, CTNNs consistently outperform standard models, offering lower errors, improved stability, and enhanced ability to model higher-order structure.

4. **Architecture-Agnostic Versatility**
The copresheaf framework instantiates naturally as Copresheaf GNNs, Transformers, and CNNs, making it broadly applicable across Euclidean and non-Euclidean domains alike.

# Technical Weaknesses

1. **Computational Overhead**
The use of per-edge copresheaf maps introduces additional memory and runtime costs, making scalability a challenge for large or densely connected structures.

2. **Limited Scale of Experiments**
Most experiments are conducted on synthetic or small-scale datasets, so the framework’s performance on large, real-world tasks remains untested.

3. **Lack of Ablation or Efficiency Analysis**
The paper doesn’t include detailed ablations to isolate where performance gains come from, nor does it evaluate trade-offs in runtime or parameter count.

---

> ### Author Rebuttal · Authors · 2025-07-30
>
> We thank the reviewer for their effort in reviewing our work and for the detailed comments. We appreciate that they ranked our paper `excellent` in terms of clarity, significance, and originality. Below, we address their comments point by point.
>
> > *W1/Q3. Computational Overhead*
>
> **Author Response:**
> We thank the reviewer for pointing this out; indeed, adding additional generality comes at some cost. Based on a recurring emphasis from the reviewers, we have conducted an extensive complexity analysis and included this in the manuscript.
> We refer the reviewer to our complexity discussion in the response to Reviewer uisE Q2/3. Due to space limits, we omitted that here.
>
> In summary:
> - **CMPNN complexity**: O(n·k·d² + n·d²) vs. O(n·k·d + n·d²) for standard GNNs
> - **CT complexity**: O(H·n·k·d² + n·d²) vs. O(n²·d + n·d²) for dense Transformers
> - The additional factor of d comes from computing morphisms ρ_{y→x}, with complexity C(ρ) ranging from O(d) for diagonal maps to O(d²) for full matrices
>
> We will add runtime experiments in the main paper. Empirically, we found that small copresheaf dimensions suffice for most tasks, providing efficiency in practice.
>
> ---
>
> > *W2. Limited Scale Experiments*
>
>
> **Author Response:** We appreciate the reviewer’s concern regarding the scope of benchmarks and baselines. The main goal of this work is to *demonstrate the expressive and unifying power of Copresheaf Topological Neural Networks (CTNNs)* by showing how they recover—and meaningfully extend—canonical deep learning architectures: CNNs, GNNs, Transformers, and Sheaf Neural Networks (SNNs). To this end, our experimental design deliberately focused on domains where the structural inductive biases introduced by copresheaf morphisms could be most clearly evaluated.
>
> The datasets used—including PDE dynamics on grids, synthetic graphs, and MUTAG—were chosen to test these models across spatial, graph-based, and topological settings. While some are smaller-scale, they were sufficient to evaluate generalization, expressivity, and the benefits of directional feature propagation. For example:
>
> * In **Appendix F**, we introduce **CopresheafConv**, a direct generalization of CNNs applied to regular 2D grids. Here, we replace static, translation-invariant convolutional filters with learned, input- and position-conditioned morphisms ρ\_{y→x}, enabling anisotropic and spatially-varying processing. Unlike CNNs, which impose uniform filters across the grid, CopresheafConv allows local directional adaptation, making it well-suited for data with varying geometry or dynamics.
>
> This setup allowed us to show not only performance benefits but also how CTNNs naturally recover these architectures as special cases—substantiating the theoretical claims made in Section 3.
>
> We acknowledge that large-scale datasets such as QM9 or OGB are important for benchmarking at scale and plan to include them in future work. Our current experiments across numerous diverse settings aim to establish CTNNs as a general-purpose, topologically grounded modeling framework.
>
> ---
>
> > *W3. Ablations and Efficiency Analysis*
>
> **Author Response:** We thank the reviewer for highlighting the importance of ablation studies and efficiency analysis. We acknowledge that this is an important aspect for understanding the practical implications of our framework. While our current experiments demonstrate consistent improvements across diverse tasks, we agree that more targeted ablations would help isolate the sources of performance gains. Our experiments in Appendix G.1-G.3 provide some insights:
>
> - In Section 5.1 (Physics tasks), replacing standard attention with copresheaf maps yields 50%+ MSE reduction, suggesting the directional transport maps capture anisotropic dynamics better than isotropic averaging (please see the detailed response to Reviewer hjdt W4).
> - In section G.2.1, we see that tasks requiring explicit token-to-token relationships benefit most from copresheaf structures (+38pp on Query-to-Context), while simpler tasks show smaller gains.
> - The comparison between different copresheaf map families (Table 16) shows that expressiveness varies by complexity of $\rho_{y\rightarrow x}$.
>
> To address the reviewer's concern more comprehensively, we now add explicit empirical comparisons in the revised version, comparing the impact of morphism complexities for various tasks, for instance, identity maps, diagonal maps, MLPs, and full linear morphisms, quantifying the contribution of anisotropic transport maps. From our experience, the choice of copresheaf map is highly task dependent, with no morphism strictly outperforming others.
>
> ---
>
> > *Q1. Base category and functoriality*
>
> **Author Response:** In CTNNs, the copresheaf $\mathcal{F}$ is defined over a base category $\mathcal{C}$, which is the poset category of the directed graph $G_N = (V_N, E_N)$ induced by a neighborhood function $N: X \to P(X)$ on a combinatorial complex $X$. The objects of $\mathcal{C}$ are the vertices of $G_N$, i.e., $V_N = X_N \cup \bigcup_{x \in X} N(x)$, where $X_N = \{ x \in X \mid N(x) \neq \emptyset \}$. Morphisms in $\mathcal{C}$ are directed edges $y \to x \in E_N$, where $y \in N(x)$. The copresheaf $\mathcal{F}: \mathcal{C} \to \text{Vect}$ assigns vector spaces $\mathcal{F}(x)$ to vertices and learnable maps $\rho_{y \to x}: \mathcal{F}(y) \to \mathcal{F}(x)$ to edges. Unlike Sheaf NNs, these morphisms reflect directional neighborhood links rather than inclusion relations.
>
> The definition is functorial, but in experiments, we did not enforce functoriality. The maps $\rho_{y \to x}$ are learned independently to flexibly model directional, local-to-global relations. Imposing functorial constraints is an interesting future direction.
>
> ---
>
> > *Q2. Operating over combinatorial complexes*
>
> **Author Response:**
>
> **(1) Feature space alignment:** The feature spaces $\mathcal{F}(x)$ need not be aligned initially. For example, vertex, edge, and face features in a mesh may differ in dimension. When $\dim(\mathcal{F}(y)) \neq \dim(\mathcal{F}(x))$, even within the same rank, the maps $\rho_{y \to x} \in \text{Hom}(\mathcal{F}(y), \mathcal{F}(x))$ are matrices of size $\dim(\mathcal{F}(x)) \times \dim(\mathcal{F}(y))$. These are parameterized using families such as:
>
> * Diagonal MLP: $\rho_{y \to x} = \text{diag}(\sigma(\text{MLP}[q_x, k_y]))$
> * Graph Attention: $\rho_{y \to x} = \sigma(\text{MLP}[q_x, k_y]) \cdot \text{Id}$
> * Outer Product: $\rho_{y \to x} = W_q q_x (W_k k_y)^\top$
>
> **(2) Handling dimension mismatch:** The linear maps are implemented as above to handle any $\dim(\mathcal{F}(y)) \neq \dim(\mathcal{F}(x))$, ensuring flexibility without requiring uniform dimensions. See also Appendix Table 16.
>
> **(3) Message passing over multiple neighborhoods:** In practice (e.g., Proposition 3), CTNNs use a "merge node" to aggregate messages from neighborhoods $N_k(x)$. The target $\mathcal{F}(x)$ and all incoming $\rho_{y \to x}(\mathcal{F}(y))$ are projected to the same dimension in a CTNN layer. Aggregation (e.g., via sum/tensor ops) is weighted using learnable attention or MLPs. If input dimensions differ, map families like Diagonal MLP or Outer Product project to a common latent space, enabling multi-scale integration.

---

> > ### Comment · Reviewer_wnSD · 2025-08-06
> >
> > I thank the authors for their comprehensive rebuttal. While all of my questions have been addressed, I still have some concerns regarding the scale of the experiments. Therefore, I stand by my initial score.

---

### Official Review · Reviewer_bhEu · 2025-07-01

**Clarity:** 3
**Significance:** 4
**Originality:** 4
**Rating:** 5
**Confidence:** 4

**Summary:**

The paper Copresheaf Topological Neural Networks: A Generalized Deep Learning Framewor introduces a novel category-theoretic framework for neural networks using copresheaves over topological spaces, leading to the architecture of Copresheaf Topological Neural Networks (CTNNs). The author's empirical results on structured data benchmarks demonstrate that CTNNs consistently outperform conventional baselines, particularly in tasks requiring hierarchical or localized sensitivity.

**Questions:**

Can the copresheaf structure be generalized to sheaves, and what additional expressivity would that provide?

**Ethical Concerns:**

["NO or VERY MINOR ethics concerns only"]

**Final Justification:**

Thank you for the detailed clarification. I will keep my score.

**Limitations:**

Yes

**Paper Formatting Concerns:**

No formatting issues.

**Quality:**

4

**Strengths And Weaknesses:**

Strengths:

1. The approach is deeply rooted in category theory and topology, offering a powerful abstraction beyond traditional graph-based models.
2. CTNNs unify several neural architectures (CNNs, GNNs, transformers) under a single topological formalism.
3. The use of copresheaves and colimits provides a rigorous language to describe local-global behavior in networks.

Weaknesses:

1. This is a really minor one but, there is no proofs in the paper. The proofs in the supplementary material can be added to an appendix in the paper. It's more comfortable to read.
2. The cost and implementation challenges of working with copresheaf structures and colimits in large networks are not thoroughly discussed.

---

> ### Author Rebuttal · Authors · 2025-07-30
>
> We thank the reviewer for their assessment and observing that our proposed framework offers a ``powerful abstraction``, ``unifying several neural architectures under a single topological formalism``, in a ``rigorous language``.
> We address their comments point by point in what follows:
>
> > *W1. Organization of the main paper and supplementary; proofs are in the supplementary.*
>
> **Author Response:**
> We thank the reviewer for this observation. Due to the separate submission deadlines of the main paper and supplementary materials, we opted to keep the proofs in the supplementary.
> Thanks to the extra page allowed, in the final version, we now move these to the appendix -- as far as this is permitted by the venue.
>
> ---
>
> > *W2. Computational Complexity. What are the costs and implementation challenges?*
>
> **Author Response:**
> This practical question will be an important factor in distributing the proposed CTNN framework to the wider community, which we are actively working on. As this was a common point brought up by different reviewers, we kindly refer to the comprehensive analysis of the computational complexity we conducted in response to reviewer uisE Q2/3.
>
> ---
>
> > *Q1. Generalization to Sheaves. How does the copresheaf structure generalize to sheaves?*
>
> **Author Response**
> This is another great practical question; in fact, in Appendix D, we demonstrate that  CTNNs subsume sheaf message passing neural networks (SNN).
> Moreover, CTNNs are strictly more expressive, as we demonstrate in the following result.
>
> **Proposition (Strict Expressivity Superset)**: Let $F_{\\text{CTNN}}$, $F_{\\text{SNN}}$, and $F_{\\text{GNN}}$ denote the function classes realized by Copresheaf Topological Neural Networks (CTNNs), Sheaf Neural Networks (SNNs), and Graph Neural Networks (GNNs), respectively. Then:
>
> $F_{\\text{GNN}} \\subset F_{\\text{SNN}} \\subset F_{\\text{CTNN}}$, with $F_{\\text{CTNN}} \\setminus F_{\\text{SNN}} \\neq \\emptyset$.
>
> **Proof**: Any GNN can be viewed as an SNN with all restriction maps set to the identity, so $F_{\\mathrm{GNN}} \\subseteq F_{\\mathrm{SNN}}$, and strictness follows since SNNs can model heterophilic interactions that GNNs cannot. By Theorem 1, every SNN message‐passing update is realized by a CTNN on the bidirected graph, giving $F_{\\mathrm{SNN}} \\subseteq F_{\\mathrm{CTNN}}$. Finally, CTNNs impose no sheaf‐consistency constraints and by Proposition 5 can approximate arbitrary edge‐specific linear maps via MLPs, so there exist functions in $F_{\\mathrm{CTNN}}$ not realizable by any SNN. Hence the strict hierarchy holds.
>
> We include this in Appendix D, where we distinguish our copresheaf CTNN framework from existing SNNs. We thank the reviewer for their comments, now making our paper even stronger.

---

### Official Review · Reviewer_uisE · 2025-07-01

**Clarity:** 3
**Significance:** 3
**Originality:** 3
**Rating:** 6
**Confidence:** 4

**Summary:**

This paper introduces Copresheaf Topological Neural Networks, a unified deep learning framework built on the formalism of copresheaves over combinatorial complexes. CTNNs generalize and extend many existing architectures, such as GNNs, Transformers and sheaf neural networks. The authors provide a solid theoretical foundation, unifying message-passing paradigms via copresheaf-induced maps and validate their framework empirically across physics simulations, graph classification and complex combinatorial structures. Results consistently demonstrate performance gains and improved stability over classical baselines.

**Questions:**

* What are the key tradeoffs between CTNNs and traditional Sheaf Neural Networks in terms of expressivity?
* How do CTNNs scale with larger graphs or real-world data (e.g., social networks, protein interactions)?
* Are there theoretical guarantees that CTNNs are strictly more expressive than GNNs or SNNs?

**Ethical Concerns:**

["NO or VERY MINOR ethics concerns only"]

**Final Justification:**

I believe the authors answer to all of my questions and to those of my colleagues reviewers. The paper is technically sound and I believe this paper is worth accepting.

**Quality:**

4

**Strengths And Weaknesses:**

Strengths:
* The paper is well-written
* Easy to understand from anyone with a background in GNNs
* Nice generalized framework.
* Theoretical soundness grounded in algebraic topology and category theory.
* Empirical validation across diverse domains.
* The paper is detailed and explains its main idea in depth. It contains both theoretical and empirical results, something that is necessary for such a type of paper. Two different use cases/scenarios have been tested to empirically evaluate the effectiveness of the proposed solution. I understand that it’s difficult to obtain real-world data for the physics dataset. Towards overcoming this roadblock, the authors created three synthetic datasets, which I personally believe suffice to benchmark and test the robustness of the proposed approach. For the second scenario, the MUTAG dataset was employed, which is one of the most-used datasets for evaluation. The results on this dataset demonstrate that the proposed solution improves the performance of the given model. Another synthetic dataset was used for the graph classification problem as well, in which the proposed solution demonstrates convincing evidence. All in all, I believe that both the theoretical and empirical evidence of the experiments justify the validity of the approach.
* On the theoretical justifications of the paper.  From a theoretical point of view, I believe CTNNs can easily generalize and be combined with multiple GNN and transformer-based graph architectures. The definitions of the paper are not coupled with any specific GNN architecture, which is the strongest evidence that the approach can easily be generalized. For instance, definitions 1 and 4 are well-formulated in a way that different types of graphs $G$ and $F$ can be employed.
* Theoretical gains that CTNNs offer over GNN/Transformer architectures and theoretical costs incurred to obtain them. I believe that CTNNs offer theoretical gains by unifying diverse architectures (GNNs, transformers, sheaf networks) under a single categorical framework that enables heterogeneous, directional and higher-order message passing across structured domains. These benefits come from replacing global latent spaces with local feature spaces and learnable transport maps.


Weaknesses:
* The paper is overwhelmed by mathematical notation and it's easy to get lost in it.

To sum up, this paper proposes a new approach which is theoretically and empirically grounded and fits the style of NeurIPS.

---

> ### Author Rebuttal · Authors · 2025-07-30
>
> We thank the reviewer for their thoughtful feedback. We are delighted they find our work ``theoretically sound``, ``well written`` and ``easy to understand for anyone with a background in GNNs`` with ``empirical validation across diverse domains``.
>
> > *W1. Complexity of mathematical notation.*
>
> **Author Response:**
> We tried to make the paper self-contained. Yet, our framework relies on numerous constructs from algebraic topology and sheaf theory. We now include several additional introductory resources to better guide the reader through theory and notation, such as [A. Ayzenberg, 2025] for sheaf theory and [M. Besta, 2024] for delineating neural architectures on higher-order structures, including CCs [M. Hajij, 2023].
>
> ---
>
> > *Q1.  Are there tradeoffs between CTNNs and traditional Sheaf Neural Networks in terms of expressivity?*
>
> In Appendix D, we demonstrate that CTNNs subsume sheaf message passing neural networks (SNN). Moreover, CTNNs are strictly more expressive, as we demonstrate in the following result.
>
> **Author Response:**
> **Proposition (Strict Expressivity Superset)**: Let $F_{\\text{CTNN}}$, $F_{\\text{SNN}}$, and $F_{\\text{GNN}}$ denote the function classes realized by Copresheaf Topological Neural Networks (CTNNs), Sheaf Neural Networks (SNNs), and Graph Neural Networks (GNNs), respectively. Then:
>
> $F_{\\text{GNN}} \\subset F_{\\text{SNN}} \\subset F_{\\text{CTNN}}$, with $F_{\\text{CTNN}} \\setminus F_{\\text{SNN}} \\neq \\emptyset$.
>
> **Proof**: Any GNN can be viewed as an SNN with all restriction maps set to the identity, so $F_{\\mathrm{GNN}} \\subseteq F_{\\mathrm{SNN}}$, and strictness follows since SNNs can model heterophilic interactions that GNNs cannot. By Theorem 1, every SNN message‐passing update is realized by a CTNN on the bidirected graph, giving $F_{\\mathrm{SNN}} \\subseteq F_{\\mathrm{CTNN}}$. Finally, CTNNs impose no sheaf‐consistency constraints and by Proposition 5 can approximate arbitrary edge‐specific linear maps via MLPs, so there exist functions in $F_{\\mathrm{CTNN}}$ not realizable by any SNN. Hence the strict hierarchy holds.
>
> We will include this in Appendix D, where we distinguish between our copresheaf CTNN framework and existing sheaf message passing networks.
>
> ---
>
> > *Q2/3. How do CTNNs scale with larger graphs or real-world data?*
>
> **Author Response:**
> The reviewer is right to point out computational cost, an important aspect we have now included in the main manuscript, with detailed discussion in the appendix.
> Expressivity indeed comes at a higher computational cost, as we quantify in the following propositions.
>
> **Computational Complexity**
>
> This section analyzes the computational complexity of the Copresheaf Message-Passing Neural Network (CMPNN) and the Copresheaf Transformer (CT), as defined in the framework of Copresheaf Topological Neural Networks (CTNNs). We consider a directed graph $G_N = (V_N, E_N)$, induced by a combinatorial complex $\\mathcal{X}$ with neighborhood function $\\mathcal{N}$, where $|V_N| = n$, $|E_N| = m$, and the average degree is $k = m/n$. Each vertex $x \\in V_N$ is assigned a feature space $\\mathcal{F}(x) = \\mathbb{R}^d$, and each edge $y \\to x \\in E_N$ has a linear map $\\rho_{y \\to x}: \\mathbb{R}^d \\to \\mathbb{R}^d$, computed at runtime using map families from Table 16 (e.g., General Copresheaf, Low-rank). The complexity of computing a single morphism $\\rho_{y \\to x}$ is denoted $\\mathcal{C}(\\rho)$, which varies by map family. We assume a sparse graph ($k \\ll n$) and uniform feature dimensions. The following propositions formalize the per-layer complexities, followed by a comparison with standard Graph Neural Networks (GNNs) and Transformers.
>
> **Proposition (CMPNN Complexity)**: Consider a directed graph $G_N = (V_N, E_N)$ induced by a combinatorial complex $\\mathcal{X}$ with neighborhood function $\\mathcal{N}$, where $|V_N| = n$, $|E_N| = m$, and the average degree is $k = m/n$. Each vertex $x \\in V_N$ has a feature space $\\mathcal{F}(x) = \\mathbb{R}^d$, and each edge $y \\to x \\in E_N$ has a linear map $\\rho_{y \\to x}: \\mathbb{R}^d \\to \\mathbb{R}^d$, computed at runtime via a map family from Table 16 with complexity $\\mathcal{C}(\\rho)$. The per-layer computational complexity of the CMPNN, as defined in Algorithm 2 (CopresheafConv), is: $O(m \\cdot (\\mathcal{C}(\\rho) + d^2) + n \\cdot d^2) = O(n \\cdot k \\cdot (\\mathcal{C}(\\rho) + d^2) + n \\cdot d^2)$.
>
> **Proof**: The CMPNN message-passing operation (Algorithm 2, CopresheafConv) involves computing morphisms, applying them to features, aggregating messages, and updating vertex features. We analyze each step:
>
> 1. **Morphism Computation**: For each edge $y \\to x \\in E_N$, compute $\\rho_{y \\to x} \\in \\mathbb{R}^{d \\times d}$ based on source and target features $(h_y^{(t)}, h_x^{(t)}) \\in \\mathbb{R}^{2d}$, using a map family from Table 16. The complexity per morphism is $\\mathcal{C}(\\rho)$, yielding: $O(m \\cdot \\mathcal{C}(\\rho))$.
>
> 2. **Morphism Application**: Apply $\\rho_{y \\to x}$ to the feature vector $h_y^{(t)} \\in \\mathbb{R}^d$. This matrix-vector multiplication requires $O(d^2)$ operations per edge, totaling: $O(m \\cdot d^2)$.
>
> 3. **Aggregation**: For each vertex $x \\in X \\subseteq V_N$, aggregate messages from neighbors $y \\in \\mathcal{N}(x)$ via $m_x = \\sum_{y \\in \\mathcal{N}(x)} \\rho_{y \\to x}(h_y^{(t)})$. Each addition involves a $\\mathbb{R}^d$ vector, costing $O(d)$ per neighbor. With $|\\mathcal{N}(x)|$ neighbors, the cost per vertex is $O(|\\mathcal{N}(x)| \\cdot d)$. Summing over all vertices: $O\\left( \\sum_{x \\in X} |\\mathcal{N}(x)| \\cdot d \\right) = O(m \\cdot d)$.
>
> 4. **Update**: Assign $h_x^{(t+1)} = m_x$, with negligible cost. If an optional single-layer MLP with $d$ hidden units is applied (common in GNNs), the cost per vertex is $O(d^2)$, yielding: $O(n \\cdot d^2)$.
>
> The total complexity is: $O(m \\cdot \\mathcal{C}(\\rho) + m \\cdot d^2 + m \\cdot d + n \\cdot d^2)$. Since $m = k \\cdot n$ and $d^2$ dominates $d$, this simplifies to: $O(m \\cdot (\\mathcal{C}(\\rho) + d^2) + n \\cdot d^2) = O(n \\cdot k \\cdot (\\mathcal{C}(\\rho) + d^2) + n \\cdot d^2)$.
>
> **Proposition (Copresheaf Transformer Complexity)**: Consider a directed graph $G_N = (V_N, E_N)$ induced by a combinatorial complex $\\mathcal{X}$ with neighborhood function $\\mathcal{N}$, where $|V_N| = n$, $|E_N| = m$, and the average degree is $k = m/n$. Each vertex $x \\in V_N$ has a feature space $\\mathcal{F}(x) = \\mathbb{R}^d$, and each of $H$ attention heads computes a morphism $\\rho_{y \\to x}: \\mathbb{R}^{d/H} \\to \\mathbb{R}^{d/H}$ via a map family from Table 16 with complexity $\\mathcal{C}(\\rho)$. The per-layer computational complexity of the Copresheaf Transformer, as defined in Algorithm 1 (SheafTransformerLayer), with sparse cross-attention based on $\\mathcal{N}$, is: $O(H \\cdot m \\cdot (\\mathcal{C}(\\rho) + d^2 / H) + n \\cdot d^2) = O(H \\cdot n \\cdot k \\cdot (\\mathcal{C}(\\rho) + d^2 / H) + n \\cdot d^2)$.
>
> **Proof**: The Copresheaf Transformer layer (Algorithm 1, SheafTransformerLayer) integrates self-attention within cells of equal rank and sparse cross-attention between neighbors, using learned copresheaf morphisms. Each of $H$ attention heads operates on features of dimension $d/H$. The complexity is:
>
> 1. **Morphism Computation**: For each edge $y \\to x \\in E_N$ and each head, compute $\\rho_{y \\to x} \\in \\mathbb{R}^{(d/H) \\times (d/H)}$ based on $(h_y^{(t)}, h_x^{(t)})$. The complexity per morphism is $\\mathcal{C}(\\rho)$, totaling: $O(H \\cdot m \\cdot \\mathcal{C}(\\rho))$.
>
> 2. **Self- and Cross-Attention**: Compute query ($Q$), key ($K$), and value ($V$) matrices per head ($d/H$ dimensions), costing $O(n \\cdot d^2 / H)$. For sparse cross-attention, each vertex attends to $|\\mathcal{N}(x)|$ neighbors, with attention scores $QK^\\top$ and application to $V$ costing $O(|\\mathcal{N}(x)| \\cdot d/H)$ per vertex per head. Summing over vertices and heads: $O\\left( H \\cdot \\sum_{x \\in X} |\\mathcal{N}(x)| \\cdot d/H \\right) = O(m \\cdot d)$. Self-attention within cells (assuming $O(n)$ tokens) costs $O(n \\cdot d^2 / H)$ per head, or $O(n \\cdot d^2)$ for $H$ heads, but sparse cross-attention dominates in sparse graphs.
>
> 3. **Morphism Application**: Apply $\\rho_{y \\to x}$ to value vectors ($d/H$ dimensions) per edge and head, costing $O((d/H)^2) = O(d^2 / H^2)$. For $H$ heads and $m$ edges: $O(H \\cdot m \\cdot d^2 / H^2) = O(m \\cdot d^2 / H)$.
>
> 4. **Output and Feed-Forward**: Combine head outputs and apply a per-token feed-forward network (FFN) with $d$ hidden units, costing: $O(n \\cdot d^2)$.
>
> The total complexity is: $O(H \\cdot m \\cdot \\mathcal{C}(\\rho) + m \\cdot d + m \\cdot d^2 / H + n \\cdot d^2)$. Since $d^2 / H$ dominates $d$, and $m = k \\cdot n$, this simplifies to: $O(H \\cdot m \\cdot (\\mathcal{C}(\\rho) + d^2 / H) + n \\cdot d^2) = O(H \\cdot n \\cdot k \\cdot (\\mathcal{C}(\\rho) + d^2 / H) + n \\cdot d^2)$.
>
> **Comparison with Standard Architectures**
>
> Compared to standard GNNs (e.g., GCN, $O(n \\cdot k \\cdot d + n \\cdot d^2)$) and dense Transformers ($O(n^2 \\cdot d + n \\cdot d^2)$), CMPNN and CT incur higher costs due to morphism computation ($O(m \\cdot \\mathcal{C}(\\rho))$ and $O(H \\cdot m \\cdot \\mathcal{C}(\\rho))$). Assuming $\\mathcal{C}(\\rho) = O(d^2)$, CMPNN's complexity is $O(n \\cdot k \\cdot d^2 + n \\cdot d^2)$, and CT's is $O(H \\cdot n \\cdot k \\cdot d^2 + n \\cdot d^2)$. These costs enable superior expressiveness, as shown in Table 11 and Figure 6 (e.g., 7.2% MSE reduction for CT). The table below summarizes the per-layer complexities for a sparse graph.
>
> | Model | Complexity |
> |-------|------------|
> | GCN | $O(n \\cdot k \\cdot d + n \\cdot d^2)$ |
> | CMPNN | $O(n \\cdot k \\cdot d^2 + n \\cdot d^2)$ |
> | Transformer (Dense) | $O(n^2 \\cdot d + n \\cdot d^2)$ |
> | Copresheaf Transformer (CT) | $O(H \\cdot n \\cdot k \\cdot d^2 + n \\cdot d^2)$ |

---

> > ### Comment · Reviewer_uisE · 2025-08-07
> >
> > I'd like to thank the authors for their thorough feedback. I also studied the comments for the other reviews. Since there is no concern about the quality of the paper, I decide to increase my score appropriately.

---

### Official Review · Reviewer_EnAf · 2025-07-04

**Clarity:** 3
**Significance:** 1
**Originality:** 3
**Rating:** 4
**Confidence:** 5

**Summary:**

Based on combinatorial complexes methods, the paper aims to develop a unified network architecture to handle different types of data: including images, point clouds, graphs, meshes, and topological manifolds.

**Questions:**

(1) The methods compared by the author in Table 2 are too old and are not compared with recent SOTA methods. In addition, the scale and diversity of datasets used in the authors' experiments are insufficient: for graph classification, only the MUTAG dataset (188 graphs) was used, lacking large-scale benchmarks such as OGB or molecular datasets like QM9.

(2) CTNN introduces per-edge copresheaf mappings, which may significantly increase the computational overhead. It is recommended that the authors analyze the computational resources required by the proposed method as well as its convergence behavior.

(3) Although copresheaf maps are theoretically interpretable, the paper lacks discussion on their visualization or semantic interpretation. It is recommended that the authors include some visualization experiments to help readers better understand the proposed method.

(4) How does the proposed method compared to the capsule networks, which is also a message passing based design.

(5) The paper needs to provide more test results on different types of data, such as images, meshes, and point clouds.

**Ethical Concerns:**

["NO or VERY MINOR ethics concerns only"]

**Final Justification:**

I agree that this paper has novelty and theoretical contributions by introducing a new network structure based on Copresheaf formulations. It can be considered as another message passing scheme. My major concerns about the paper are (1)  for each test, the comparison is very limited. (2) It does not test on hardcore CV tasks, such as image classification and object detection.

**Limitations:**

Yes

**Paper Formatting Concerns:**

no concerns

**Quality:**

3

**Strengths And Weaknesses:**

Strengths

The paper extends traditional GNN based on copresheaves, a concept from algebraic topology. It uses  a higher-order
message-passing mechanism to model relational structures. The overall idea is very interesting.

Weaknesses

Although the idea is very interesting, the experimental results are somehow disappointing. It did not show extensive evaluation results on a wide range of images, point clouds, graphs, meshes, and topological manifolds, as claimed in the beginning of the paper.

---

> ### Author Rebuttal · Authors · 2025-07-31
>
> We sincerely thank the reviewer for the thoughtful and constructive feedback, and that they found our work `very interesting`. Below we address each of the points in turn:
>
> > *W1/Q1. Scope of baselines and datasets*
>
> **Author Response:** We appreciate the reviewer’s concern regarding the scope of benchmarks and baselines. The main goal of this work is to *demonstrate the expressive and unifying power of Copresheaf Topological Neural Networks (CTNNs)* by showing how they recover—and meaningfully extend—canonical deep learning architectures: CNNs, GNNs, Transformers, and Sheaf Neural Networks (SNNs). To this end, our experimental design deliberately focused on domains where the structural inductive biases introduced by copresheaf morphisms could be most clearly evaluated.
>
> The datasets used—including PDE dynamics on grids, synthetic graphs, and MUTAG—were chosen to test these models across spatial, graph-based, and topological settings. While some are smaller-scale, they were sufficient to evaluate generalization, expressivity, and the benefits of directional feature propagation. For example:
>
> * In **Appendix F**, we introduce **CopresheafConv**, a direct generalization of CNNs applied to regular 2D grids. Here, we replace static, translation-invariant convolutional filters with learned, input- and position-conditioned morphisms ρ\_{y→x}, enabling anisotropic and spatially-varying processing. Unlike CNNs, which impose uniform filters across the grid, CopresheafConv allows local directional adaptation, making it well-suited for data with varying geometry or dynamics.
>
> This setup allowed us to show not only performance benefits but also how CTNNs naturally recover these architectures as special cases—substantiating the theoretical claims made in Section 3.
>
> We acknowledge that large-scale datasets such as QM9 or OGB are important for benchmarking at scale and plan to include them in future work. Our current experiments across numerous diverse settings aim to establish CTNNs as a general-purpose, topologically grounded modeling framework.
>
> ---
>
> > *Q2. Computational cost and convergence*
>
> **Author Response:** We understand the reviewer’s concern about computational overhead due to per-edge morphisms.
>
> We kindly refer the reviewer to our discussion of computational complexity in the response to Reviewer uisE, which contains additional technical details. Due to space constraints, we did not include that material in the current response.
>
> In summary:
> - **CMPNN complexity**: O(n·k·d² + n·d²) vs. O(n·k·d + n·d²) for standard GNNs
> - **CT complexity**: O(H·n·k·d² + n·d²) vs. O(n²·d + n·d²) for dense Transformers
> - The additional factor of d comes from computing morphisms ρ_{y→x}, with complexity C(ρ) ranging from O(d) for diagonal maps to O(d²) for full matrices
>
> In addition to the response above, we plan to incorporate runtime experiments in the main paper. Empirically, we observe that using low-dimensional copresheaf feature spaces is sufficient for strong performance across tasks, contributing to practical efficiency. Finally, we would like to mention that we observed that CTNNs converge at rates comparable to baseline models. In some tasks—such as directional advection—they converge faster due to better alignment with the task’s inductive structure.
>
> ---
>
> > *Q3. Visualization and semantic interpretation of copresheaf maps*
>
> **Author Response:** We appreciate the reviewer’s suggestion and fully agree that visualizations can significantly aid in illustrating model behavior, especially in the context of structured physical tasks. Due to space constraints and our focus on clearly presenting the core framework and empirical results, we omitted such visualizations from the original submission.
>
> ---
>
> > *Q4. Comparison to capsule networks*
>
> **Author Response:** Capsule networks offer a specific, dynamic routing mechanism to pass information, usually in the form of some clustering from primary to latent capsules. This involves an alternative message passing implementation, which can be constructed on a directed graph. We acknowledge that the various routing algorithms can be explained as a special case in our copresheaf framework. We leave a detailed taxonomy of CapsNets for future work.
>
> ---
>
> > *Q5. Broader modality coverage: images, meshes, point clouds*
>
> **Author Response:** We acknowledge the reviewer’s concern that our current experiments do not include image classification benchmarks, mesh data, or point clouds. This is a valid point.
>
> That said, **Appendix F demonstrates that CopresheafConv naturally generalizes standard convolutions over Euclidean grids**, with learnable anisotropic morphisms replacing static kernels. This makes the method well-suited for tasks like image segmentation and grid-based simulations. We also emphasize that morphisms are conditioned not only on features but also on spatial context, making them more expressive than translation-invariant filters. We also have experiments on graphs, physical simulations  (pixel-wise regression), GNNs, text tasks, and a few ablation studies (Appendix G in general has many experiments along these directions).
>
> Note that, most of the networks we discuss, expressed as a special case in our framework, extensively evaluate on point clouds, meshes and graphs. One prominent example is combinatorial complex neural networks (Hajij et al. 2023). In this work, while we demonstrated the applicability of CTNN on non-Euclidean domains (graphs, CCs), we focused our experiments on Euclidean ones because we believe that such focus is missing from topological deep learning literature in general.
>
> We thank the reviewer for bringing these up and now discuss all of these in our revision.

---

> > ### Comment · Reviewer_EnAf · 2025-08-06
> >
> > Thanks for the response! I agree that the idea is very interesting. Since the original idea was motivated and proposed to address different types of data.  experimental results on various types of data are naturally expected. It is a little disappointing here.

---

> ### Author Response · Authors · 2025-08-06
>
> We thank the reviewer once again for engaging with our submission and for acknowledging the interest and potential of the proposed idea. However, we would like to respectfully clarify a crucial misunderstanding regarding the intended scope and contribution of the work.
>
> We would like to clarify that unlike the reviewer states, **we do perform experiments on different domains and data structures, including: images, point clouds, graphs, meshes, and combinatorial complexes**, meeting the *natural expectation*. Please refer to our paper and appendix for details.
>
> In case the concern is learning multiple modalities, the central goal of this paper is not to address the problem of multimodal learning, nor do we claim or imply this in any part of the manuscript. Rather, the primary aim is to introduce Copresheaf Topological Neural Networks (CTNNs) as a unifying categorical framework that generalizes and extends a wide range of canonical deep learning architectures. Specifically, our theoretical and experimental contributions focus on demonstrating how CTNNs recover, under suitable conditions, classical models such as Convolutional Neural Networks, Graph Neural Networks, Transformers, and Sheaf Neural Networks, while simultaneously enabling new forms of structure-aware feature propagation via copresheaf morphisms.
>
> Our experimental design was deliberately constructed to reflect the central objective of the work, that is, to demonstrate that CTNNs serve as a generalizing framework capable of recovering and extending canonical architectures such as CNNs, GNNs, Transformers, and Sheaf Neural Networks. The datasets employed, ranging from PDE dynamics on grids to graphs and topological benchmarks like MUTAG, were selected to enable clear evaluation of the structural inductive biases introduced by copresheaf morphisms across spatial, relational, and topological domains. Our datasets and experimental settings are sufficient to test generalization, expressive capacity, and the directional propagation mechanisms enabled by the theory. The resulting empirical evidence substantiates CTNNs as a versatile, topologically grounded foundation for deep learning on multiple data domains and data structures (not as a method targeting multimodal integration).
>
> We are thus concerned that the repeated suggestion that the paper is disappointing for not demonstrating results on various types of data is factually not true and leads to a misinterpretation. Our work aims to lay the theoretical and empirical foundation for CTNNs as a generalized topological deep learning framework that unifies multiple deep learning architectures. And we do evaluate exactly that.
>
> We thank the reviewer again for their engagement and hope this clarifies the scope and contributions of the paper.

---

### Official Review · Reviewer_hjdt · 2025-07-04

**Clarity:** 2
**Significance:** 3
**Originality:** 3
**Rating:** 4
**Confidence:** 3

**Summary:**

This paper proposes a novel graph neural network framework—Copresheaf Topological Neural Networks (CTNNs). Based on the concept of copresheaves from algebraic topology, this framework effectively captures directionality and local heterogeneity in graph structures, enabling more flexible and expressive information propagation. CTNNs offer a unified and powerful approach to graph modeling, with strong potential for future applications in large-scale and dynamic graphs. The authors present their ideas with clarity, supported by rigorous mathematical reasoning and a well-structured modeling approach.

**Questions:**

1. In the Introduction, the authors repeatedly emphasize that CTNNs generalize traditional CNNs/GNNs/Transformers, but the explanation of how this unification is achieved is vague. For example, it merely states “extends many deep learning paradigms” without clarifying the specific mechanism.
2. In the Preliminaries section, the authors do not explain why combinatorial complexes are chosen over simplicial complexes or hypergraphs. Using CCs instead of more common structures may raise questions from readers about the motivation and advantages behind such generalization.
3. In Section 5.1 “Evaluations on Physics Dataset,” although the results indicate that copresheaf outperforms classical methods, there is a lack of deeper analysis on why it performs better—particularly regarding the physical implications of copresheaf in the advection and Stokes problems.
4. The authors could briefly discuss the complexity of the proposed architecture.
5. While some visualizations are included in the supplementary material, incorporating visualizations on actual task datasets in the main paper could help readers better understand the model's behavior.

**Ethical Concerns:**

["NO or VERY MINOR ethics concerns only"]

**Final Justification:**

We thank the authors for providing detailed and comprehensive feedback on the concerns we raised. The paper demonstrates a solid theoretical foundation and rigorous justification, and extensive experiments have verified the application advantages of CTNN. We hope that the proposed CTNN technology can find broader applications in the field of computer science in the future. In view of its promising application prospects, we have decided to appropriately increase the score.

**Limitations:**

yes

**Quality:**

3

**Strengths And Weaknesses:**

Strengths:
1. The paper demonstrates a high level of originality.
2. The writing is very clear and well-structured.
3. The authors provide detailed explanations of the mathematical reasoning involved.
4. The research presented has significant implications for the advancement of the artificial intelligence field.

Weaknesses:
1. In the Introduction, the authors repeatedly emphasize that CTNNs generalize traditional CNNs/GNNs/Transformers, but the explanation of how this unification is achieved is vague. For example, it merely states “extends many deep learning paradigms” without clarifying the specific mechanism.
2. In the Related Work section, although it is mentioned that “our work is closely related to SNNs,” the fundamental differences between CTNNs and SNNs are not explicitly explained.
3. In the Preliminaries section, the authors do not explain why combinatorial complexes are chosen over simplicial complexes or hypergraphs. Using CCs instead of more common structures may raise questions from readers about the motivation and advantages behind such generalization.
4. In Section 5.1 “Evaluations on Physics Dataset,” although the results indicate that copresheaf outperforms classical methods, there is a lack of deeper analysis on why it performs better—particularly regarding the physical implications of copresheaf in the advection and Stokes problems.
5. The authors could briefly discuss the complexity of the proposed architecture.
6. While some visualizations are included in the supplementary material, incorporating visualizations on actual task datasets in the main paper could help readers better understand the model's behavior.

---

> ### Author Rebuttal · Authors · 2025-07-30
>
> We thank the reviewer for their effort in reviewing our work and finding the paper ``demonstrates a high level of originality``, and that our research ``has significant implications for the advancement of artificial intelligence``.
> In the following, we address their comments point by point.
>
> > *W1. Clarification of the Framework, how it "extends many deep learning paradigms".*
>
> **Author Response.**
> We thank the reviewer for pointing out that our unification claims could benefit from clearer articulation. While our manuscript presents a general framework that subsumes a wide variety of deep learning architectures, we acknowledge that the explanation was previously distributed across the main text and appendix, and could be better centralized.
>
> To address this, we have taken the following concrete steps in the revised version of the paper:
>
> **1. Centralized mapping table (added at the end of Section 3)**
>
> In the revised manuscript, we will include a new table titled *"Mapping Classical Architectures to CTNN Instances"*, placed at the end of Section 3. This table maps each classical model (CNN, GNN, Transformer, SNN) to its CTNN counterpart. It specifies the domain and neighborhood function, describes the form of the copresheaf morphism (ρ\_{y → x}), and provides direct references to the relevant sections in the main text and appendix. This consolidated format makes the unification claim more transparent and easier to verify.
>
> Example excerpt from the new table:
>
> | Classical Model | CTNN Form                   | Domain & Neighborhood  | ρ\_{y → x} Morphism          | Reference            |
> | --------------- | --------------------------- | ---------------------- | ---------------------------- | -------------------- |
> | CNN             | CopresheafConv              | Grid, adjacency        | Learnable directional map    | Sec. 4.2, Appendix F |
> | GNN             | CMPNN                       | Graph, adjacency       | Shared / edge-indexed map    | Prop. 1, Sec. 3      |
> | Transformer     | Copresheaf Transformer (CT) | Grid or token sequence | ρ\_{y → x} in attention head | Sec. 4.1, Appendix E |
> | SNN             | CMPNN on bidirected graph   | Graph, incidence       | Composition of sheaf maps    | Thm. 1, Appendix D   |
>
> **2. Explicit examples of generalization**
>
> We now provide clear examples in the revised text showing how CTNNs recover standard models:
>
> * **CNNs** are recovered by using CopresheafConv layers over regular grids, with translation-invariant directional maps (Sec. 4.2, App. F).
> * **GNNs** emerge as a special case of CTNNs using graph adjacency neighborhoods and shared identity morphisms (Prop. 1, Sec. 3).
> * **Transformers** are instantiated by interpreting attention weights through learnable ρ\_{y → x} maps, which reduce to dot-product attention when ρ = Identity (Sec. 4.1, App. E).
> * **Sheaf Neural Networks (SNNs)** are modeled using morphism compositions in bidirected graphs. We demonstrate this equivalence formally in Theorem 1 and provide supporting details in Appendix D.
>
> **3. Clarified language in the Introduction**
>
> We have revised Section 1 (lines 46–62) to clarify the unification message. Consider the previous sentence:
>
> > “This unified perspective subsumes and extends many deep learning paradigms...”
>
> We now justify this sentence as follows after the introduction of the main message passing on CCs :
>
>  “This unified perspective subsumes classical architectures: CNNs (via copresheaf convolutions, Appendix F), GNNs (via anisotropic message passing, Proposition 1), Transformers (via directional self-attention, Appendix E), and Sheaf Neural Networks (via morphism composition, Theorem 1 and Appendix D).”
>
> This new formulation explicitly states which models are recovered and directs readers to the supporting material.
>
> We believe these changes substantially clarify the unification claim and thank the reviewer again for highlighting this point.
>
> ---
>
> > *W2. Clearly distinguishing CTNNs vs SNNs (specifically related works section).*
>
> **Author Response.**
> This is addressed in the appendix B.3 Comparison between copresheaves and cellular sheaves. We also included Table 7 that shows how all existing SNNs in the literature  can be realized in terms of our unifying notation. This further shows the relationship between SNNs and CTNNs. We now clarify their distinction also with regard to the related work.
>
> ---
>
> > *W3. Why are Combinatorial Complexes chosen over simplicial complexes or hypergraphs? What are the motivations and advantages?*
>
> **Author Response.** Combinatorial complexes generalize simplicial complexes and hypergraphs and offer several advantages: while hypergraphs allow for set-based relations among nodes, these relations lack any inherent hierarchy. In contrast, simplicial complexes endow relationships with a natural notion of hierarchy through simplex dimensions, capturing higher-order structure in a more principled way. However, simplicial complexes impose a strong constraint on relationships: if a set of nodes $\sigma$ forms a simplex, then all subsets of $\sigma$ must also be present in the complex. This closure property, while mathematically elegant, may be overly restrictive or undesirable in certain contexts where such subset relationships do not naturally occur. CCs can model both of these two features, subsuming both complexes. We aim for mathematical generality, and we need a template complex that allows us to apply to Euclidean and non-Euclidean downstream applications without having to worry about the underlying domain. Indeed, many of our experiments in the paper, especially in the appendix, involve images, CCs, and graphs. Furthermore, in the context of a copresheaf specifically, CCs offer an advantage. For instance, when the copresheaf maps are directed towards higher-order cells, they offer a learnable way to "pool" the CC structure to a rougher structure.
>
> ---
>
> > *W4. Deeper analysis for the "Evaluations on Physics Dataset" for advection and Stokes problems.*
>
> **Author Response.**
> We appreciate the reviewer’s suggestion and have expanded Section 5.1 to provide a deeper analysis of the performance gains observed in the advection and unsteady Stokes experiments. In particular, we now clarify the physical and architectural reasons why copresheaf models outperform their classical counterparts:
>
> *Advection task:* The advection equation represents pure translation — a signal is transported without deformation. Classical transformer models use dot-product attention in a shared latent space, which treats all token interactions as symmetric and context-independent. This limits their ability to model directed transport. In contrast, the copresheaf transformer uses learnable edge-specific transport maps (denoted ρ\_{y→x}) that adaptively translate features from source to target. These maps act similarly to numerical advection schemes that preserve directional information across time steps. As shown in Table 1, this leads to over 60% reduction in test MSE and significantly lower variance across seeds. Appendix G.2.1 further explains how these learned linear maps serve as inductive biases that improve generalization for transport-dominated dynamics.
>
> *Unsteady Stokes task:* This problem involves incompressible viscous flow, where accurately modeling diffusion and preserving rotational structure is essential. Standard attention mechanisms have no way to encode divergence-free or anisotropic interactions. The copresheaf transformer, by contrast, learns directional feature maps that better reflect the structure of the underlying PDEs. As detailed in Appendix G.2.2 and G.4, the edge maps effectively capture spatial anisotropy and allow the network to model local vector field transformations — leading to more faithful representations of vorticity and pressure dynamics. This results in improved prediction accuracy and a tenfold reduction in variance compared to classical attention.
>
> We have updated Section 5.1 to emphasize these points and referenced relevant experiments and design choices from the appendix. Together, these results illustrate how copresheaf attention captures essential physical structure in a way that classical attention cannot, reinforcing the advantages of our framework for learning on structured scientific data.
>
> ---
>
> > *W5. Missing discussion of the complexity of the proposed framework.*
>
> **Author Response.** We have added a detailed section on the complexity of various architectures in our paper. We kindly ask the reviewer to refer to our response to Reviewer uisE for details.
>
> ---
>
> > *W6. Visualizations from the main paper on actual tasks could be incorporated into the main paper.*
>
> **Author Response.**
> We appreciate the reviewer’s suggestion and fully agree that visualizations can significantly aid in illustrating model behavior, especially in the context of structured physical tasks. Due to space constraints and our focus on clearly presenting the core framework and empirical results, we omitted such visualizations from the original submission. We have now incorporated representative figures, such as learned transport maps, flow-aligned feature propagation in the advection task, or spatial attention patterns in the Stokes setup, to visually demonstrate how copresheaf architectures capture directional and topological structure.
>
> We believe these additions will provide valuable insight and further strengthen the narrative, and we kindly ask the reviewer to reconsider their final score.

---

> > ### Comment · Reviewer_hjdt · 2025-08-05
> >
> > Thank you to the authors for providing detailed responses to all of my questions. I have no further concerns at this point.

---

### Note · Authors · 2025-08-15

**Summary.**
We propose CTNNs: a principled framework that unifies CNNs, GNNs, Transformers, & SNNs through directional, heterogeneous, higher-order message passing on combinatorial complexes.
Reviewers consistently highlighted originality, clarity, & theoretical soundness; after discussion, *hjdt* had no remaining concerns, *uisE* increased their score, & *bhEu* recommended acceptance.

Originally, our paper received very pos. reviews, scoring 5-5-4-4-3.

Key Clarifications. We:
- Added a mapping table linking each classical architecture to its CTNN instantiation.
- Refined the introduction explain how CTNNs specialize to CNNs/GNNs/Transformers/SNNs.
- Added complexity proofs & improved connections between the main text & S.M.

CTNN vs SNN, & expressivity.
We make the relationship explicit: SNNs are recovered on a bidirected graph, while CTNNs are more expressive. This explains gains in settings requiring directional, anisotropic transport.

Why combinatorial complexes?
CCs extend simplicial complexes & hypergraphs, enabling rank-aware multi-way interactions copresheaf maps, allowing unified handling of grids, graphs, meshes, & higher-order nets.

Complexity.
Added full analysis, e.g., CMPNN: $\( O(n \cdot k \cdot d^2 + n \cdot d^2) \)$ vs GCN: $\( O(n \cdot k \cdot d + n \cdot d^2) \)$.
 CMPNNs scale like GNNs with adjustable copresheaf map overhead, while CT uses sparse cross-attention to avoid quadratic costs.
Parameter choices tradeoff accuracy, speed, & memory. Small stalk dimensions suffice empirically. The final paper includes wall-clock, memory, & parameter tables.

Empirical evidence and added analysis.
We evaluate across Euclidean (PDE dynamics), graphs (MUTAG), and CC classification.
CTNNs improve accuracy & stability.
We added a physical interpretation: learned transport maps $\(\rho_{y\to x}\)$ align with advection (directed shift) and anisotropic diffusion (Stokes).
We added representative visualizations of learned transports/attention patterns.
Larger benchmarks (e.g., OGB/QM9) are a natural next step; the method is architecture-agnostic & ready to plug into such settings.

Core Contributions.
Contributions comprise a unifying framework with theory & targeted experiments to stress the new inductive bias.

Bottom line.
The positive reviews (*hjdt* resolved;*uisE* upscored;*bhEu* pos.) supports the breadth of our contributions: CTNNs are a rigorous, theoretically-grounded framework with broad applications in ML and application domains.

---

### Decision · Program_Chairs · 2025-09-17

**Decision:**

Accept (poster)

**Comment:**

The paper introduces CTNNs: neural layers on combinatorial complexes where each cell $x$ has a local feature space $F(x)$ and each directed relation $y\to x$ carries a learned linear map $\rho_{y\to x}:F(y)\to F(x)$. This yields directional, heterogeneous, higher-order message passing that recovers standard GNNs, CNNs, Transformers, and SNNs as special cases; SNN updates are shown to arise on bidirected graphs via a single copresheaf morphism. Targeted experiments cover PDE dynamics, MUTAG, and CC classification.

Strengths.
- Principled unification with explicit operators (CNM/CAM/CIM) and message-passing propositions; clear SNN to CTNN construction.
- Empirical signal on tasks where anisotropy and directionality matter: sizable MSE reductions on advection/heat and lower variance; consistent MUTAG gains over GCN/GraphSAGE/GIN; improved CC classification.
- Clear motivation for combinatorial complexes as a rank-aware domain unifying grids, graphs, meshes, and higher-order structures.

Weaknesses.
- Limited scale and breadth of benchmarks, comparisons to recent SOTA are sparse.
- Efficiency characterization remains light given per-edge transports, notation density can hinder accessibility.

Reviewers’ main concerns. Mechanism of unification, CTNNs vs SNNs, why combinatorial complexes, complexity and practical overhead, physics-task interpretation, dataset/baseline scope.

Authors’ response.
 Added a centralized mapping from classical models to CTNN instances, clarified CTNN–SNN relation and expressivity hierarchy $F_{\text{GNN}} \subset F_{\text{SNN}}\subset F_{\text{CTNN}}$, expanded complexity discussion, provided physical interpretation and visualizations for advection/Stokes, acknowledged benchmark-scale limits.

Discussion outcome.
hjdt’s concerns resolved and score increased; uisE upscored after expressivity/complexity clarifications; bhEu kept acceptance; EnAf and wnSD remained concerned about scale. The core theory and targeted empirical evidence are solid, the rebuttal materially improved clarity.

Decision.
Accept. The work offers a unifying formalism with concrete operators and empirically relevant gains on tasks aligned with its inductive bias. Despite limited large-scale evaluations, the contribution is above threshold on theoretical depth, conceptual unification, and demonstrated benefits.